# Is a verification phase useful for confirming maximal oxygen uptake in apparently healthy adults? A systematic review and meta-analysis

**Victor A. B. Costa**[1,2], **Adrian W. Midgley**[3], **Sean Carroll**[4☯], **Todd A. Astorino**[5☯], **Tainah de Paula**[6☯], **Paulo Farinatti**[1,2☯], **Felipe A. Cunha**[1,2]*

**1** Graduate Program in Exercise Science and Sports, University of Rio de Janeiro State, Rio de Janeiro, Brazil, **2** Laboratory of Physical Activity and Health Promotion, University of Rio de Janeiro State, Rio de Janeiro, Brazil, **3** Department of Sport and Physical Activity, Edge Hill University, Ormskirk, Lancashire, England, **4** Department of Sport, Health and Exercise Science, University of Hull, Hull, England, **5** Department of Kinesiology, California State University, San Marcos, California, United States of America, **6** Department of Clinical Medicine, Clinics of Hypertension and Associated Metabolic Diseases, University of Rio de Janeiro State, Rio de Janeiro, Brazil

☯ These authors contributed equally to this work.

\* felipeac@globo.com, felipe.cunha@uerj.br

**Data Availability Statement:** All relevant data are within the paper and its Supporting information files.

## Abstract

### Background

The 'verification phase' has emerged as a supplementary procedure to traditional maximal oxygen uptake (VO$_{2max}$) criteria to confirm that the highest possible VO$_2$ has been attained during a cardiopulmonary exercise test (CPET).

### Objective

To compare the highest VO$_2$ responses observed in different verification phase procedures with their preceding CPET for confirmation that VO$_{2max}$ was likely attained.

### Methods

MEDLINE (accessed through PubMed), Web of Science, SPORTDiscus, and Cochrane (accessed through Wiley) were searched for relevant studies that involved apparently healthy adults, VO$_{2max}$ determination by indirect calorimetry, and a CPET on a cycle ergometer or treadmill that incorporated an appended verification phase. RevMan 5.3 software was used to analyze the pooled effect of the CPET and verification phase on the highest mean VO$_2$. Meta-analysis effect size calculations incorporated random-effects assumptions due to the diversity of experimental protocols employed. I$^2$ was calculated to determine the heterogeneity of VO$_2$ responses, and a funnel plot was used to check the risk of bias, within the mean VO$_2$ responses from the primary studies. Subgroup analyses were used to test the moderator effects of sex, cardiorespiratory fitness, exercise modality, CPET protocol, and verification phase protocol.

**Funding:** The present systematic review and meta-analysis derived from a research project involving cardiorespiratory fitness assessment in healthy and clinical populations only financed for material support from the Carlos Chagas Filho Foundation for the Research Support in Rio de Janeiro (FAPERJ, E-26/202.70 /2019, recipient FC; E-26/202.880/2017, recipient PF) and Brazilian Council for Technological and Scientific Development (CNPq, 303629/2019-3, recipient PF). The funders had no role in study design, data collection, and analysis, decision to publish, or preparation of the manuscript. The authors have not received a salary from any of your funders.

**Competing interests:** The authors have declared that no competing interests exist.

## Results

Eighty studies were included in the systematic review (total sample of 1,680 participants; 473 women; age 19–68 yr.; $VO_{2max}$ 3.3 ± 1.4 L/min or 46.9 ± 12.1 mL·kg$^{-1}$·min$^{-1}$). The highest mean $VO_2$ values attained in the CPET and verification phase were similar in the 54 studies that were meta-analyzed (mean difference = 0.03 [95% CI = -0.01 to 0.06] L/min, $P = 0.15$). Furthermore, the difference between the CPET and verification phase was not affected by any of the potential moderators such as verification phase intensity ($P = 0.11$), type of recovery utilized ($P = 0.36$), $VO_{2max}$ verification criterion adoption ($P = 0.29$), same or alternate day verification procedure ($P = 0.21$), verification-phase duration ($P = 0.35$), or even according to sex, cardiorespiratory fitness level, exercise modality, and CPET protocol ($P = 0.18$ to $P = 0.71$). The funnel plot indicated that there was no significant publication bias.

## Conclusions

The verification phase seems a robust procedure to confirm that the highest possible $VO_2$ has been attained during a ramp or continuous step-incremented CPET. However, given the high concordance between the highest mean $VO_2$ achieved in the CPET and verification phase, findings from the current study would question its necessity in all testing circumstances.

## PROSPERO Registration ID

CRD42019123540.

## Introduction

Maximal oxygen uptake ($VO_{2max}$) represents the upper physiological limit of the utilization of oxygen for producing energy during strenuous exercise performed until volitional exhaustion [1, 2]. The $VO_{2max}$ is widely regarded as the gold standard measure of cardiorespiratory fitness and is typically determined using a cardiopulmonary exercise test (CPET) in clinical, applied physiology, and sport and exercise science settings [1, 3–6]. The $VO_{2max}$ is often used to diagnose cardiovascular disease [7], predict all-cause mortality [8–10], develop exercise prescriptions [3, 11, 12], and evaluate the efficacy of exercise programmes [13–15]. Consequently, the validity of $VO_{2max}$ values obtained during CPETs has widespread importance in clinical, sporting, and research-related contexts.

The use of indirect calorimetry for the determination of $VO_{2max}$ during exercise testing to volitional exhaustion on a treadmill or cycle ergometer has become common during the past few decades [16–18]. This has largely been attributed to the development of fast-responding metabolic gas analyzers allowing the time-efficient acquisition of real-time, breath-by-breath, respiratory gas exchange and flow rate data during CPET [see 19 for a review]. These technological advances have contributed to a transition from the Douglas bag method and time-consuming discontinuous step-incremented protocols to more time-efficient continuous ramp or pseudo-ramp protocols for determining $VO_{2max}$ [20–25]. Despite the considerable progress in the efficiency by which CPET can be conducted and evaluated, there is still much to be learned about the determination of $VO_{2max}$ [2, 24–30]. One particularly problematic aspect has been

the challenge in identifying a lack of $VO_{2max}$ attainment due to inappropriate test protocols, premature fatigue, or poor participant motivation and lack of effort [31].

The concept of a $VO_{2max}$ originated almost 100 years ago with the seminal works of Hill and colleagues [32, 33]. They proposed the existence of an individual upper limit or 'ceiling' of $VO_2$ during maximal exercise, beyond which no further increase in $VO_2$ occurs despite increasing work rate (WR) and higher metabolic demand. The primary criterion for confirming that a $VO_{2max}$ has been elicited has historically been based on the occurrence of a $VO_2$ plateau, commonly defined as a small or no increase in $VO_2$ despite a continued increase in WR [34]. The landmark study of Taylor et al. [34] was the first to use a formal $VO_2$ plateau criterion, which was defined as an increase in $VO_2$ of less than 0.150 L/min (or $\leq 2.1$ mL·kg$^{-1}$·min$^{-1}$, considering an average body mass of 72 kg from 115 male participants) in response to a specific discontinuous step-incremented protocol performed over 3–5 laboratory visits. Subsequent studies have often used the Taylor et al. [34] criterion or alternative thresholds to confirm the attainment of a $VO_2$ plateau [see 29 for a review]. Since the widespread adoption of continuous short-duration and ramp-based CPET protocols, several studies have reported low incidences of the $VO_2$ plateau [35–39]. The variability in $VO_2$ plateau incidence has been attributed to differences in the criteria used for detecting the $VO_2$ plateau [29, 40], $VO_2$ sampling intervals [36, 41, 42], exercise modality [43], the warm-up prior to the CPET [44], type of CPET protocol [45–48], and various participant characteristics [49–51].

In the absence of a $VO_2$ plateau, secondary $VO_{2max}$ criteria based upon achievement of threshold values for the respiratory exchange ratio (RER), percentage of age-predicted maximal heart rate, post-exercise blood lactate concentration, and ratings of perceived exertion (RPE) have become commonly used to evaluate whether a true $VO_{2max}$ has been attained [29, 40]. However, this approach has been widely criticized by numerous investigators due to the individual variability in maximal physiological responses for these variables and lack of specificity in identifying individuals who did not continue the CPET to their limit of exercise tolerance. Research has shown that some individuals can satisfy some of the secondary criteria thresholds long before the highest $VO_2$ value observed in the CPET has been attained [2, 29, 37, 39]. The maximal RER criterion, for example, can be satisfied at $VO_2$ values 27–39% lower than the highest $VO_2$ value achieved in the CPET [37, 39]. Like the $VO_2$ plateau, secondary $VO_{2max}$ criteria are often dependent on exercise modality, test protocol, and participant characteristics [29].

A review by Midgley et al. [29] suggested a new set of standardized $VO_{2max}$ criteria should be developed that are independent of exercise modality, test protocol, and participant characteristics, so they can be universally applied. In 2009, Midgley and Carroll [28] provided an early narrative review of an evolving test procedure that showed promise for developing more standardized $VO_{2max}$ criteria, the so-called 'verification phase'. The verification phase consists of an appended square wave bout of severe-intensity exercise (e.g. above critical power), or similar multistage exercise bout, performed until the limit of exercise tolerance [28]. It is commonly applied after a short recovery period from a CPET, however, longer recovery periods of up to 24–48 hours also have been used [52]. The verification phase is based on the premise that when the highest $VO_2$ values in the CPET are consistent with the verification phase (typically within 2–3% in accordance with the test-retest reliability of $VO_{2max}$), this provides substantial empirical support that the highest possible $VO_2$ has been elicited. Poole and Jones [2] recently stated that to confirm the attainment of $VO_{2max}$ a verification phase should be performed at a higher WR than the last load attained in the CPET (i.e. $>$ WR$_{peak}$) in all future studies. Conversely, Iannetta et al. [25] recommended WRs within the upper limit of the severe exercise intensity domain to allow the verification phase to be maintained long enough for $VO_{2max}$ attainment. According to their recent findings, verification phases performed at 110% of the

$WR_{peak}$ attained during CPETs with increment rates of 25 and 30 W/min resulted in exercise durations that were too short to allow $VO_2$ to reach the highest $VO_2$ recorded at the end of the preceding ramp CPETs [25]. Along with exercise intensity and duration, it is also unclear whether other factors affect the utility of the verification phase such as exercise modality, differences in the type and duration of the recovery period between the verification phase and CPET, whether a verification criterion threshold is adopted, and participant characteristics such as sex and cardiorespiratory fitness levels.

Given the considerable uncertainty regarding the application of the verification phase, it is feasible to think that a systematic review and meta-analysis is needed to comprehensively summarize the evidence for improving our understanding of the strengths and weaknesses of the substantial number of different verification procedures that have been utilized and its impact on the attainment of $VO_{2max}$. Thus, the aim of the present study was to systematically review and provide a meta-analysis on the application of the verification phase for confirming whether the highest possible $VO_2$ has been attained during ramp or step-incremented CPETs in apparently healthy adults.

## Methods

### Protocol and registration

The systematic review was performed in accordance with the Preferred Reporting Items for Systematic Reviews and Meta-Analyses (PRISMA) guidelines. A completed PRISMA checklist is shown in S1 Checklist. The protocol for this study was recorded at http://www.crd.york.ac.uk/PROSPERO (CRD42019123540). The main questions addressed by the present study were: To what extent does the highest $VO_2$ attained in the CPET differ from that attained in the verification phase? Secondly, are the highest $VO_2$ values in the CPET and verification phase affected by the verification-phase characteristics (e.g. intensity, adoption of a criterion threshold, and aspects of the recovery period between the CPET and the verification phase), or even with respect to particular subgroups (e.g. sex, cardiorespiratory fitness levels, exercise test modality, and CPET protocol design) in apparently healthy adults?

### Search strategy

MEDLINE (accessed through PubMed), Web of Science, SPORTDiscus, and Cochrane (accessed through Wiley) were searched for peer-reviewed literature using a combination of medical subject heading (MeSH) descriptors, with a time frame that spanned the inception of each database until the search date (September 30$^{th}$, 2020). The search strategy was developed based on the PICO method [i.e. *P*articipants: apparently healthy humans; *I*nterventions: any intervention involving exercise; *C*omparisons: incremental CPET and an appended square-wave or multistage verification phase; and *O*utcome: $VO_{2max}$ confirmation]. The electronic search strategies for all databases are provided in S1 Text.

The terms were adapted for use with other bibliographic databases. Reference lists and citations of eligible articles were also hand searched for additional relevant studies. The search was performed in a standardized manner by two independent researchers (VABC and TP). Only English language studies were eligible for inclusion and only if they satisfied three *a priori* criteria: (1) involved apparently healthy participants who were $\geq$ 18 years of age; (2) determined $VO_{2max}$ using expired gas analysis indirect calorimetry; and (3) the CPET was carried out using bipedal cycle ergometer or bipedal treadmill running or walking. Studies were excluded if they involved: (1) participants who had taken dietary supplements or drugs that could affect body mass, metabolic profile, or exercise performance; or (2) the use of non-maximal test protocols.

## Study selection

Potential studies were screened for inclusion using three methods: (1) title only; (2) title and abstract; and (3) full-text review. Two investigators independently searched and selected articles, and coauthors subsequently confirmed articles to be included in the analysis. Disagreements were resolved by consensus. Agreement between investigators with respect to inclusion and/or exclusion of potential trials was ratified in 252 randomly selected abstracts by means of Cohen's kappa ($\kappa = 0.811$, $P < 0.05$). Fig 1 summarizes the screening and selection process.

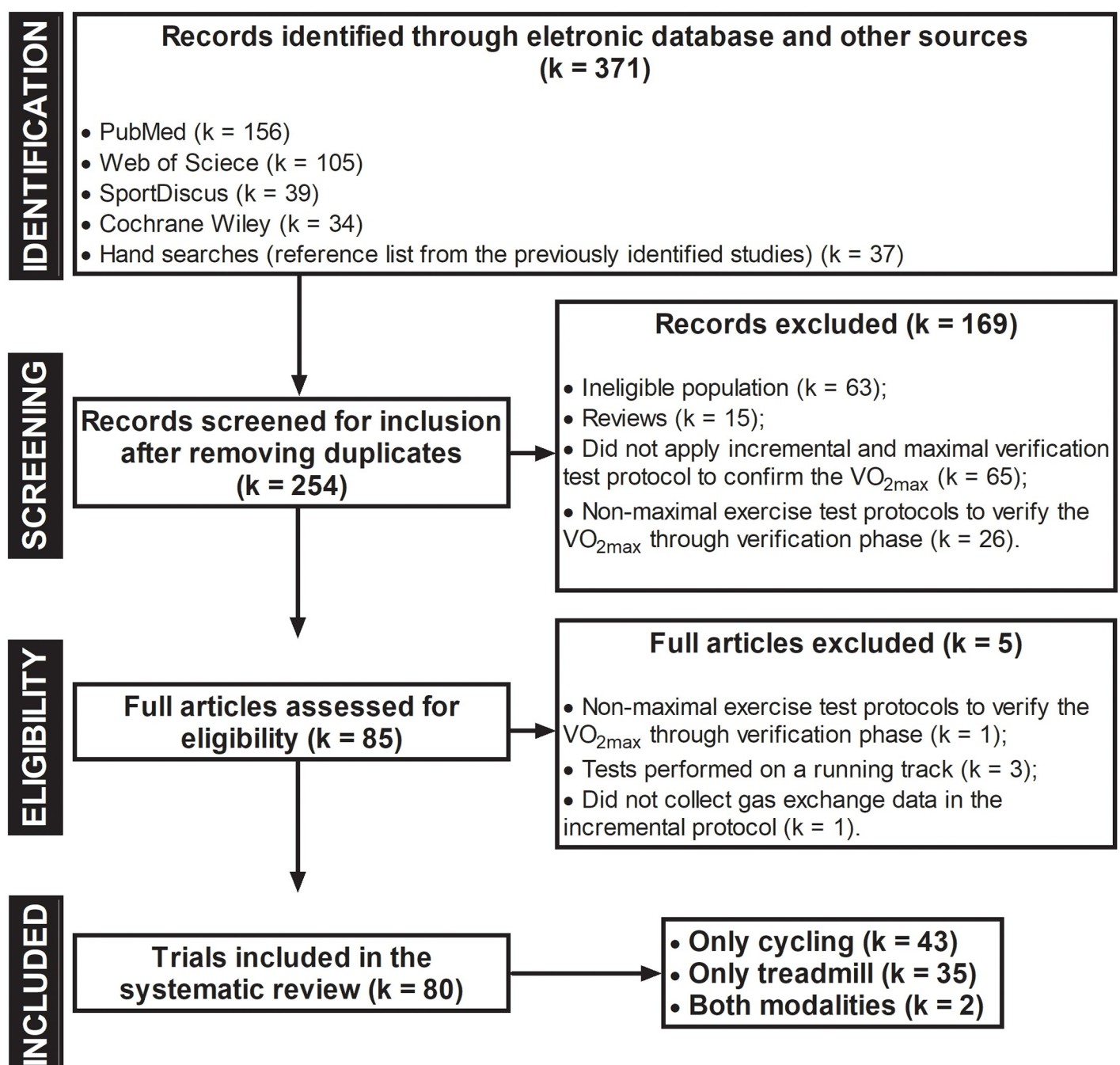

**Fig 1. Flowchart of the systematic review and meta-analysis according to the PRISMA guidelines.** $VO_{2max}$: maximal oxygen uptake.

## Data extraction and management

Two independent reviewers extracted data using a standardized form. The following data were summarized: (1) characteristics of study participants (total sample number, sex, age, body mass index [BMI], and cardiorespiratory fitness); (2) type of intervention (CPET and verification-phase duration, exercise modality, and exercise test protocol used); and (3) outcome measures (mean ± standard deviation [SD] for group $VO_{2max}$ and protocol duration during the CPET and verification phase). Disagreements were resolved by consensus. When the relevant quantitative data were not reported, authors of the original studies were contacted to request the data.

## Quality assessment

The risk of bias for all eligible studies was not assessed because it does not apply to the characteristics of the present review. For example, randomization sequence generation and treatment allocation concealment were not applied, since there were no comparison groups and each individual acted as their own control. It is also noteworthy to mention the absence of blinding in both participants undergoing testing and evaluators who applied the CPET and verification phases, because procedurally all exercise protocols were performed in a fixed order (i.e. CPET followed by the verification phase). Given that $VO_{2max}$ is the evaluation of an objective numerical variable, the blinding of the evaluator does not generate a different interpretation of the $VO_{2max}$ values obtained in a CPET and verification phase. Finally, the assessment of incomplete outcome data (sample loss) and selective reporting of outcomes also does not apply, because it is a cross-sectional study with a single outcome of interest.

## Statistical analysis

All meta-analyses were performed using Review Manager (RevMan) software version 5.3 (Copenhagen, The Nordic Cochrane Centre, The Cochrane Collaboration, 2014). Data are presented as the mean ± SD unless otherwise stated. The outcome was the mean difference (95% confidence interval [CI]) between the CPET and verification phase for the highest absolute $VO_2$ (L/min). Given that absolute $VO_2$ are continuous data, the weighted mean difference (WMD) method was used for combining study effect size estimates. With the WMD method, the pooled effect estimate represents a weighted mean of all included study group comparisons. The weighting assigned to each individual study group (i.e. the comparison of the CPET and verification phase results) in the analysis is inversely proportional to the variance of the absolute $VO_2$ (L/min). This method typically assigns more weight in the meta-analysis to studies with the highest precision (inverse variance) /larger sample sizes. The WMDs were calculated using random-effects models given the study group differences in CPET modalities and protocols, types of recovery, and verification phase protocols.

Heterogeneity of net study group changes in $VO_{2max}$ (L/min) was examined using the Q statistic. Cochran's Q statistic is computed by summing the squared deviations of each trial's estimate from the overall meta-analytic estimate and weighting each trial's contribution in the same manner as in the meta-analysis. *P*-values were obtained by comparing the statistic with a $\chi^2$ distribution with k-1 degrees of freedom (where k is the number of trials). A *P*-value of < 0.10 was adopted since the Q statistic tends to suffer from low differential power. The formal Q statistic was used in conjunction with the methods for assessing heterogeneity. The $I^2$ statistic measures the extent of inconsistency among the results of the primary study groups, interpreted approximately as the proportion of total variation in point estimates that is due to heterogeneity rather than sampling error. Effect sizes with a corresponding $I^2$ value of $\leq 50\%$

were considered to have low heterogeneity. The publication bias of the articles was assessed using a funnel plot.

Subgroup analyses were defined *a priori* to investigate the magnitude of differences between CPETs and verification phases due to variations in sex, cardiorespiratory fitness level, exercise modality, CPET protocol design, or how the verification phase was performed. Forest plots were constructed to display values at the 95% confidence level. Effect sizes were calculated by subtracting the highest mean values for $VO_2$ (L/min) observed in the CPET from the verification phase values, on the basis of grouping studies with selected verification-phase characteristics for intensity (i.e. sub *vs*. supra $WR_{peak}$) and type of recovery between the CPET and verification phase (i.e. active *vs*. passive). The studies were also classified according to whether a criterion threshold for $VO_{2max}$ was used for the verification phase (i.e. yes *vs*. no), whether the verification phase was performed in the same testing session as the CPET or on a different day, and the duration of the verification phase (i.e. $\leq 80$ s, 81–120 s, and $> 120$ s). Stratified analyses were also conducted according to particular subgroups such as sex (i.e. male and female), cardiorespiratory fitness level using the cut-off points proposed by Astorino et al. [53] (i.e. low: $< 40$ mL·kg$^{-1}$·min$^{-1}$; moderate: 40–50 mL·kg$^{-1}$·min$^{-1}$; high: $> 50$ mL·kg$^{-1}$·min$^{-1}$), exercise test modality (i.e. cycling and running), and CPET protocol design (i.e. discontinuous step-incremented, continuous step-incremented, and ramp protocols).

## Results

The literature search identified 371 potential articles, with 334 obtained from electronic database searches and 37 from the wider inspection of reference lists and electronic citations of these articles. Eighty studies published between 1980 and 2020 met the eligibility criteria and were included in the systematic review (see Fig 1).

## Participants

The total number of participants recruited across all included studies was 1,680 (1,077 men, 473 women, and the sex of 130 participants was not specified). Included studies had a median (interquartile range [IQR]) sample size of 13 [10] participants. Participants were aged between 19 and 68 yr, all apparently healthy, and with a physical activity status ranging from sedentary to highly-trained endurance athletes. Thirty-six studies included only men, two included only women, 41 included both men and women, and one study did not specify the sex of the participants (see Table 1). On average, participants had a BMI within the normal range (mean ± SD [range]: 24.4 ± 2.5 [19.4–32.0] kg/m$^2$) and a moderate level of cardiorespiratory fitness ($VO_{2max}$ mean ± SD [range]: 46.9 ± 12.1 [23.9–68.6] mL·kg$^{-1}$·min$^{-1}$).

## Characteristics of studies regarding the CPET and verification phase protocols to evaluate $VO_{2max}$

Table 2 summarizes the characteristics of the CPET and verification phase protocols of the 80 studies included in this systematic review. Forty-three studies (54%) performed the CPET on a cycle ergometer, 35 (44%) on a treadmill, and two studies (3%) used both modalities. Seventy-three studies (91%) used continuous step-incremented or ramp/pseudo-ramp CPET protocols. Three (4%) used only discontinuous step-incremented protocols. Two studies (3%) used both discontinuous and continuous step-incremented protocols and another two studies (3%) applied self-paced protocols. Thirty-three (41%) of the 80 studies included in the review used one or more $VO_2$ plateau or secondary $VO_{2max}$ criteria to confirm the attainment of $VO_{2max}$. Thirty studies used the $VO_2$ plateau, 21 used the heart rate plateau or a criterion based on age-

**Table 1. Sample characteristics for studies that incorporated a cardiopulmonary exercise test (CPET) (k = 80).**

| Study | Population | Sex | N | mean values | | |
|---|---|---|---|---|---|---|
| | | M/F | | Age | BMI | VO$_{2max}$ |
| | | | | Years | kg/m$^2$ | mL·kg$^{-1}$·min$^{-1}$ |
| Alexander and Mier [54] | Soccer players | M/F | 5/6 | 21.3 | 22.7 | 57.7 |
| Arad et al. [55] | Sedentary | M | 19 | 33.4 | 25.8 | 30.0 |
| | | F | 16 | 26.8 | 26.6 | 27.1 |
| Astorino and DeRevere [56] | Recreationally trained | M/F | 19/11 | 26 | NS | 47.2 |
| | | M/F | 41/38 | 23.3 | NS | 40.5 |
| Astorino and White [57] | Physically active | M | 13 | 23.5 | 24.3 | 43.8 |
| | | F | 17 | 22.9 | 22.0 | 40.7 |
| Astorino et al. [53] | Low CRF | M/F | 5/5 | 25.7 | 22.7 | 36.2 |
| | Moderate CRF | M/F | 5/5 | 26.3 | 24.1 | 46.4 |
| | High CRF | M/F | 9/1 | 26 | 23.7 | 57.9 |
| Astorino et al. [58] | Active adults (HIIT-Baseline) | M/F | 3/11 | 27 | 22 | 38.0 |
| | Active adults (HIIT—Week 3) | | | | | 40.4 |
| | Active adults (Control—Baseline) | M/F | 8/6 | 23 | 24 | 40.2 |
| | Active adults (Control—Week 3) | | | | | 40.5 |
| Astorino et al. [59] | Active adults | M/F | 14 | 27 | 22.5 | 38.0 |
| Astorino et al. [60] | Sedentary | M/F | 6/9 | 22.4 | 24.5 | 32.7 |
| | Sedentary | M/F | 1/8 | 21.8 | 22.9 | 42.1 |
| Beltrami et al. [61] | Runners or cross-country skiers | M/F | 23/3 | 29 | 23.5 | 61.3 |
| Beltz et al. [62] | Recreationally trained | M | 16 | 23.6 | 26.6 | 47.4 |
| Bisi et al. [63] | Healthy adults | M | 11 | 23.5 | 22.6 | 35.0 |
| Chidnok et al. [64] | Active adults | M | 7 | 20 | 24.8 | 57.7 |
| Clark et al. [65] | Adults of various fitness levels | M/F | 3/12 | 22 | 22.0 | NS |
| Colakoglu et al. [66] | Athletes | M | 9 | 24.2 | 23.0 | 59.7 |
| Colakoglu et al. [67] | Well-trained athletes | M | 9 | 23.6 | 23.1 | 60.2 |
| Colakoglu et al. [68] | Athletes | M | 9 | 23.6 | 23.1 | 60.2 |
| Dalleck et al. [69] | Healthy adults | M/F | 9/9 | 59.7 | 27.8 | 27.7 |
| Day et al. [35] | Healthy adults | M | 38 | 19–61 | NS | NS |
| Del Giudice et al. [70] | Healthy adults | M | 14 | 21.5 | 22.8 | 60.2 |
| Dexheimer et al. [71] | Active adults | M | 12 | 29 | 31.4 | 50.6 |
| | | F | 5 | 25.6 | 24.4 | 43.7 |
| Dicks et al. [72] | Firefighters | M | 30 | 34.5 | 28.7 | 41.0 |
| Dogra et al. [73] | Older adults (trained) | F | 7 | 62.7 | 23.4 | 37.8 |
| | Older adults (untrained) | F | 10 | 68.8 | 26.1 | 24.1 |
| Ducrocq et al. [74] | Recreationally trained | M/F | 9/4 | 21.2 | 22.5 | 56.0 |
| Elliott et al. [75] | Cyclists | M | 8 | 40.5 | 25.2 | 53.7 |
| Faulkner et al. [76] | Recreationally trained | M | 13 | 25.5 | 24.5 | 63.9 |
| Foster et al. [77] | Physically active non-athletes (cycling) | M | 16 | 31.5 | 24.0 | 51.7 |
| | | F | 4 | 28 | 21.6 | |
| | Competitive runners (treadmill running) | M | 12 | 21.6 | 22.9 | 56.3 |
| | | F | 8 | 21 | 20.5 | |
| Freeberg et al. [78] | Healthy adults | M/F | 17/13 | 21.7 | 23.7 | 49.9 |
| Goodall et al. [79] | Cyclists | M | 9 | 28.1 | 23.1 | 61.1 |
| Hanson et al. [80] | Recreationally trained | M/F | 8/5 | 24 | 24.7 | 56.2 |
| Hawkins et al. [81] | Distance runners | M/F | 36/16 | NS | NS | 63.3 |
| Hogg et al. [82] | Highly trained | M | 14 | 28 | 23.2 | 68.6 |

(*Continued*)

**Table 1.** (Continued)

| Study | Population | Sex M/F | N | Age Years | BMI kg/m² | VO₂max mL·kg⁻¹·min⁻¹ |
|---|---|---|---|---|---|---|
| | | | | | | |

| Study | Population | Sex M/F | N | Age Years | BMI kg/m² | VO2max mL·kg⁻¹·min⁻¹ |
|---|---|---|---|---|---|---|
| Iannetta et al. [25] | Recreationally trained | M/F | 6/5 | 28 | 21.8 | 52.6 |
| James et al. [83] | Squash players | M/F | 6/2 | 20.3 | 22.1 | 48.8 |
| Jamnick et al. [84] | Trained cyclists | M | 17 | 36.2 | 24.1 | 62.1 |
| Jamnick et al. [85] | Active adults | M | 31 | 29 | 25.2 | 48.6 |
| | | F | 26 | 27 | 23.4 | 39.8 |
| Johnson et al. [86] | Recreationally trained runners and cyclists | M/F | 6/5 | 22 | 24.1 | 46.9 |
| Keiller and Gordon [87] | Recreationally trained | M/F | 9/2 | 22.4 | 24.4 | 51.6 |
| Kirkeberg et al. [88] | Recreational-trained men | M | 12 | 29 | 27.5 | 49.2 |
| Knaier et al. [89] | Athletes | M | 10 | 27.5 | 23.1 | 61.1 |
| | | F | 7 | 28.4 | 22.5 | 54.3 |
| Knaier et al. [90] | High cardiorespiratory fitness | M | 8 | 27.4 | 22.8 | 62.8 |
| | | F | 5 | 27.6 | 22.7 | 55.2 |
| Kramer et al. [91] | Soccer players | M | 15 | 23.1 | 23.0 | 50.5 |
| Mann et al. [92] | Runners | M | 20 | 30 | 24.2 | 60.2 |
| | | F | 12 | 28 | 21.7 | 51.9 |
| Mann et al. [93] | Runners | M | 8 | 36 | 24.1 | 57.9 |
| | | F | 2 | 32 | 24.9 | 49.9 |
| Mauger et al. [94] | Well-trained runners | M | 14 | 22.7 | 23.4 | 64.4 |
| McGawley [95] | Recreational runners | M/F | 5/5 | 32 | NS | 59.8 |
| McKay et al. [96] | Healthy adults | M | 12 | 25 | NS | 44.5 |
| Midgley et al. [39] | Runners | M | 10 | 39.3 | 23.6 | 53.6 |
| | Cyclists | M | 10 | 36.0 | 23.2 | 57.7 |
| Midgley et al. [97] | Middle- and long-distance runners | M | 16 | 38.7 | 23.0 | 57.1 |
| Midgley et al. [98] | Distance runners | M | 9 | 38.2 | 24.6 | 55.0 |
| Mier et al. [99] | College athletes | M/F | 8/27 | 20 | 23.5 | 55.5 |
| Murias et al. [100] | Younger adults | M | 30 | 25 | 24.9 | 49.4 |
| | Older adults | M | 31 | 68 | 25.8 | 33.0 |
| Murias et al. [101] | Older adults | F | 6 | 69 | 27.0 | 23.9 |
| | Younger adults | F | 8 | 25 | 23.8 | 41.2 |
| Murias et al. [102] | Older adults | M | 8 | 68 | 26.0 | 28.3 |
| | Younger adults | M | 8 | 23 | 25.2 | 48 |
| Nalcakan [103] | Healthy adults | M | 15 | 21.7 | 25.0 | 40.3 |
| Niemela et al. [104] | Healthy adults | M | 16 | 25–35 | 23.3 | 42.5 |
| Niemeyer et al. [105] | Physically active | M | 24 | 26.2 | 24.2 | 49.8 |
| Niemeyer et al. [106] | Recreationally trained | M | 46 | 25.6 | 24.0 | 50.8 |
| Nolan et al. [107] | Active adults | M/F | 6/6 | 23 | 22.7 | 57.5 |
| Poole et al. [37] | Healthy adults | M | 8 | 27 | NS | 50.8 |
| Possamai et al. [108] | Recreationally trained cyclists | M | 19 | 23 | 25.3 | 48.0 |
| Riboli et al. [109] | Soccer players | M | 16 | 22.5 | 22.4 | 59.2 |
| Rossiter et al. [38] | Healthy adults | M | 7 | 26 | 25.1 | 51.5 |
| Sabino-Carvalho et al. [110] | Runners | M | 14 | 22.3 | 21.2 | 67.0 |
| | | F | 4 | 24 | 20.4 | 60.1 |
| Scharhag-Rosenberger et al. [111] | Healthy adults | M/F | 20/20 | 24 | 23.0 | 50.0 |
| Scheadler and Devor [112] | Experienced runners | NS | 13 | 25 | 22.5 | 64.9 |
| Sedgeman et al. [113] | Recreationally trained | M/F | 6/7 | 29 | 23.9 | 50.1 |

(*Continued*)

**Table 1.** (Continued)

| Study | Population | Sex | N | mean values | | |
|---|---|---|---|---|---|---|
| | | M/F | | Age | BMI | VO$_{2max}$ |
| | | | | Years | kg/m$^2$ | mL·kg$^{-1}$·min$^{-1}$ |
| Stachenfeld et al. [114] | Healthy adults | M/F | 33/18 | 30.6 | NS | 49.2 |
| Straub et al. [115] | Trained cyclists | M | 12 | 33 | 24.8 | 56.5 |
| | | F | 4 | 38 | 22.1 | |
| Strom et al. [116] | Healthy adults | M/F | 21/29 | 30.3 | 24.0 | 47.3 |
| Taylor et al. [117] | Runners and triathlon athletes | M | 11 | 28.5 | 22.6 | 63.7 |
| | | F | 8 | 26.3 | 21.8 | 52.3 |
| Tucker et al. [118] | Nonexercise-trained youth | M | 17 | 27 | 25.6 | 41.6 |
| Vogiatzis et al. [119] | Cyclists | M | 11 | 38 | 22.1 | 62.0 |
| Weatherwax et al. [120] | Sedentary adults | M | 5 | 53.6 | 32.0 | 32.3 |
| | | F | 11 | 52.2 | 29.4 | 24.8 |
| Weatherwax et al. [15] | Sedentary adults (standardized—baseline) | M/F | 4/16 | 51.2 | 29.6 | 24.3 |
| | Sedentary adults (standardized—week 4) | M/F | 4/16 | 51.2 | 29.7 | 25.0 |
| | Sedentary adults (standardized—week 8) | M/F | 4/16 | 51.2 | 29.6 | 26.3 |
| | Sedentary adults (standardized—week 12) | M/F | 4/16 | 51.2 | 29.6 | 26.3 |
| | Sedentary adults (individualized—baseline) | M/F | 5/14 | 44.9 | 27.2 | 29.5 |
| | Sedentary adults (individualized—week 4) | M/F | 5/14 | 44.9 | 27.2 | 31.1 |
| | Sedentary adults (individualized—week 8) | M/F | 5/14 | 44.9 | 27.1 | 31.3 |
| | Sedentary adults (individualized—week 12) | M/F | 5/14 | 44.9 | 27.0 | 32.8 |
| Weatherwax et al. [121] | Sedentary adults (control—baseline) | M/F | 2/6 | 45.6 | 25.5 | 28.4 |
| | Sedentary adults (control—week 12) | | | | 25.5 | 27.7 |
| | Sedentary adults (standardized—baseline) | M/F | 4/16 | 51.2 | 29.6 | 24.3 |
| | Sedentary adults (standardized—week 12) | | | | 29.6 | 26.0 |
| | Sedentary adults (individualized—baseline) | M/F | 5/14 | 44.9 | 27.1 | 29.5 |
| | Sedentary adults (individualized—week 12) | | | | 26.8 | 32.8 |
| Weatherwax et al. [122] | Elite endurance-trained | M | 18 | 21.9 | 19.8 | 62.8 |
| | | F | 6 | 20.2 | 19.4 | 51.7 |
| Wilhelm et al. [123] | Healthy adults | M | 9 | 25 | 25.1 | 41.0 |
| Williams et al. [124] | Healthy adults | M | 8 | 27 | NS | 43.0 |
| | | M | 5 | 23 | NS | 48.0 |
| Wingo et al. [125] | Healthy adults | M | 9 | 25 | 22.4 | 61.2 |
| Yeh et al. [126] | Healthy adults | M/F | 14/1 | 23.3 | 21.9 | 48.9 |

**Abbreviations:** *BMI* = body mass index; *CRF* = cardiorespiratory fitness level; *F* = female; *HIIT* = high-intensity interval training; *M* = male; *NS* = not stated; *VO$_{2max}$* = maximal oxygen uptake. *Note*: Whenever possible, authors were contacted to provide unpublished data.

predicted maximal heart rate, 18 used the maximal RER attained in the CPET (RER$_{max}$), and 8 used the post-CPET blood lactate concentration.

In terms of processing respiratory VO$_2$ data at volitional exhaustion, the most common approach was based on time averages. Thirty-eight studies (48%) reported stationary time averages of 5- to 30-s, whereas 29 (36%) used VO$_2$ data points at fixed intervals of 15- to 30-s, two studies (3%) used 15-breath averages, two studies (3%) used 10-25-s moving averages, one (1%) used 10-s epochs, two (3%) used 20-s rolling averages, one (1%) used 30-s rolling means, and one study (1%) used Douglas bag collections. Four studies (5%) did not detail which VO$_2$ data processing method was applied.

**Table 2. Characteristics of the cardiopulmonary exercise test (CPET) and verification phase protocols used in the reviewed studies (k = 80).**

| Study | $VO_2$ data sampling method | Traditional $VO_{2max}$ criteria adopted | Exercise Modality | CPET Protocol | Recovery Phase Protocol | Verification Phase (VP) Protocol | Verification Criteria Threshold |
|---|---|---|---|---|---|---|---|
| Alexander and Mier [54] | 30-s time average | $VO_2$ plateau of 2.1 mL·kg$^{-1}$·min$^{-1}$ | TR | CSI | 10-min walking | 1st min: ↑ WR until matching the final stage of CPET; then ↑ slope to 2.5% and encouraged to running for 2-min | NS |
| | | | | DisCSI | | | |
| Arad et al. [55] | 20-s rolling mean | $VO_2$ plateau (linear portion of the $VO_2$-WR relationship); $RER_{max} \geq 1.10$; $\geq$ 95% APMHR | CYC | Ramp | 10-min active and 2–3 min passive | 100% $WR_{peak}$ | NS |
| Astorino and DeRevere [56] | 2×15-s | NS | CYC | Ramp | 8-min active | 105% $WR_{peak}$ | CPET *vs.* VP: $VO_{2max}$ difference ≤ 3.0% and 3.3% and $HR_{max}$ ≤ 4 bpm |
| | | | | | 10-min active | 110% $WR_{peak}$ | |
| Astorino and White [57] | 15-s time average | NS | CYC | CSI | 10-min active | one stage > CPET-Stage$_{final}$ | CPET *vs.* VP: $VO_{2max}$ difference ≤ 3% and $HR_{max}$ ≤ 4 bpm |
| Astorino et al. [53] | 2×15-s | NS | CYC | Ramp | 8-min active | 2-min at 40–45% $WR_{peak}$ and then 105% $WR_{peak}$ | CPET *vs.* VP: $VO_{2max}$ difference < 2 mL·kg$^{-1}$·min$^{-1}$ |
| Astorino et al. [58] | 30-s time average | NS | CYC | Ramp | 10-min active | 105% $WR_{peak}$ | NS |
| Astorino et al. [59] | 30-s time average | NS | CYC | CSI | 10-min active | 105% $WR_{peak}$ | $VO_{2max}$ identified as the average of CPET and VP values |
| Astorino et al. [60] | 2×15-s | NS | CYC | Ramp | ≥ 24h | 105%$WR_{peak}$ reached in the CPET | NS |
| | 30-s time average | | | | 1–1.5h | 115%$WR_{peak}$ reached in the CPET | |
| Beltrami et al. [61] | 30-s intervals | $VO_2$ plateau (difference between modelled and actual value >50% of the regression slope for the linear portion of the $VO_2$-WR relationship—an average of 1.7 mL·kg$^{-1}$·min$^{-1}$) | TR | CSI (control) | 15-min active or passive (self-choose: walk, jog or rest) | 1st min at 10 km/h (5% slope) and then ↑ 1 km/h > CPET-Speed$_{peak}$ | CPET *vs.* VP: $VO_{2max}$ difference ≤ 123 ± 18 mL/min (or 1.7 mL·kg$^{-1}$·min$^{-1}$) |
| | | | | CSI (reverse) | | | |
| Beltz et al. [62] | 2×15-s | NS | TR | SPV | 20-min passive | 2-min at 30% CPET-$WR_{peak}$, 1-min at 40–45% CPET-$WR_{peak}$ and then until exhaustion at 105% CPET-$WR_{peak}$ | CPET *vs.* VP: $VO_{2max}$ difference ≤ 3% |
| | | | | Ramp | | | |
| Bisi et al. [63] | 25-s moving-average | $VO_2$ plateau (increase < than 3% or 2.1 mL·kg$^{-1}$·min$^{-1}$ between 2 steps of increment); $RER_{max} \geq$ 1.08 or 1.15; $HR_{max}$ within 10 bpm of APMHR | CYC | CSI | 6-min active | at least 3 min of cycling at 105% of the $WR_{peak}$ | CPET *vs.* VP: $VO_{2max}$ difference ≤ 3% |
| Chidnok et al. [64] | 30-s rolling-mean | NS | CYC | Ramp | Different day | See the formula for a proper reporting 3-min of 'all-out' cycling | NS |
| Clark et al. [65] | 15-s time average | NS | CYC | CSI | 3-min active | $WR_{peak}$ minus 2 stages | NS |
| Colakoglu et al. [66] | 30-s average | $VO_2$ plateau of 150 mL/min; $RER_{max} \geq$ 1.10; ≥ 90% APMHR; | CYC | Ramp | Different day | 100% $WR_{peak}$ | NS |
| Colakoglu et al. [67] | 30-s average | $VO_2$ plateau of 150 mL/min; $RER_{max} \geq$ 1.10; $HR_{max}$ within 10 bpm of APMHR; RPE ≥? | CYC | CSI | Different day | 100% $WR_{peak}$ | $VO_2$ plateau of 150 mL/min; $RER_{max} \geq$ 1.10; $HR_{max}$ within 10 bpm of APMHR; RPE ≥? |
| Colakoglu et al. [68] | 30-s average | $VO_2$ plateau of 150 mL/min; $RER_{max} \geq$ 1.10; ≥ 90% APMHR; RPE ≥ 19–20 | CYC | CSI | Different day | 100%, 105%, and 110% $WR_{peak}$ to attain the highest $VO_{2peak}$ value | $VO_2$ plateau of 150 mL/min; $RER_{max} \geq$ 1.10; ≥ 90% APMHR; RPE ≥ 19–20 |
| Dalleck et al. [69] | 2×15-s | NS | CYC | CSI | 60-min passive | 2-min at 50 Watts; then increased 105% $WR_{peak}$ | CPET *vs.* VP: $VO_{2max}$ difference ≤ 3% and $HR_{max}$ ≤ 4 bpm |
| Day et al. [35] | 30-s time average | NS | CYC | CSI | Different day | 90% $WR_{peak}$ reached in the CPET | NS |
| Del Giudice et al. [70] | 30-s time average | NS | TR | CSI | 10-min passive | 0.8 km/h > CPET-Speed$_{peak}$ | NS |
| Dexheimer et al. [71] | 2×15-s | NS | TR | Pseudo-ramp protocol | 5-10-min active | 105% $WR_{peak}$ | CPET *vs.* VP: $VO_{2max}$ difference ≤ 3% |
| Dicks et al. [72] | 15-s time average | NS | TR | Pseudo-ramp protocol | 3-min active | $WR_{peak}$ minus 2 stages | NS |

(*Continued*)

**Table 2.** (*Continued*)

| Study | VO$_2$ data sampling method | Traditional VO$_{2max}$ criteria adopted | Exercise Modality | CPET Protocol | Recovery Phase Protocol | Verification Phase (VP) Protocol | Verification Criteria Threshold |
|---|---|---|---|---|---|---|---|
| Dogra et al. [73] | every 20 ms | NS | CYC | Ramp | Different day | 85%WR$_{peak}$ reached in the CPET | NS |
| Ducrocq et al. [74] | breath-by-breath | NS | TR | CSI | 5-min passive | 105% WR$_{peak}$ | NS |
| Elliott et al. [75] | 10-s epochs | NS | CYC | CSI | 60-min | 110%WR$_{peak}$ reached in the CPET | NS |
| Faulkner et al. [76] | 20-s time average | VO$_2$ plateau of 2 mL·kg$^{-1}$·min$^{-1}$; RER$_{max}$ ≥ 1.10; RPE ≥ 17; HR$_{max}$ within 10 bpm of APMHR; La$_{max}$ ≥ 8 mmol | TR | CSI | 15-min passive | ↑ speed over a 30-second period up to a 1 km/h > CPET-Speed$_{peak}$ | NS |
| Foster et al. [77] | 30-s time average | rate of increase in VO$_2$ during the last min < 50% when compared to the mid portion of the test | CYC | CSI | 1-min active | 25 Watts > CPET-WR$_{peak}$ | NS |
| | | | TR | CSI | 3-min active | 1.6 km/h > CPET-Speed$_{peak}$ or 0.8 km/h if in the non-athlete group | |
| Freeberg et al. [78] | 2×15-s | NS | TR | Incline-based protocol | 10-min active | 110% WR$_{peak}$ | CPET *vs.* VP: VO$_{2max}$ difference ≤ 3% |
| Goodall et al. [79] | 30-s mean | NS | CYC | CSI | 5-min passive | as described by [38]; however, the intensity was not stated (i.e. 95 or 105%WR$_{peak}$ reached in the CPET) | NS |
| Hanson et al. [80] | 15-breath moving average | VO$_2$ plateau of 2 mL·kg$^{-1}$·min$^{-1}$; RER$_{max}$ ≥ 1.10 | TR | CSI | 10-min active | one stage > CPET-WR$_{peak}$ | CPET *vs.* VP: VO$_{2max}$ difference ≤ 50 mL/min |
| Hawkins et al. [81] | 40-s Douglas bag collection | NS | TR | CSI | Different day | 130% WR$_{peak}$ | NS |
| Hogg et al. [82] | 30-s time average | VO$_2$ plateau (difference between modelled and actual value > 50% of the regression slope for the linear portion of the VO$_2$-WR relationship); RER$_{max}$ ≥ 1.10; RPE ≥ 17; HR$_{max}$ within 10 bpm of APMHR | TR | CSI | 10-min active (walking around the laboratory and stretching) | ↑ speed over a 30-second period up to a speed stage > CPET-Stage$_{final}$ | CPET *vs.* VP: VO$_{2max}$ difference ≤ 3% |
| | | | | Incline-based SPV | | speed halfway between speed$_{peak}$ from the SPV$_{incline}$ *vs.* predicted verification-stage speed of the CSI protocol | |
| | | | | Speed-based SPV | | speed halfway between speed$_{peak}$ from the SPV$_{speed}$ *vs.* predicted stage speed of the CSI protocol | |
| Ianetta et al. [25] | 20-s rolling mean | VO$_2$ plateau (linear portion of the VO$_2$-WR relationship) | CYC | Ramp | 10-min | 110% WR$_{peak}$ | CPET *vs.* VP: VO$_{2max}$ difference ≤ 0.1 L/min |
| James et al. [83] | 10-s average | VO$_2$ plateau of 2 mL·kg$^{-1}$·min$^{-1}$ | TR | CSI | 5-min active | ↑ 1% > CPET-Slope | VO$_2$ plateau of 2 mL·kg$^{-1}$·min$^{-1}$ |
| Jamnick et al. [84] | 20-s average | NS | CYC | CSI$_1$ (1-min stage length) | 5-min passive | 90% WR$_{peak}$—CSI$_1$ | NS |
| | | | | CSI$_3$ (3-min stage length) | | | |
| | | | | CSI$_5$ (5-min stage length) | | | |
| | | | | CSI$_7$ (7-min stage length) | | | |
| | | | | CSI$_{10}$ (10-min stage length) | | | |

(*Continued*)

**Table 2.** (Continued)

| Study | VO$_2$ data sampling method | Traditional VO$_{2max}$ criteria adopted | Exercise Modality | CPET Protocol | Recovery Phase Protocol | Verification Phase (VP) Protocol | Verification Criteria Threshold |
|---|---|---|---|---|---|---|---|
| Jamnick et al. [85] | 15-s time average | NS | CYC | CSI | 3-min active | mean WR$_{peak}$ minus 2 stages | CPET vs. VP: VO$_{2max}$ difference $\leq$ 1.5 mL·kg$^{-1}$·min$^{-1}$ (or 3% CV) |
| Johnson et al. [86] | 15-s intervals | NS | CYC | CSI | 3-min active (50% WR$_{peak}$) | WR$_{peak}$ minus 2 stages | CPET vs. VP: VO$_{2max}$ difference $\leq$ 3% |
| Keiller and Gordon [87] | 30-s intervals | VO$_2$ plateau (increase < than 50 or 100 mL/min) and HR plateau (increase < than 2 or 4 bpm) over the final two consecutive 30 s sampling periods | TR | CSI (Trials 1 and 2) | 6-min passive | 10 (female) and 9 (male) km/h and the ↑ 1% > CPET-Slope | CPET vs. VP: HR$_{max}$ difference $\leq$ 2 or $\leq$ 4 bpm |
| Kirkeberg et al. [88] | 30-s time average | NS | TR | CSI (short-term) | 3-min active | CPET-Speed$_{end}$ minus 2 stages, where stages were derived using specific equation | NS |
| | | | | CSI (middle-term) | | | |
| | | | | CSI (large-term) | | | |
| Knaier et al. [89] | 30-s time average | RER$_{max}$ $\geq$ 1.10; $\geq$ 95% APMHR; RPE $\geq$ 19; La$_{max}$ $\geq$ 8 mmol | CYC | CSI | 10-min active | 2 min at 50% WR$_{peak}$, 1 min at 70% WR$_{peak}$, and then 1 stage > CPET-WR$_{peak}$ | CPET vs. VP: VO$_{2max}$ difference $\leq$ 3% |
| Knaier et al. [90] | 30-s time average | RER$_{max}$ $\geq$1.05, 1.10 and 1.15; 90, 95 and 100% APMHR; RPE $\geq$ 19 and = 20; La$_{max}$ $\geq$ 8 and 10 mmol | CYC | CSI | 10-min active | 2 min at 50% WR$_{peak}$, 1 min at 70% WR$_{peak}$, and then 1 stage > CPET-WR$_{peak}$ | CPET vs. VP: VO$_{2max}$ difference $\leq$ 3% |
| Kramer et al. [91] | 30-s intervals | NS | TR | CSI | 3-min active | 2 stages < CPET-WR$_{peak}$ | CPET vs. VP: VO$_{2max}$ difference $\leq$ 3% |
| Mann et al. [92] | 15-s | NS | TR | CSI | 8-10-min | 0.5 km/h > CPET-Speed$_{peak}$ | NS |
| Mann et al. [93] | 15-s | NS | TR | CSI | 8-10-min | 0.5 km/h > CPET-Speed$_{peak}$ | NS |
| Mauger et al. [94] | 5-s time average | VO$_2$ plateau (increase < than 1.8 mL·kg$^{-1}$·min$^{-1}$ between 2 steps of increment); RER$_{max}$ $\geq$ 1.10; HR$_{max}$ within 10 bpm of APMHR; RPE $\geq$ 17; La$_{max}$ $\geq$ 8 mmol | TR | CSI | 10-min active | one stage > the last completed stage of the CPET | CPET vs. VP: VO$_{2max}$ difference $\leq$ 1.8 mL·kg$^{-1}$·min$^{-1}$ |
| McGawley [95] | 30-s time average | VO$_2$ plateau (increase < than 3% or 2 mL·kg$^{-1}$·min$^{-1}$ between 2 steps of increment); RER$_{max}$ $\geq$ 1.15; HR$_{max}$ within 10 bpm of APMHR; La$_{max}$ $\geq$ 8 mmol | TR | CSI | 9-min passive | 105% at CPET-WR$_{peak}$ (Trials 1 to 5) | CPET vs. VP: VO$_{2max}$ difference $\leq$ 3% |
| McKay et al. [96] | 15-s time average | NS | CYC | Ramp | 5-min active | 105%WR$_{peak}$ reached in the CPET | NS |
| Midgley et al. [39] | 30-s time average | VO$_2$ plateau (difference between modelled and actual value > 50% of the regression slope for the linear portion of the VO$_2$-WR relationship) | CYC / TR | CSI | 10-min passive | 2 min at 50% WR$_{peak}$, 1 min at 70% WR$_{peak}$, and then 1 stage > CPET-WR$_{peak}$, 2 min at 50% WR$_{peak}$, 1 min at 70% WR$_{peak}$, and then 1 stage > CPET-WR$_{peak}$ | CPET vs. VP: modelled and verification VO$_2$ difference > 50% of the regression slope of the individual VO$_2$-WR relationship; HR$_{max}$ $\leq$ 4 bpm |
| Midgley et al. [97] | 30-s time average | absolute plateau in VO$_2$; RER$_{max}$ $\geq$ 1.10; HR$_{max}$ within 10 bpm of APMHR | TR | CSI | 10-min active | 0.5 km/h > CPET-Speed$_{peak}$ | CPET vs. VP: VO$_{2max}$ difference $\leq$ 2% and HR$_{max}$ $\leq$ 2 bpm |
| Midgley et al. [98] | 15 and 30-s time average | NS | TR | CSI 1-min stages | 5-min passive | one stage > CPET | NS |
| | | | | DisCSI 2-min stages | | | |
| | | | | DisCSI 3-min stages | | | |

(Continued)

**Table 2.** (*Continued*)

| Study | VO$_2$ data sampling method | Traditional VO$_{2max}$ criteria adopted | Exercise Modality | CPET Protocol | Recovery Phase Protocol | Verification Phase (VP) Protocol | Verification Criteria Threshold |
|---|---|---|---|---|---|---|---|
| Mier et al. [99] | 30-s | VO$_2$ plateau (2 mL·kg$^{-1}$·min$^{-1}$ and ≤ SD of the expected increase); RER$_{max}$ ≥ 1.05, 1.10 and 1.15; ≥ 85% APMHR and HR$_{max}$ within 10 bpm of APMHR | TR | CSI | 10-min active (walking at slow pace) | intensity gradually increased over 2-min until match CPET-WR$_{peak}$; after 1 min, the slope was increased 2.5% to running for 2-min | CPET *vs.* VP: VO$_{2max}$ difference ≤ 2.2 mL·kg$^{-1}$·min$^{-1}$ |
| Murias et al. [100] | 20-s average time | NS | CYC | Ramp | 5-min active | 85% WR$_{peak}$ | CPET *vs.* VP: VO$_{2max}$ difference ≤ 2.0 mL·kg$^{-1}$·min$^{-1}$ |
| | | | | | | 105% WR$_{peak}$ | |
| Murias et al. [101] | 20-s | NS | CYC | Ramp | 5-min active | 85%WR$_{peak}$ reached in the CPET | NS |
| Murias et al. [102] | 20-s | NS | CYC | Ramp | 5-min active | 85%WR$_{peak}$ reached in the CPET | NS |
| Nalcakan [103] | 30-s | VO$_2$ plateau; RER$_{max}$ ≥ 1.20; ≥ 90% APMHR | CYC | CSI | Different day | 100% WR$_{peak}$ | NS |
| Niemela et al. [104] | every min | VO$_2$ plateau (≤60 mL/min for men and ≤50 mL/min for women); adequacy of a subjective criterion for establishing the end point; RER$_{max}$ ≥ 1.15; HR$_{max}$ within 10 bpm of APMHR | CYC | CSI I / CSI II | Different day | 1 or 2 sub peak WRs, then 100% of the highest VO$_{2max}$ reached from two CPET | ≤5% difference between the ramp test and VP |
| Niemeyer et al. [105] | 30-s time average | < half of expected increase in VO$_2$ (i.e. <4.5 mL·kg$^{-1}$·min$^{-1}$) | CYC | Ramp | 10-min active | 90% WR$_{peak}$ | CPET *vs.* VP: VO$_{2max}$ difference ≤ 5% |
| Niemeyer et al. [106] | 30-s time average | VO$_2$ plateau (difference between modelled and actual value > 50% of the regression slope for the linear portion of the VO$_2$-WR relationship) | CYC | Ramp | Different day | 90% WR$_{peak}$ | CPET *vs.* VP: VO$_{2max}$ difference ≤ 5% |
| Nolan et al. [107] | 2×15-s | NS | TR | CSI | 20-min passive | 105% WR$_{peak}$ | CPET *vs.* VP: VO$_{2max}$ difference ≤ 3% |
| | | | | | | 115% WR$_{peak}$ | |
| | | | | | 60-min passive | 105% WR$_{peak}$ | |
| | | | | | | 115% WR$_{peak}$ | |
| Poole et al. [37] | 20 s | VO$_2$ plateau of regarding the mL/min; RER$_{max}$ ≥ 1.10, 1.15; HR$_{max}$ within 10 bpm of APMHR; La$_{max}$ ≥ 8 mmol | CYC | Ramp | Different day | 105%WR$_{peak}$ reached in the CPET | NS |
| Possamai et al. [108] | 30-s intervals | plateau in VO$_2$ and HR (i.e. ≤ 50 mL/min or ≤ 2 bpm) over the final two consecutive 30 s sampling periods; HR$_{max}$ within 10 bpm of APMHR | CYC | CSI | 15-min passive | 5-min warm-up at the first stage of the CPET; 3-min of passive recovery; 2-min at 20 Watts; then increased 100% WR$_{peak}$ | CPET *vs.* VP: VO$_{2max}$ difference ≤ 3% |
| Riboli et al. [109] | 30-s intervals | VO$_2$ plateau of 2.1 mL·kg$^{-1}$·min$^{-1}$ | TR | CSI with 1 min stages / CSI with 2 min stages / DisCSI | 5-min passive | if the CPET did not show a VO$_2$ plateau, a verification bout was performed as described by [38]; however, the intensity was not stated (i.e. 95 or 105%WR$_{peak}$ reached in the CPET) | NS |
| Rossiter et al. [38] | 15-s average | VO$_2$ plateau (linear least squares fitting technique) | CYC | Ramp | 5-min active | 105%WR$_{peak}$ reached in the CPET | NS |
| | | | | | | 95%WR$_{peak}$ reached in the CPET | |
| Sabino-Carvalho et al. [110] | 20-s average | NS | TR | DisCSI | 3-min passive (standing on treadmill) and 7-min active (walking at 5 km/h) | 2-min at 60% WR$_{peak}$ and then ↑ 0.5 km/h > CPET-Speed$_{peak}$ | CPET *vs.* VP: VO$_{2max}$ difference ≤ 2% |
| Scharhag-Rosenberger et al. [111] | 3×10-s average | VO$_2$ plateau (increase < than one-third of the oxygen requirement of a stage change ~ 150 mL/min); RER$_{max}$ ≥ 1.10; ± 10 bpm APMHR; La$_{max}$ > 8 mmol | TR | DisCSI | 10-min passive (VerifDay1) / Different day (VerifDay2) | 1 min at 60% CPET-Speed$_{peak}$ and then continued at 110% (or 115% if necessary, a second VF bout in VerifDay1) CPET-Speed$_{peak}$ | CPET *vs.* VP: VO$_{2max}$ difference ≤ 5.5% |

(*Continued*)

**Table 2.** (Continued)

| Study | VO$_2$ data sampling method | Traditional VO$_{2max}$ criteria adopted | Exercise Modality | CPET Protocol | Recovery Phase Protocol | Verification Phase (VP) Protocol | Verification Criteria Threshold |
|---|---|---|---|---|---|---|---|
| Scheadler and Devor [112] | 30-s | NS | TR | CSI | Different day | 8% slope/ individualized speed for a WR greater than CPET (mean estimated 10.2% WR$_{peak}$) | CPET *vs.* VP: VO$_{2max}$ difference ≤ 50 mL/min |
| Sedgeman et al. [113] | 15-s time average | VO$_2$ plateau of 2.1 mL·kg$^{-1}$·min$^{-1}$ during the last two 15-s average samples | CYC | CSI | 3-min active | WR$_{peak}$ minus 2-stages | CPET *vs.* VP: VO$_{2max}$ difference ≤ 3% |
| | | | | | | 105%WR$_{peak}$ | |
| Stachenfeld et al. [114] | 20-s averaging | VO$_2$ plateau of 150 mL/min; RER$_{max}$ ≥ 1.10, 1.15; ≥ 85% APMHR; La$_{max}$ ≥ 8 mmol | CYC | CSI | Different day | 115% WR$_{peak}$ reached in the CPET or 125% if the plateau has not been attained | VO$_2$ plateau of 150 mL/min |
| Straub et al. [115] | 15-s time average | NS | CYC | Ramp | 10-min passive | 1$^{st}$ min: 60% WR$_{peak}$ and then 110% WR$_{peak}$ | NS |
| Strom et al. [116] | 30-s time average | NS | TR | CSI | 3-min active (walking pace of 67 m/min) | 2 stages < CPET-WR$_{peak}$ | CPET *vs.* VP: VO$_{2max}$ difference ≤ 3% |
| Taylor et al. [117] | 15-breath average | NS | TR | CSI | 15-min active or passive | 1$^{st}$ min at 10 km/h (5% slope) and then ↑ 1 km/h > CPET-Speed$_{peak}$ | NS |
| Tucker et al. [118] | 2×15-s | NS | CYC | CSI | 5–10 min active | 100%WR$_{peak}$ | NS |
| Vogiatzis et al. [119] | NS | NS | CYC | CSI | 20-min passive | 110% WR$_{peak}$ | NS |
| Weatherwax et al. [120] | 2×15-s | NS | TR | Pseudo-ramp protocol | 20-min passive | 105% WR$_{peak}$ (Trials 1 and 2) | CPET *vs.* VP: VO$_{2max}$ difference ≤ 3% |
| Weatherwax et al. [15] | 2×15-s | NS | TR | Pseudo-ramp protocol | 20-min passive | 105% WR$_{peak}$ | CPET *vs.* VP: VO$_{2max}$ difference ≤ 3% |
| Weatherwax et al. [121] | 2×15-s | NS | TR | Pseudo-ramp protocol | 20-min passive | 105% WR$_{peak}$ | CPET *vs.* VP: VO$_{2max}$ difference ≤ 3% |
| Weatherwax et al. [122] | 2×15-s | NS | TR | DisCSI | 20-min passive | 3 min at 4.82 km/h and then ↑ 0.64 km/h > CPET-Speed$_{peak}$ (males) | CPET *vs.* VP: VO$_{2max}$ difference ≤ 3% |
| | | | | | | 3 min at 4.82 km/h and then ↑ 0.48 km/h > CPET-Speed$_{peak}$ (females) | |
| Wilhelm et al. [123] | 10-s moving average | NS | CYC | CSI | 5-min passive | 105%WR$_{peak}$ | NS |
| Williams et al. [124] | 20-s | NS | CYC | Ramp | 5-min active | 105%WR$_{peak}$ | NS |
| Wingo et al. [125] | 2×30-s | VO$_2$ plateau of 135 mL/min; HR within 5 bpm of that on the control test was obtained | CYC | CSI control / CSI post-15 min / CSI post-45 min | 20-min passive | 100% WR$_{peak}$ (if <1-min was completed during the last stage of the CPET) or 25 Watts > CPET-WR$_{peak}$ (if ≥1-min was completed during the last stage of the CPET) | VO$_2$ plateau of 135 mL/min |
| Yeh et al. [126] | NS | NS | TR | CSI | 10-min passive | 1 km/h > CPET-Speed$_{peak}$ or 5% slope every minute until exhaustion | NS |

**Abbreviations:** *APMHR* = age-predicted maximal heart rate; *CPET* = cardiopulmonary exercise test; *CSI* = continuous step-incremented; *CV* = coefficient of variation; *CYC* = cycling; *DisCSI* = discontinuous step-incremented; *HR* = heart rate; *HR$_{max}$* = maximal heart rate; *La$_{max}$* = maximal blood lactate concentration; *NS* = not stated; *RER$_{max}$* = maximal respiratory exchange ratio; *RPE* = rating of perceived exertion; *SD* = standard deviation; *SPV* = self-paced maximal oxygen uptake; *TR* = treadmill; *VO$_2$* = oxygen uptake; *VO$_{2max}$* = maximal oxygen uptake; *VP* = verification phase; *WR* = work rate; *WR$_{peak}$* = peak work rate. *Note*: whenever possible, authors were contacted to provide unpublished data.

Regarding the period between the CPET and verification phase procedure, 34 studies (43%) used a short-term active recovery (e.g. pedaling at light-intensity, walking at a slow pace, or stretching) of 1, 3, 5, 6, 8, 10, or 5–10 min, while 26 studies (33%) employed passive recovery of 5, 6, 9, 10, 15, 20, 60, or 60–90 min. Two studies (3%) employed a combination of passive

and active recovery and another (1%) used a self-paced approach where participants were permitted to choose their own WR. Three studies (4%) employed short-term recovery (e.g. 8–10 min) without stating whether it was active or passive. Fifteen studies (19%) carried out the verification phase on a different day to the CPET.

Sixty studies (75%) used square-wave verification phase protocols, while 20 studies (25%) used multistage verification protocols characterized by an initial warm-up stage. Overall, 53 studies (66%) adopted "supra $WR_{peak}$" verification phases based upon the $WR_{peak}$ achieved during the CPET (e.g. one treadmill or cycle ergometer WR stage higher than that completed in the CPET, or 105–130% of the $WR_{peak}$ achieved in the previous CPET). Seven studies (9%) used only 100% of $WR_{peak}$, while two other studies (3%) used both $WR_{peak}$ and supra $WR_{peak}$ verification phases. Three studies (4%) examined both sub and supra $WR_{peak}$ within the same study and one study (1%) used a predicted WR based on the following formula to elicit the participant's limit of tolerance within 180 s: power output = (finite work capacity ÷ 180 s) + critical power. Fourteen studies (18%) used only sub $WR_{peak}$ verification phases ranging from 85%-95% $WR_{peak}$ (typically two stages below the $WR_{peak}$ achieved during the CPET) (see Table 2).

Forty-two studies (53%) employed cut-off points to analyze differences between the highest $VO_2$ values obtained during the CPET and verification phase to confirm that $VO_{2max}$ was likely attained. Criteria for $VO_{2max}$ verification were frequently based on the intra-subject coefficient of variation acquired from the researchers' laboratories or from published literature, including a $VO_2$ difference $\leq$ 2%, $\leq$ 3%, $\leq$ 5.0–5.5%, $\leq$ 1.5–2.2 mL·kg$^{-1}$·min$^{-1}$, $\leq$ 50–150 mL/min, or alternative methods.

## Quantitative data synthesis: Differences between the highest $VO_2$ attained in the CPET and verification phase

Table 3 shows comparisons between the highest $VO_2$ values elicited in the CPET and verification phase for each study. Fig 2 displays the forest plots of effect sizes and 95% CIs for the highest $VO_2$ values (54 studies) based on the random effects meta-analysis results. Notably, the mean highest $VO_2$ values were similar between the CPET and verification phase (mean difference = 0.03 [95% CI = -0.01 to 0.06] L/min, $P$ = 0.15). Pooled data for $VO_{2max}$ following the CPET and verification phase showed no significant heterogeneity among the studies overall (see Fig 2). Except for one of the included studies judged to have a high risk of bias [68], the meta-analyzed studies were judged to have a low-risk of bias as shown by the funnel plot (Fig 3).

Results of subgroup analyses according to the characteristics of the verification phase protocol are summarized in Fig 4. There were no significant differences between the CPET and verification phase for the highest $VO_2$ values attained after stratifying studies for verification-phase intensity (mean difference = 0.03 [95% CI = -0.01 to 0.07] L/min, $P$ = 0.11), type of recovery utilized (mean difference = 0.02 [95%CI = -0.02 to 0.07] L/min, $P$ = 0.36), $VO_{2max}$ verification criterion adoption (mean difference = 0.02 [95% CI = -0.02 to 0.06] L/min, $P$ = 0.29), verification procedure with regards to whether or not it was performed on the same day as the CPET (mean difference = 0.03 [95%CI -0.01 to 0.06] L/min, $P$ = 0.21), or verification-phase duration (i.e. no longer than 80 s, from 81 to 120 s and longer than 120 s) (mean difference = 0.03 [95%CI -0.03 to 0.09] L/min, $P$ = 0.35).

Subgroup analyses regarding sex, cardiorespiratory fitness level, exercise modality, and CPET protocol are summarized in Table 4. The median time to exhaustion was 665 s (IQR, 600 s) for the CPET and 148 s (IQR, 110 s) for the verification phase. Considering all sub-

**Table 3. Overall comparisons in the meta-analyzed studies for the highest VO$_2$ values attained in the cardiopulmonary exercise test (CPET) and verification phase (VP) (k = 54).**

| Study | Specific Experimental Condition | | CPET | | | VP | | | % Weight | Mean Difference |
|---|---|---|---|---|---|---|---|---|---|---|
| | | | Mean [L/min] | SD [L/min] | Total | Mean [L/min] | SD [L/min] | Total | | IV, Random, 95%CI [L/min] |
| Alexander and Mier [54] | CPET protocol (CSI) | | 3.79 | 0.39 | 11 | 3.80 | 0.49 | 11 | 1.00% | -0.01 [-0.38, 0.36] |
| | CPET protocol (DisCSI) | | 3.94 | 0.40 | 11 | 3.84 | 0.45 | 11 | 1.00% | 0.10 [-0.25, 0.46] |
| Arad et al. [55] | N/A | | 2.18 | 0.61 | 35 | 2.26 | 0.65 | 35 | 1.40% | -0.08 [-0.38, 0.22] |
| Astorino and DeRevere [56] | CPET-VP recovery (8 min) VP intensity (105% WR$_{peak}$) | | 3.35 | 1.01 | 30 | 3.32 | 1.00 | 30 | 0.50% | 0.03 [-0.48, 0.54] |
| | CPET-VP recovery (10 min) VP intensity (110% WR$_{peak}$) | | 2.82 | 0.62 | 79 | 2.78 | 0.59 | 79 | 3.70% | 0.04 [-0.15, 0.23] |
| Astorino and White [57] | N/A | | 3.00 | 0.45 | 30 | 3.00 | 0.45 | 30 | 2.50% | 0.00 [-0.23, 0.23] |
| Astorino et al. [53] | Experimental groups (low CRF) | | 2.35 | 0.37 | 10 | 2.36 | 0.33 | 10 | 1.40% | -0.01 [-0.32, 0.30] |
| | Experimental groups (moderate CRF) | | 3.32 | 0.58 | 10 | 3.28 | 0.60 | 10 | 0.50% | 0.04 [-0.48, 0.56] |
| | Experimental groups (high CRF) | | 4.38 | 0.70 | 10 | 4.29 | 0.74 | 10 | 0.30% | 0.09 [-0.54, 0.72] |
| Astorino et al. [58] | Training effect (HIIT-Baseline) | | 2.51 | 0.62 | 14 | 2.50 | 0.61 | 14 | 0.60% | 0.01 [-0.45, 0.47] |
| | Training effect (HIIT—Week 3) | | 2.66 | 0.67 | 14 | 2.60 | 0.64 | 14 | 0.60% | 0.06 [-0.43, 0.55] |
| | Training effect (Control—Baseline) | | 2.94 | 0.72 | 14 | 2.87 | 0.71 | 14 | 0.50% | 0.07 [-0.46, 0.60] |
| | Training effect (Control—Week 3) | | 2.97 | 0.74 | 14 | 2.84 | 0.69 | 14 | 0.50% | 0.13 [-0.40, 0.66] |
| Astorino et al. [59] | N/A | | 2.55 | 0.62 | 14 | 2.57 | 0.61 | 14 | 0.60% | -0.02 [-0.47, 0.43] |
| Astorino et al. [60] | CPET-VP recovery (at least 24h) | | 2.37 | 0.69 | 15 | 2.31 | 0.75 | 15 | 0.50% | 0.06 [-0.45, 0.58] |
| | CPET-VP recovery (60 to 90 min) | | 2.72 | 0.65 | 9 | 2.73 | 0.72 | 9 | 0.30% | -0.01 [-0.64, 0.62] |
| Beltrami et al. [61] | Experimental groups (control group) | | 4.50 | 0.58 | 13 | 4.43 | 0.46 | 13 | 0.80% | 0.07 [-0.33, 0.47] |
| | Experimental groups (reverse group) | | 4.52 | 0.36 | 13 | 4.54 | 0.33 | 13 | 1.90% | -0.02 [-0.28, 0.24] |
| Beltz et al. [62] | CPET protocol (SPV) | | 3.84 | 0.28 | 16 | 3.74 | 0.50 | 16 | 1.70% | 0.10 [-0.18, 0.38] |
| | CPET protocol (Ramp) | | 3.86 | 0.28 | 16 | 3.77 | 0.50 | 16 | 1.70% | 0.09 [-0.19, 0.37] |
| Bisi et al. [63] | N/A | | 2.41 | 0.13 | 11 | 2.56 | 0.36 | 11 | 2.60% | -0.15 [-0.38, 0.08] |
| Chidnok et al. [64] | N/A | | 4.32 | 0.61 | 7 | 4.32 | 0.69 | 7 | 0.30% | 0.00 [-0.68, 0.68] |
| Colakoglu et al. [68] | N/A | | 4.11 | 0.69 | 9 | 4.56 | 0.60 | 9 | 0.40% | -0.45 [-1.05, 0.15] |
| Dalleck et al. [69] | N/A | | 2.33 | 0.76 | 18 | 2.31 | 0.76 | 18 | 0.50% | 0.02 [-0.48, 0.52] |
| Day et al. [35] | N/A | | 3.64 | 0.70 | 38 | 3.64 | 0.70 | 38 | 1.30% | 0.00 [-0.31, 0.31] |
| Dicks et al. [72] | N/A | | 3.84 | 0.65 | 28 | 3.72 | 0.60 | 28 | 1.20% | 0.12 [-0.21, 0.45] |
| Ducrocq et al. [74] | N/A | | 3.73 | 0.47 | 13 | 3.76 | 0.45 | 13 | 1.10% | -0.03 [-0.39, 0.32] |
| Elliott et al. [75] | N/A | | 4.26 | 0.61 | 8 | 4.26 | 0.70 | 8 | 0.30% | 0.00 [-0.64, 0.64] |
| Foster et al. [77] | VP exercise modality (TR) | | 4.09 | 0.97 | 20 | 4.03 | 1.16 | 20 | 0.30% | 0.06 [-0.60, 0.72] |
| | VP exercise modality (CYC) | | 3.95 | 0.75 | 20 | 4.06 | 0.75 | 20 | 0.60% | -0.11 [-0.57, 0.35] |
| Freeberg et al. [78] | N/A | | 3.49 | 0.85 | 30 | 3.49 | 0.85 | 30 | 0.70% | 0.00 [-0.43, 0.43] |
| Goodall et al. [79] | N/A | | 4.11 | 0.56 | 9 | 3.82 | 0.71 | 9 | 0.40% | 0.29 [-0.30, 0.88] |
| Hogg et al. [82] | N/A | | 4.87 | 0.43 | 14 | 4.82 | 0.48 | 14 | 1.20% | 0.05 [-0.29, 0.39] |
| Iannetta et al. [25] | WR$_{peak}$ 5 W/min | 1$^{st}$ VP at 110% WR$_{peak}$ (25 W/min) | 3.35 | 0.68 | 11 | 3.30 | 0.65 | 11 | 0.4% | 0.05 [-0.51, 0.61] |
| | | 2$^{nd}$ VP at 110% WR$_{peak}$ (5 W/min) | 3.35 | 0.68 | 11 | 3.45 | 0.68 | 11 | 0.4% | -0.10 [-0.67, 0.47] |
| | WR$_{peak}$ 10 W/min | 1$^{st}$ VP at 110% WR$_{peak}$ (25 W/min) | 3.44 | 0.67 | 11 | 3.33 | 0.62 | 11 | 0.4% | 0.11 [-0.43, 0.65] |
| | | 2$^{nd}$ VP at 110% WR$_{peak}$ (10 W/min) | 3.44 | 0.67 | 11 | 3.47 | 0.7 | 11 | 0.4% | -0.03 [-0.60, 0.54] |
| | WR$_{peak}$ 15 W/min | 1$^{st}$ VP at 110% WR$_{peak}$ (25 W/min) | 3.44 | 0.69 | 11 | 3.3 | 0.68 | 11 | 0.4% | 0.14 [-0.43, 0.71] |
| | | 2$^{nd}$ VP at 110% WR$_{peak}$ (15 W/min) | 3.44 | 0.69 | 11 | 3.39 | 0.64 | 11 | 0.4% | 0.05 [-0.51, 0.61] |
| | WR$_{peak}$ 25 W/min | 1$^{st}$ VP at 110% WR$_{peak}$ (25 W/min) | 3.44 | 0.74 | 11 | 3.28 | 0.67 | 11 | 0.4% | 0.16 [-0.43, 0.75] |
| | | 2$^{nd}$ VP at 110% WR$_{peak}$ (25 W/min) | 3.44 | 0.74 | 11 | 3.29 | 0.66 | 11 | 0.4% | 0.15 [-0.44, 0.74] |
| | WR$_{peak}$ 30 W/min | 1$^{st}$ VP at 110% WR$_{peak}$ (25 W/min) | 3.44 | 0.72 | 11 | 3.31 | 0.67 | 11 | 0.4% | 0.13 [-0.45, 0.71] |
| | | 2$^{nd}$ VP at 110% WR$_{peak}$ (30 W/min) | 3.44 | 0.72 | 11 | 3.28 | 0.65 | 11 | 0.4% | 0.16 [-0.41, 0.73] |
| Jamnick et al. [84] | CPET protocol (CSI$_1$: 1-min stage length) | | 4.72 | 0.41 | 17 | 4.65 | 0.45 | 17 | 1.60% | 0.07 [-0.22, 0.36] |
| | CPET protocol (CSI$_3$: 3-min stage length) | | 4.62 | 0.42 | 17 | 4.56 | 0.46 | 17 | 1.50% | 0.06 [-0.23, 0.36] |
| | CPET protocol (CSI$_5$: 5-min stage length) | | 4.55 | 0.46 | 17 | 4.55 | 0.47 | 17 | 1.30% | 0.00 [-0.31, 0.31] |
| | CPET protocol (CSI$_7$: 7-min stage length) | | 4.44 | 0.42 | 17 | 4.37 | 0.46 | 17 | 1.50% | 0.07 [-0.22, 0.36] |
| | CPET protocol (CSI$_{10}$: 10-min stage length) | | 4.35 | 0.43 | 17 | 4.23 | 0.51 | 17 | 1.30% | 0.12 [-0.20, 0.43] |
| Jamnick et al. [85] | N/A | | 3.24 | 0.57 | 57 | 3.25 | 0.57 | 57 | 3.00% | -0.02 [-0.23, 0.19] |
| Johnson et al. [86] | N/A | | 3.31 | 0.76 | 11 | 3.34 | 0.82 | 11 | 0.30% | -0.03 [-0.69, 0.63] |
| Keiller and Gordon [87] | N/A | | 3.65 | 0.71 | 11 | 3.50 | 0.58 | 11 | 0.50% | 0.15 [-0.39, 0.69] |
| Kirkeberg et al. [88] | CPET protocol (short-term CSI) | | 4.43 | 0.48 | 12 | 4.41 | 0.54 | 12 | 0.80% | 0.03 [-0.38, 0.43] |
| | CPET protocol (middle-term CSI) | | 4.40 | 0.46 | 12 | 4.27 | 0.40 | 12 | 1.00% | 0.13 [-0.21, 0.47] |
| | CPET protocol (large-term CSI) | | 4.42 | 0.42 | 12 | 4.36 | 0.45 | 12 | 1.00% | 0.06 [-0.29, 0.41] |
| Kramer et al. [91] | N/A | | 3.45 | 0.29 | 15 | 3.42 | 0.25 | 15 | 3.50% | 0.03 [-0.16, 0.22] |

*(Continued)*

**Table 3.** (Continued)

| Study | Specific Experimental Condition | CPET | | | VP | | | % Weight | Mean Difference |
|---|---|---|---|---|---|---|---|---|---|
| | | Mean [L/min] | SD [L/min] | Total | Mean [L/min] | SD [L/min] | Total | | IV, Random, 95%CI [L/min] |
| Mann et al. [93] | N/A | 4.11 | 0.78 | 10 | 4.13 | 0.85 | 10 | 0.30% | -0.02 [-0.74, 0.70] |
| Mann et al. [92] | N/A | 3.80 | 0.87 | 32 | 3.78 | 0.92 | 32 | 0.70% | 0.03 [-0.41, 0.46] |
| Mauger et al. [94] | N/A | 4.66 | 0.55 | 14 | 4.65 | 0.59 | 14 | 0.70% | 0.01 [-0.42, 0.43] |
| McGawley [95] | N/A | 4.08 | 0.47 | 10 | 4.01 | 0.46 | 10 | 0.80% | 0.08 [-0.33, 0.48] |
| Midgley et al. [39] | VP exercise modality (CYC) | 3.86 | 0.39 | 10 | 3.92 | 0.47 | 10 | 0.90% | -0.05 [-0.43, 0.33] |
| | VP exercise modality (TR) | 4.05 | 0.47 | 10 | 3.96 | 0.38 | 10 | 0.90% | 0.10 [-0.28, 0.47] |
| Midgley et al. [98] | CPET protocol (CSI 1-min stages) | 4.09 | 0.54 | 9 | 4.07 | 0.53 | 9 | 0.50% | 0.03 [-0.47, 0.52] |
| | CPET protocol (DisCSI 2-min stages) | 4.10 | 0.52 | 9 | 4.08 | 0.52 | 9 | 0.60% | 0.02 [-0.46, 0.50] |
| | CPET protocol (DisCSI 3-min stages) | 3.98 | 0.49 | 9 | 4.07 | 0.53 | 9 | 0.60% | -0.09 [-0.56, 0.38] |
| Midgley et al. [97] | N/A | 4.03 | 0.42 | 16 | 4.01 | 0.44 | 16 | 1.50% | 0.01 [-0.28, 0.31] |
| Mier et al. [99] | N/A | 3.64 | 0.38 | 10 | 3.77 | 0.38 | 10 | 1.20% | -0.13 [-0.46, 0.20] |
| Murias et al. [100] | VP intensity (younger: 85% WR$_{peak}$) | 3.73 | 0.51 | 8 | 3.76 | 0.48 | 8 | 0.60% | -0.03 [-0.52, 0.45] |
| | VP intensity (younger: 105% WR$_{peak}$) | 3.90 | 0.65 | 22 | 3.89 | 0.64 | 22 | 0.90% | 0.02 [-0.36, 0.40] |
| | VP intensity (older: 85% WR$_{peak}$) | 2.18 | 0.55 | 8 | 2.18 | 0.55 | 8 | 0.50% | 0.00 [-0.54, 0.54] |
| | VP intensity (older: 105% WR$_{peak}$) | 2.52 | 0.54 | 23 | 2.57 | 0.51 | 23 | 1.40% | -0.05 [-0.36, 0.25] |
| Niemela et al. [104] | N/A | 3.05 | 0.55 | 16 | 3.05 | 0.49 | 16 | 1.00% | 0.00 [-0.36, 0.35] |
| Niemeyer et al. [105] | N/A | 4.06 | 0.43 | 24 | 4.06 | 0.46 | 24 | 2.10% | 0.00 [-0.25, 0.24] |
| Niemeyer et al. [106] | N/A | 4.01 | 0.47 | 46 | 3.95 | 0.51 | 46 | 3.30% | 0.06 [-0.14, 0.26] |
| Nolan et al. [107] | CPET-VP recovery (20 min) VP intensity (105% WR$_{peak}$) | 3.64 | 0.61 | 12 | 3.66 | 0.58 | 12 | 0.60% | -0.02 [-0.50, 0.46] |
| | CPET-VP recovery (20 min) VP intensity (115% WR$_{peak}$) | 3.68 | 0.59 | 12 | 3.64 | 0.61 | 12 | 0.60% | 0.04 [-0.44, 0.52] |
| | CPET-VP recovery (60 min) VP intensity (105% WR$_{peak}$) | 3.60 | 0.58 | 12 | 3.60 | 0.58 | 12 | 0.60% | 0.00 [-0.46, 0.46] |
| | CPET-VP recovery (60 min) VP intensity (115% WR$_{peak}$) | 3.65 | 0.54 | 12 | 3.58 | 0.60 | 12 | 0.60% | 0.07 [-0.38, 0.52] |
| Poole et al. [37] | N/A | 4.03 | 0.28 | 7 | 3.95 | 0.29 | 7 | 1.50% | 0.08 [-0.22, 0.38] |
| Possamai et al. [108] | N/A | 3.83 | 0.41 | 19 | 3.72 | 0.42 | 19 | 1.90% | 0.11 [-0.15, 0.37] |
| Rossiter et al. [38] | VP intensity (105%WR$_{peak}$) | 4.15 | 0.50 | 5 | 4.09 | 0.45 | 5 | 0.40% | 0.06 [-0.53, 0.65] |
| | VP intensity (95%WR$_{peak}$) | 4.11 | 0.48 | 5 | 4.12 | 0.53 | 5 | 0.30% | -0.01 [-0.64, 0.61] |
| Sabino-Carvalho et al. [110] | Pre-CPET intervention (IPC) | 4.24 | 0.46 | 16 | 4.23 | 0.40 | 16 | 1.50% | 0.01 [-0.29, 0.31] |
| | Pre-CPET intervention (Sham) | 4.23 | 0.48 | 16 | 4.23 | 0.43 | 16 | 1.30% | 0.01 [-0.31, 0.32] |
| | Pre-CPET intervention (Control) | 4.23 | 0.38 | 16 | 4.15 | 0.32 | 16 | 2.20% | 0.08 [-0.17, 0.32] |
| Scharhag-Rosenberger et al. [111] | CPET-VP recovery (same day after 10 min) | 3.82 | 0.99 | 34 | 3.72 | 0.99 | 34 | 0.60% | 0.10 [-0.37, 0.57] |
| | CPET-VP recovery (different day) | 3.82 | 0.99 | 34 | 3.75 | 1.00 | 34 | 0.60% | 0.07 [-0.40, 0.54] |
| Sedgeman et al. [113] | VP intensity (WR$_{peak}$ minus 2-stages) | 3.69 | 0.41 | 13 | 3.70 | 0.49 | 13 | 1.10% | -0.01 [-0.36, 0.34] |
| | VP intensity (105%WR$_{peak}$) | 3.71 | 0.51 | 13 | 3.64 | 0.50 | 13 | 0.90% | 0.07 [-0.31, 0.46] |
| Straub et al. [115] | N/A | 3.86 | 0.73 | 16 | 3.84 | 0.68 | 16 | 0.60% | 0.02 [-0.47, 0.51] |
| Taylor et al. [117] | N/A | 4.03 | 0.53 | 19 | 3.83 | 0.52 | 19 | 1.20% | 0.21 [-0.13, 0.54] |
| Weatherwax et al. [120] | N/A | 2.29 | 0.73 | 16 | 2.29 | 0.73 | 16 | 0.50% | 0.00 [-0.50, 0.51] |
| Weatherwax et al. [15] | Training effect (standardized—baseline) | 2.03 | 0.62 | 20 | 2.03 | 0.60 | 20 | 0.90% | 0.00 [-0.38, 0.38] |
| | Training effect (standardized—week 12) | 2.17 | 0.62 | 20 | 2.18 | 0.63 | 20 | 0.90% | -0.01 [-0.40, 0.38] |
| | Training effect (individualized—baseline) | 2.37 | 0.79 | 19 | 2.37 | 0.77 | 19 | 0.50% | 0.00 [-0.50, 0.50] |
| | Training effect (individualized—week 12) | 2.63 | 0.89 | 19 | 2.65 | 0.89 | 19 | 0.40% | -0.02 [-0.59, 0.55] |
| Weatherwax et al. [121] | Training effect (control—baseline) | 2.18 | 0.74 | 8 | 2.16 | 0.73 | 8 | 0.30% | 0.02 [-0.70, 0.74] |
| | Training effect (control—week 12) | 2.11 | 0.73 | 8 | 2.10 | 0.69 | 8 | 0.30% | 0.01 [-0.69, 0.71] |
| | Training effect (standardized—baseline) | 2.03 | 0.62 | 20 | 2.03 | 0.60 | 20 | 0.90% | 0.00 [-0.38, 0.38] |
| | Training effect (standardized—week 12) | 2.17 | 0.62 | 20 | 2.18 | 0.63 | 20 | 0.90% | -0.01 [-0.40, 0.38] |
| | Training effect (individualized—baseline) | 2.37 | 0.79 | 19 | 2.37 | 0.77 | 19 | 0.50% | 0.00 [-0.50, 0.50] |
| | Training effect (individualized—week 12) | 2.63 | 0.89 | 19 | 2.65 | 0.89 | 19 | 0.40% | -0.02 [-0.59, 0.55] |
| Weatherwax et al. [122] | Experimental groups (males) | 3.98 | 0.36 | 18 | 3.94 | 0.32 | 18 | 2.60% | 0.04 [-0.19, 0.26] |
| | Experimental groups (females) | 2.68 | 0.13 | 6 | 2.67 | 0.10 | 6 | 8.00% | 0.01 [-0.12, 0.14] |

**Abbreviations:** CI = confidence interval; CPET = cardiopulmonary exercise test; CRF = cardiorespiratory fitness level; CSI = continuous step-incremented; CYC = cycling; DisCSI = discontinuous step-incremented; HIIT = high-intensity interval training; IPC = ischemic preconditioning; N/A = not applicable; TR = treadmill; SD = standard deviation; SPV = self-paced maximal oxygen uptake; VO$_2$ = oxygen uptake; VP = verification phase; WR$_{peak}$ = peak work rate; W/min = incremental phase based on watts *per* minute. *Note*: whenever possible, authors were contacted to provide unpublished data. %Weight = weight attributed to each study due to its statistical power.

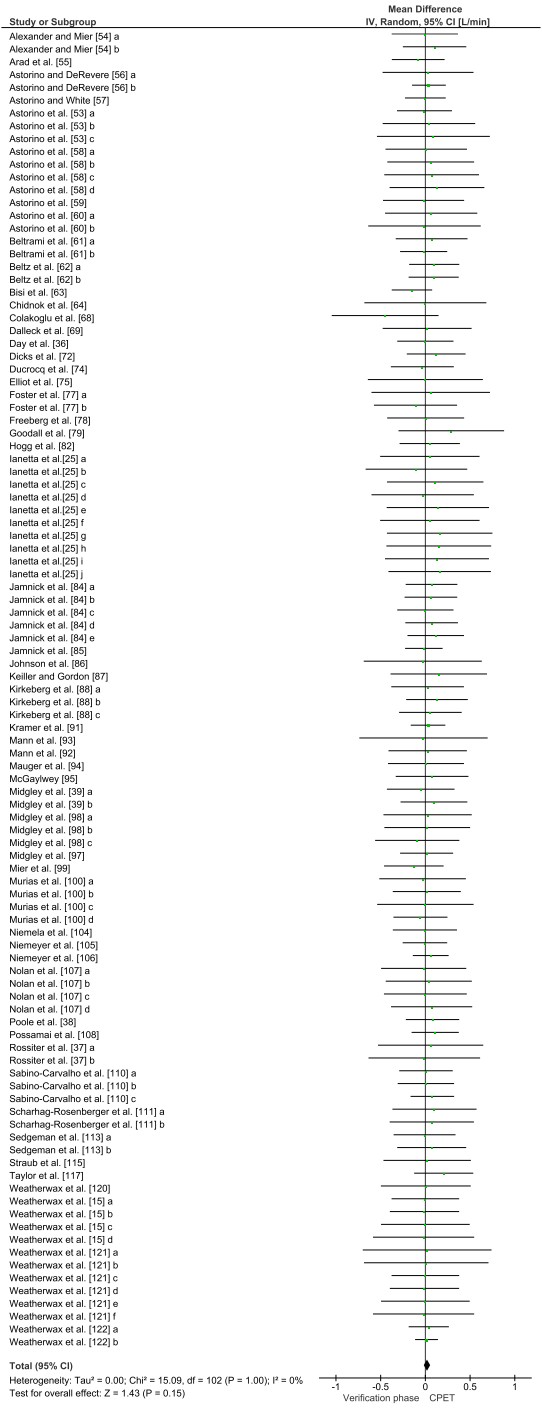

**Fig 2. Forest plot of all studies included in the meta-analysis (k = 54) for the highest VO$_2$ responses attained in the cardiopulmonary exercise test and verification phase using random effects analyses.** Data are reported as mean differences (MD) adjusted for control data (95% CIs).

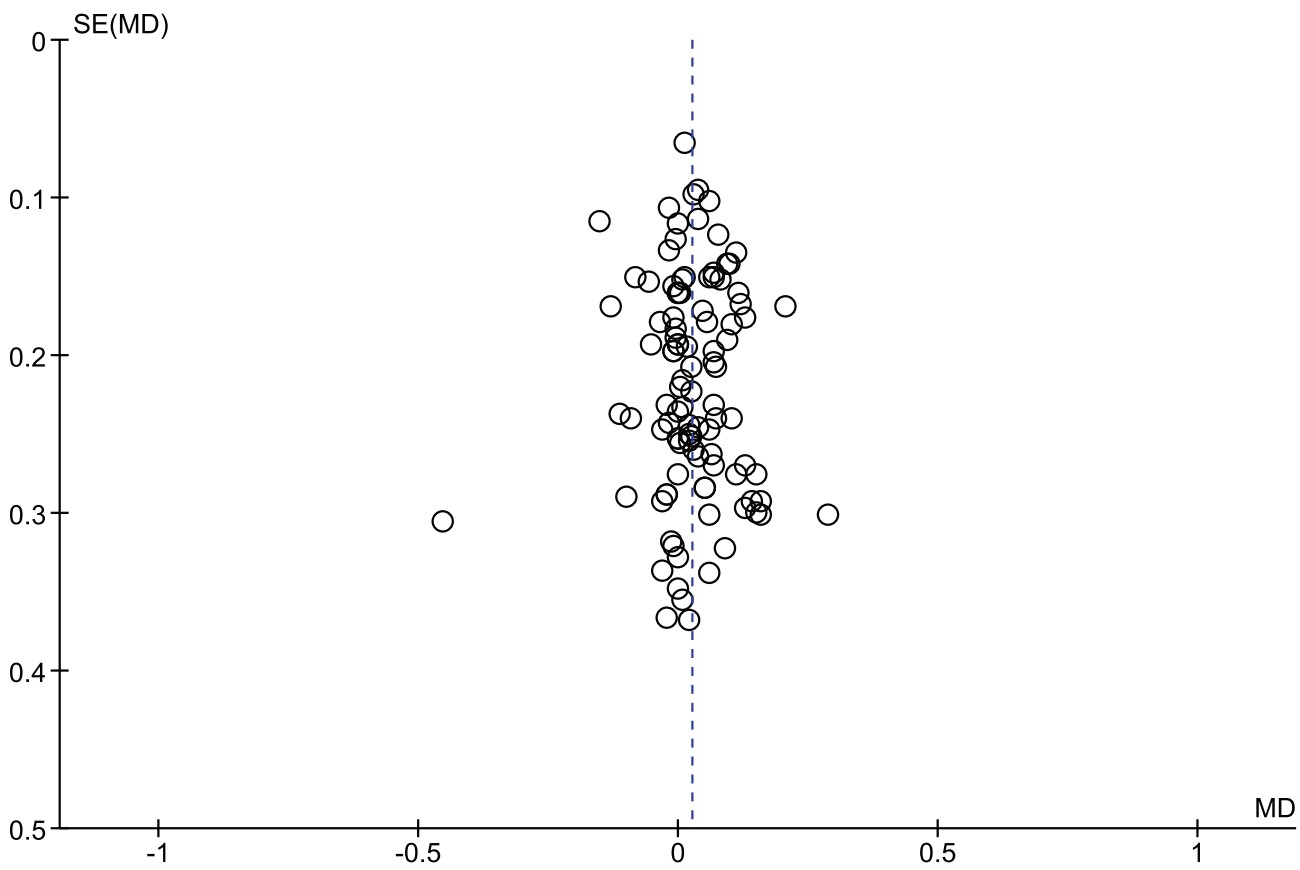

**Fig 3. Funnel plot assessment of publication bias for the studies investigating the highest VO$_2$ responses attained in the cardiopulmonary exercise test and verification phase.**

analyses presented in Table 4, there were no significant differences between the CPET and verification phase for VO$_{2max}$ ($P = 0.18$ to $P = 0.71$).

## Discussion

A growing number of studies have included the verification phase procedure to increase confidence that the highest possible VO$_2$ has been elicited by apparently healthy adults during a CPET. To the best of our knowledge this is the first systematic review and meta-analysis of these studies, and evidences that 90% of which have been published since 2009. The major findings were: (a) in general, the verification phase protocols elicited similar highest VO$_2$ values to those obtained in the preceding CPET protocols; and (b) concordance between the highest VO$_2$ values in the CPETs and verification phases were not affected by sex, cardiorespiratory fitness level, exercise modality, CPET protocol, or verification phase protocol.

The present systematic review and meta-analysis shows that the highest mean VO$_2$ values elicited by verification phase bouts were similar to those elicited in continuous ramp or pseudo-ramp CPET protocols in the majority of studies. In fact, the mean absolute difference of 0.03 L/min for the 54 studies included in the meta-analysis represents a relative difference of only 0.85% between the highest VO$_2$ values attained in the CPET and verification phase. This is within the most commonly adopted measures of test variability of 2–3% [57, 97]. The

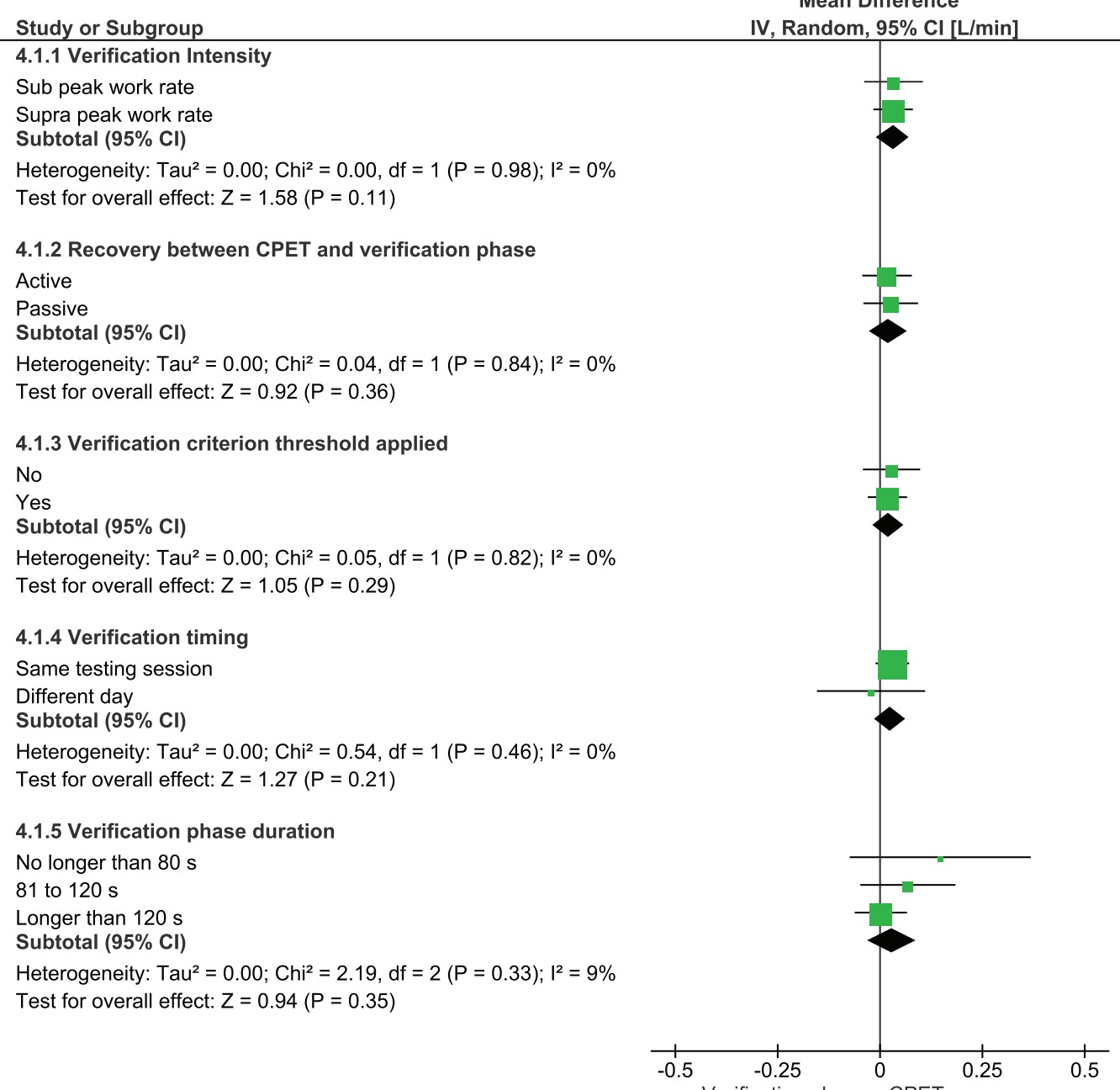

**Fig 4. Mean differences (95% CIs) between the highest VO$_2$ responses in the cardiopulmonary exercise test (CPET) and verification phase according to the verification-phase characteristics for intensity (i.e. sub WR$_{peak}$ *vs.* supra WR$_{peak}$), recovery (i.e. active *vs.* passive), adoption of criterion threshold (i.e. yes *vs.* no), timing (performed on the same day *vs.* a different day to the CPET), and duration (i.e. no longer than 80 s, from 81 to 120 s and longer than 120 s).**

present findings also provide evidence that the similarity between the highest VO$_2$ values attained during the CPETs and verification phases are not affected by sex, cardiorespiratory fitness, exercise modality, CPET protocol design, or how the verification phase was performed (see Table 4 and Fig 4). This contrasts with traditional VO$_{2max}$ criteria, which are test-protocol

**Table 4. Subgroup analyses for the cardiopulmonary exercise test (CPET) and verification phase (VP).**

| | | Time to exhaustion (s) | | | VO$_{2max}$ (L/min) | | | |
| --- | --- | --- | --- | --- | --- | --- | --- | --- |
| | N | CPET Mean ± SD | VP Mean ± SD | N | CPET Mean ± SD | VP Mean ± SD | Effect Size (95% CI) | P-value |
| **Sex** | | | | | | | | |
| Male | 146 | 734 ± 90 | 244 ± 43 | 630 | 3.95 ± 0.48 | 3.93 ± 0.50 | 0.02 (-0.02 to 0.08) | 0.25 |
| Female | 23 | 659 ± 119 | 152 ± 46 | 68 | 2.63 ± 0.39 | 2.58 ± 0.40 | 0.05 (-0.08 to 0.12) | 0.71 |
| Both | 677 | 765 ± 140 | 146 ± 28 | 941 | 3.24 ± 0.67 | 3.21 ± 0.67 | 0.03 (-0.04 to 0.08) | 0.50 |
| **Cardiorespiratory fitness level** | | | | | | | | |
| Low | 170 | 617 ± 111 | 150 ± 36 | 322 | 2.30 ± 0.65 | 2.32 ± 0.65 | 0.02 (-0.07 to 0.11) | 0.63 |
| Moderate | 362 | 790 ± 101 | 200 ± 40 | 565 | 3.49 ± 0.61 | 3.45 ± 0.63 | 0.04 (-0.02 to 0.11) | 0.21 |
| High | 346 | 792 ± 149 | 161 ± 27 | 716 | 3.94 ± 0.55 | 3.90 ± 0.55 | 0.04 (-0.02 to 0.08) | 0.18 |
| **Exercise modality** | | | | | | | | |
| CYC | 477 | 823 ± 143 | 155 ± 29 | 916 | 3.47 ± 0.59 | 3.45 ± 0.59 | 0.02 (-0.03 to 0.07) | 0.43 |
| TR | 386 | 688 ± 110 | 189 ± 34 | 771 | 3.59 ± 0.58 | 3.56 ± 0.58 | 0.03 (-0.02 to 0.08) | 0.22 |
| **CPET protocol** | | | | | | | | |
| DisCSI | 92 | 876 ± 120 | 156 ± 28 | 169 | 3.90 ± 0.52 | 3.87 ± 0.51 | 0.03 (-0.05 to 0.11) | 0.49 |
| CSI | 472 | 696 ± 105 | 209 ± 40 | 924 | 3.71 ± 0.56 | 3.69 ± 0.58 | 0.02 (-0.03 to 0.07) | 0.38 |
| Ramp | 284 | 848 ± 171 | 121 ± 23 | 578 | 3.16 ± 0.63 | 3.13 ± 0.62 | 0.03 (-0.04 to 0.09) | 0.44 |

Group weighted mean differences in maximal oxygen uptake (VO$_{2max}$) according to sex, cardiorespiratory fitness level, exercise testing modality, and CPET protocol.
**Abbreviations:** *CI* = confidence interval; *CPET* = cardiopulmonary exercise test; *CSI* = continuous step-incremented; CYC = cycling; *DisCSI* = discontinuous step-incremented; *TR* = treadmill; *SD* = standard deviation; *VP* = verification phase.

dependent and vary according to the individual's physical characteristics [28, 29]. Day et al. [35], for example, observed that participants with lower cardiorespiratory fitness had a lower tendency to exhibit a deceleration in the VO$_2$ response at the end of a CPET compared to those with higher cardiorespiratory fitness and, therefore, are less likely to exhibit a VO$_2$ plateau.

Six of the 54 meta-analyzed studies reported significant mean differences between the highest VO$_2$ values observed in the CPET and verification phase [25, 55, 56, 68, 87, 95]. Astorino and DeRevere [56], for example, observed significantly higher mean VO$_{2max}$ values by 0.03 and 0.04 L/min during the CPET than in the verification phase for two samples of participants heterogeneous for cardiorespiratory fitness. However, sub-group analyses revealed that while maximal VO$_2$ in the CPET was higher than that attained in the verification phase for participants with moderate and high cardiorespiratory fitness, the opposite was true for those with lower cardiorespiratory fitness. Similar findings have been reported by Arad et al. [55], indicating that cardiorespiratory fitness level may be a key moderator of the differences between the highest VO$_2$ values attained in the CPET and verification phase. A plausible explanation is that individuals with low cardiorespiratory fitness are more susceptible to stopping early during the CPET due to fatigue-associated symptoms [29], which would tend to result in lower VO$_2$ values. In the present meta-analyses, the mean VO$_{2max}$ in the verification phase was 8% higher than in the CPET in the low cardiorespiratory fitness group, but 12% and 10% higher in the CPET than in the verification phase in the moderate and high cardiorespiratory fitness groups, respectively (see Table 4). The lack of statistical significance, however, highlights the uncertainty regarding the effects of cardiorespiratory fitness on the differences between the highest VO$_2$ values in the CPET and verification phase.

Regarding verification-phase duration, Keiller and Gordon [87] observed significantly higher VO$_2$ values during the incremental treadmill CPETs versus the verification phase with a mean duration of approximately 2 min. This is consistent with the findings of McGawley

[95] for 10 recreational runners who performed five consecutive treadmill CPET trials, plus an appended verification phase with a mean duration of < 2 min. Iannetta et al. [25] analyzed the $VO_2$ responses to ramp-incremented cycling CPETs with WR increments of 5, 10, 15, 25, and 30 W/min, each followed by two verification phases performed at different WRs. The verification phase bouts performed at 110% of the $WR_{peak}$ from ramp protocols with ramp rates of 25 and 30 W/min (i.e. short verification phase bouts of ~ 80 s) yielded $VO_2$ values significantly lower than those attained in the CPETs. In contrast, the highest $VO_2$ values attained during verification phase bouts based on slower WR increments of 5, 10, and 15 W/min, which allowed sufficient time for $VO_{2max}$ attainment (i.e. 162, 122 and 103 s, respectively) were not different to those achieved in the preceding CPETs. Although the aforementioned studies suggest that verification phase duration is a key moderator for the mean differences between the highest $VO_2$ observed in the CPET and verification phase, our sub-analysis found no difference for verification-phase durations of ≤ 80 s, ranging from 81 to 120 s, and > 120 s (see Fig 4). Notably, however, only three studies reported short durations of 80 s or less [25, 79, 113] and the lack of statistical significance may be due to the paucity of data.

In contrast to the aforementioned studies [25, 87, 95], Colakoglu et al. [68] observed significantly lower $VO_2$ values in the CPET versus the verification phase in nine cycling and track and field athletes. According to Midgley et al. [97], if the mean highest $VO_2$ attained in the verification phase is significantly higher than in the CPET, the investigator should consider that the CPET protocol was inadequate in eliciting the highest possible $VO_2$ response in all or some of the participants. In the study by Colakoglu et al. [68], participants performed a prolonged step-incremented CPET consisting of one 4-min, three 2-min, and then 1-min increments until volitional exhaustion after 1 h of recovery from a submaximal CPET of at least four 5-min stages. It is feasible that the procedures performed before the maximal CPET may have led to poor participant motivation, lack of effort and premature fatigue in the following test. Additionally, the four verification phase bouts at 100%, 105%, 110%, and 115% of the $WR_{peak}$ attained in the CPET were performed on four different days to the CPET without any preceding maximal exercise. This also may have positively favored the significantly higher mean $VO_2$ values in the verification phase compared to the CPET and contrasts with the same-day verification phase used by Keiller and Gordon [87], McGawley [95], and Iannetta et al. [25].

An aim of the present systematic review was to suggest best practices for the application of verification phase protocols. The subgroup analyses revealed no systematic bias between the highest $VO_2$ values observed in the CPET and verification phase according to the verification-phase intensity (i.e. sub $WR_{peak}$ *vs.* supra $WR_{peak}$), type of recovery between the CPET and verification phase (i.e. active *vs.* passive), whether a $VO_{2max}$ criterion threshold was used for the CPET (i.e. yes *vs.* no), whether the verification phase was performed in the same testing session or on a different day, and the verification-phase duration (see Fig 4). Considering that differences in the verification phase procedure do not appear to influence its effectiveness, a specific verification procedure currently cannot be recommended. However, some caution must be exercised to avoid an inappropriately high verification-phase WR that results in a short test duration and insufficient time for the highest possible $VO_2$ to be elicited [25], especially in untrained individuals characterized by slow $VO_2$ kinetics [127]. Midgley et al. [97] stated that this is a plausible rationale for the early recommendations of Thoden [128], that individuals who do not reach 3 min in a supra $WR_{peak}$ verification phase should undertake a subsequent verification phase at the same WR or one stage lower than verification-phase the last completed WR stage in the CPET. Poole and Jones [2] suggested that researchers should select a WR that is sufficiently higher than the $WR_{peak}$ attained in the CPET, such as ~110% $WR_{peak}$, to give the $VO_2$ signal for the higher WR the opportunity to emerge from the extant noise. If the subsequent verification phase produces a $VO_2$ plateau signifying $VO_{2max}$, this signal would

be lower than expected for the WR based on the previous $VO_2$-WR slope. Conversely, Iannetta et al. [25] advocated a verification-phase WR lower than the $WR_{peak}$ attained in the CPET in order to allow $VO_{2max}$ to be elicited, since WRs above critical power should elicit $VO_{2max}$ if the time to exhaustion is sufficiently long. Midgley et al. [39] proposed an alternative approach based on a multistage verification phase protocol that combines WRs below and above $WR_{peak}$ to obtain a protocol that incorporates a supra $WR_{peak}$ intensity with a relatively prolonged verification-phase duration. This approach has since been adopted in other studies [39, 53, 54, 61, 62, 64, 69, 76, 82, 87, 89, 90, 99, 104, 108, 110, 111, 115, 117, 122]. Notably, the only study to observe a statistically significant influence of verification phase intensity employed a multistage verification phase protocol incorporating 2 min at 50% of $WR_{peak}$, increasing to 70% for an additional minute, and then 105 or 115% until volitional exhaustion [107]. Based on their findings, the authors recommended the use of 105% of the $WR_{peak}$ attained in the CPET rather than 115% $WR_{peak}$. The confounding results and various recommended approaches regarding the verification phase intensity indicates that more research is required before an evidence-based recommendation can be made.

Regarding the recovery time between the CPET and verification phase, intervals between 10–20 min have been commonly used, although in total a wide range of intervals from 1–3 min [65, 77, 88, 113] to 90 min [41] have been used. The present meta-analysis found no significant effect of recovery time on minimizing the difference between the mean $VO_2$ elicited in the CPET and verification phase. An alternative method is to perform the verification phase on a separate day, although the additional visit to the laboratory and the day-to-day variability in $VO_{2max}$ [129] might considerably reduce the utility and robustness of this approach. Scharhag-Rosenberger et al. [111] specifically investigated this issue by comparing a 10-min recovery to a verification phase performed on a separate day. No significant difference was observed between the two verification protocols, even though the time to exhaustion was significantly longer when the verification phase was performed on a separate day (2:06 ± 0:22 min *vs*. 2:42 ± 0:38 min). These findings suggest no advantage in performing the CPET and verification phase on separate days.

Inadequate data processing may negatively impact the utility of the verification phase procedure. Myers et al. [36] suggested small sampling intervals such as 5 and 10 s result in unacceptable variability in $VO_2$ data, whereas large intervals such as 60 s may not be sufficiently sensitive to accurately track rapid changes in $VO_2$ such as those observed in ramp and pseudo-ramp CPET protocols. Midgley et al. [130] observed that the reproducibility of $VO_{2max}$ during continuous step-incremented treadmill CPETs is not affected by the length of the $VO_2$ time-average interval between the range of 10 to 60 s, however, the actual $VO_{2max}$ values were significantly different between time averages. The authors suggested that a 30-s stationary time-average for CPETs provides a good compromise between removing noise while maintaining the underlying trend in the $VO_2$ data. However, no study to date has addressed the effect of the $VO_2$ sampling interval on the verification phase.

A final issue to be addressed refers to appropriate criteria to accept that the highest possible $VO_2$ has been achieved. The most common criterion used in the reviewed studies is that the highest $VO_2$ observed in the verification phase should not exceed 3% of the highest $VO_2$ obtained in the CPET. This threshold can be justified by the technical error of measurement and intra-individual biological variation associated with the determination of $VO_{2max}$ [15, 56, 57, 62, 63, 69, 71, 78, 82, 86, 89–91, 95, 107, 108, 113, 116, 120–122]. The more restrictive value of ≤ 2% [97, 110] and the less restrictive values of ≤ 5–5.5% [104–106, 111] may also be appropriate for single or different-day variability. Further research is required before an appropriate verification-phase threshold can be recommended, which provides a high degree of confidence

that the difference between the highest $VO_2$ values observed in the CPET and verification phase are beyond the technical error of measurement and intra-individual biological variation.

Some limitations of the present review need to be acknowledged. First, the meta-analysis only included 79% of the participants that underwent CPET with verification phase protocols in the 80 studies included in the systematic review. This issue was due to unsuccessful attempts to acquire the required unpublished information from some authors. Second, the meta-analysis was based on comparison of the highest $VO_2$ responses in the CPET and verification phase averaged across study participants. Noakes [131] criticized this approach, stating that the CPET is performed on individuals and not groups and, therefore, the group average approach does not identify individuals who may not have attained $VO_{2max}$. A meta-analysis using individual participant data is therefore required. Finally, the present systematic review and meta-analysis comprised only apparently healthy adults and it is still unclear to what extent the use of the verification phase procedure is applicable to special or clinical populations. A growing number of studies have included special or clinical populations such as obese adults [132, 133], breast and prostate cancer survivors [134], wheelchair athletes [135], individuals with spinal-cord injuries [136], patients with heart failure [137] or cystic fibrosis [138–140], and pediatric populations [141–147], including children with spina bifida in an outpatient condition [148], and adolescents with cystic fibrosis [149].

## Conclusions

The present meta-analysis showed that the effect sizes calculated from the highest mean $VO_2$ in apparently healthy adults were similar between CPETs and verification phases performed on a cycle ergometer or treadmill. Furthermore, mean differences between the highest $VO_2$ values elicited in the CPETs and verification phases were not affected by participant characteristics, exercise modality, or the CPET and verification protocol design. Our findings indicate that from a practical perspective, different procedures may be applied to establish similar highest mean $VO_2$ responses during the verification phase as compared to the ramp or continuous step-incremented CPETs. It is worth mentioning, however, that some caution must be exercised concerning the selection of sub or supra $WR_{peak}$ verification phases, since any exercise above the critical power must be of sufficient duration to allow the achievement of the highest possible $VO_2$ response in the verification phase. Our data reinforce the notion that a verification phase applied after ramp or continuous step-incremented CPETs may provide additional and unbiased evidence that the highest possible $VO_2$ has been achieved. On the other hand, the invalidation of the highest $VO_2$ obtained in CPETs by subsequent verification phases was less likely on a group basis. The mean differences in highest $VO_2$ responses were typically within the test-retest variability of the experimental protocols employed. Accordingly, our findings support the usefulness of the verification phase to confirm the likely attainment of $VO_2$ on incremental CPET. However, the necessity or mandatory application of the verification phase, especially constant supra $WR_{peak}$ verification bouts, in all CPET situations remains open to question.

## Supporting information

**S1 Checklist. PRISMA 2009 checklist.**
(DOCX)

**S1 Text. Search strategy.**
(DOCX)

## Acknowledgments

We would sincerely like to thank the following authors for kindly providing additional data: Dr. Fernando Beltrami, Dr. Nicholas Beltz, Dr. Maria Cristina Bisi, Dr. Nathan Dicks, Dr. Kaitlin Freeberg, Dr. Stuart Goodall, Dr. Gianni Gnudi, Dr. Nicholas Jamnick, Dr. Theresa Mann, Dr. Lex Mauger, Dr. Max Niemeyer, Dr. Jeann Sabino-Carvalho, Dr. Brandon Sawyer, Dr. Bruno Silva, Dr. Rita Stagni, Dr. Katie Taylor, Dr. Chantal A. Vella, Dr. Ryan Weatherwax, and Dr. Eurico Wilhelm.

## Author Contributions

**Conceptualization:** Victor A. B. Costa, Adrian W. Midgley, Sean Carroll, Todd A. Astorino, Felipe A. Cunha.

**Data curation:** Victor A. B. Costa, Adrian W. Midgley, Tainah de Paula, Felipe A. Cunha.

**Formal analysis:** Victor A. B. Costa, Tainah de Paula, Felipe A. Cunha.

**Funding acquisition:** Felipe A. Cunha.

**Investigation:** Victor A. B. Costa, Felipe A. Cunha.

**Methodology:** Victor A. B. Costa, Sean Carroll, Todd A. Astorino, Felipe A. Cunha.

**Project administration:** Victor A. B. Costa, Felipe A. Cunha.

**Resources:** Felipe A. Cunha.

**Software:** Victor A. B. Costa, Tainah de Paula.

**Supervision:** Felipe A. Cunha.

**Visualization:** Felipe A. Cunha.

**Writing – original draft:** Victor A. B. Costa, Tainah de Paula, Felipe A. Cunha.

**Writing – review & editing:** Adrian W. Midgley, Sean Carroll, Todd A. Astorino, Paulo Farinatti.

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
