## [Decision Letter · Decision Letter 0]

8 Sep 2020

PONE-D-20-25408

‘Verification phase’ for confirming ‘true’ maximal oxygen uptake in apparently healthy adults: Systematic review, meta-analysis, and recommendations for best practice

PLOS ONE

Dear Dr. Cunha,

Thank you for submitting your manuscript to PLOS ONE. After careful consideration, we feel that it has merit but does not fully meet PLOS ONE’s publication criteria as it currently stands. Therefore, we invite you to submit a revised version of the manuscript that addresses the points raised during the review process. Both reviewers underlined conceptual limitations that should be adressed. The manuscript should be also shortened to help the reader to catch the main aim of this review, i.e., the use of verification phases.

We look forward to receiving your revised manuscript.

Kind regards,

Laurent Mourot

Academic Editor

PLOS ONE

Journal Requirements:

2. Please clarify if and how you assessed for publication bias. Please provide graphs supporting your analysis of publication bias. Please confirm if unpublished studies/ grey literature had been searched?

3. Please confirm whether the quality of studies was assessed by more than one person and whether there was a consensus procedure for disagreements.

4. Thank you for stating the following in the Funding Section of your manuscript:

[This study was partially supported by grants from the Carlos Chagas Filho Foundation for the Research

 Support in Rio de Janeiro (FAPERJ, E-26/202.705/2019, recipient FC; E-26/202.880/2017, recipient PF)

 and Brazilian Council for Technological and Scientific Development (CNPq, 248023/2012-8 and

Manuscript Click here to access/download;Manuscript;Manuscript.docx

303629/2019-3, recipient PF). The funders had no role in study design,  data collection and analysis,

decision to publish, or preparation of the manuscript.]

 [The funders had no role in study design, data collection and analysis, decision to publish, or preparation of the manuscript.]

Reviewers' comments:

Reviewer's Responses to Questions

**Comments to the Author**

1. Is the manuscript technically sound, and do the data support the conclusions?

Reviewer #1: Partly

Reviewer #2: Yes

2. Has the statistical analysis been performed appropriately and rigorously? 

Reviewer #1: Yes

Reviewer #2: I Don't Know

3. Have the authors made all data underlying the findings in their manuscript fully available?

Reviewer #1: Yes

Reviewer #2: No

4. Is the manuscript presented in an intelligible fashion and written in standard English?

Reviewer #1: Yes

Reviewer #2: Yes

5. Review Comments to the Author

Reviewer #1: Please see attached file for full comments as my feedback exceeded the allowed character count. I include below the introductory paragraph to my review.

This meta-analysis evaluated the validity of a verification phase to confirm the achievement of VO2max. Although the authors will notice that I am very critical of the model for reasons that I hope are clear and evident in my comments, I have to recognize that this is a very detailed analysis and that the data have the potential to make a meaningful contribution to the literature. However, I believe that the authors need to make some major adjustments in the structure of this manuscript so that the correct message is delivered. I believe that this could be solved relatively easily, but this would imply a shift in the interpretation of the physiological bases for supporting the idea of the verification phase as a valid approach to satisfy the plateau criterion for achievement of VO2max. In fact, towards the end of the manuscript, I realized that the authors fully understand the limitations of the model. However, for reasons that are difficult for me to understand at this point, they still present the model as valid to test something that it cannot test. I have provided extensive (and admittedly repetitive) feedback on this aspect throughout my review. I apologize for this, but I was trying to be as clear as possible with my words so that the chances of misunderstanding my position are minimized. I truly hope that the authors are willing to make some important changes, as I consider the data very strong and I fully respect and value the amount of work put into this analysis.

Reviewer #2: Comments to the Authors

General Comments

This study aimed to provide a systematic review and meta-analysis of the validity of deriving VO2max and HRmax from a verification phase, with a cardiopulmonary exercise test (CPET) considered the gold standard. It is a thorough investigation of the studies published within this field to date, and the data appears to have been presented objectively and in a clear manner. The research question is relevant and the analysis necessary.

One fundamental problem I have with studies discussing the attainment of a “true” VO2max (or maximal anything) via CPET and VP (or equivalent) is that we actually don’t know in any scenario whether a “true” maximal value really has been attained. This almost philosophical point (although I find it to be quite obvious and, in this case, physiological) is almost always overlooked and it has been once again in this paper. I think it would be judicious of the authors to acknowledge – and discuss in some detail – this fact (rather than just skim over it at the end of the discussion, in relation to Noakes’ [150] critique).

Another fairly major concern I have is that the analyses are limited to comparing maximal VO2 measures derived from a CPET versus a verification phase at a group level (also highlighted as an issue by Noakes [150]), which is not the same as measuring validity (i.e., through the agreement of two measures). Two measures can be similar (or not significantly different) at a group level, but the agreement can still be very poor. Without an analysis of agreement, how can validity be inferred?

Specific Comments

Title

I’m not sure why single quotation marks are needed around the selected terms. Also, “recommendations for best practice” were not scientifically investigated, but are (a small) part of the discussion/conclusion (as is the case in many papers). So I suggest removing these components of the title. This would give something more succinct and specific, like: “The verification phase for confirming maximal oxygen uptake in apparently healthy adults: A systematic review and meta-analysis”.

Abstract

On analyzing the title and aim of the study, I don’t feel this abstract is necessarily a clear summary of the most relevant findings. The results are rather heavily focused on the HR data, which is not a main focus of the study. Please reflect and re-consider.

L37: CPET on “a” cycle… (the article seems to be missing).

L41: The punctuation suggests that this is the age range (and even VO2max) of the women only. Please clarify.

L42, 44: n = 52/36… presumably these are the number of studies? Please clarify.

L44: Can (should) bpm be expressed to decimal places? Also, 5 d.p. on the P value seems quite excessive (as is the case throughout the manuscript for very small P values – is there any reason for this?).

L44-47: It is unclear what the comparisons (3 bpm greater HR) relate to for the three P values. What are they being compared to? Please clarify. That said, I’m not sure why so much focus is given to HR in the abstract, when the study is really about verifying VO2max.

L50. Why would concordance (agreement) “put [this] into question”? This seems contradictory. Please clarify.

Introduction

The Introduction is long and there is rather a lot of discussion around the general topic of VO2max testing, but sparse detail relating to the actual topic of the study (i.e., the use of verification phases in all their various make-ups). The sentence at L126-129 to me is the crux of the problem and the study, and this is what the introduction should focus on more exclusively. The study referenced in L130-131 ([53]) requires more discussion/explanation, so that there is clearer context for the current study and justification for the sentence at L131-134. I have no issue with the importance of this work, but a bit more clarity around the actual problem (and existing literature) is required.

L55-58: This is a very long opening sentence. I suggest breaking it in two (if all content is to remain).

L61: Should the refs be listed in numerical order (e.g. [1, 3-6])? (There are other examples throughout the manuscript as well.)

L69-70: A transition from that stated, but to what? Continuous fast ramp tests? It feels like the end of the sentence is missing – maybe combine (and condense?) it with some of the text from the following sentence (L70-72).

L74: Should this be “limitations of VO2max” (i.e., the method of measurement)? The limitations “to” VO2max is a different topic, as I see it, more related to training, genetics, etc.

L83-86: There seem to be two contrasting definitions of the Taylor et al. VO2 plateau criterion here. Please clarify.

L87: Should the ref ([32]) not be included with Taylor et al. even here?

L94: Typo on VO2max (the second time).

L97-99: Check and change the grammar/punctuation. Something is going on around “investigators; however, due to”, which makes the sentence incoherent (to me).

L77-104: This is a very long paragraph. I suggest splitting it in two – the first relating to VO2max criteria, the second to the secondary criteria. Or write the content more concisely to produce one shorter paragraph.

L105-139: These two paragraphs are most important in justifying the current study, so I think a more in-depth discussion of this literature is required (instead of the level of detail presented in the three preceding paragraphs, which could be significantly condensed).

Methods

L162: “ergometer or treadmill” – was this limited to bi-pedal running on a treadmill? Please specify, as many other modes of exercise are possible on a treadmill (e.g., cycling, hand-cycling, wheelchair running, inline skating, roller skiing, etc.). It would be useful to make this clarification through a clear definition somewhere in the paper, that by “treadmill” (see for example Figure 1, “Only treadmill”) you actually mean “treadmill running” (if that’s the case).

L171-172: “In the final review, we provided…” – is this referring to what is presented in the current manuscript? Please clarify, as “final review” and “we provided” is a bit unclear to me.

L196: You have previously written abbreviations within round brackets in square brackets: (95% confidence interval [CI]).

L197: Out of interest, what did you do in cases where VO2max was reported in mL/kg/min?

L210: P-values “were” obtained…?

L215: Is this less than 50%, or less than or equal to? The symbol looks unclear to me.

L223: The studies were also…

L225: Stratified analyses were also…

Results

The main issue for me throughout this section is the long lists of references accompanying each result. This is not common in other studies of this type that I’m familiar with, and to me it makes deciphering the interesting information nigh on impossible. I would recommend removing these long number strings.

L241: (interquartile range [IQR]) – or consistent with previous presentation.

L240-245: You write that “the sex of 130 participants was not specified”, and then that “one study did not specify the sex of the participants (see Table 1)”. In that study (Scheadler and Devor [92]) n = 13, so I don’t understand the mismatch between 130 and 13. Please clarify.

L246: BMI should presumably be defined after the words (body mass index) and doesn’t then need to be included in the brackets.

L246-247: The square and round brackets seem to have switched places in this sentence, any reason?

L247: Writing “(VO2max normalized to body mass)” seems superfluous when you have the unit as mL/kg/min. Consider removing.

L251: “Characteristics of studies using CPET…”?

L253, 255: “on a cycle/treadmill” (again, the article is missing).

L253-307: These long strings of references make the results very unreadable. I suggest removing them all, as finding the actual interesting numbers (i.e., the results) through the long lists is so difficult.

L272-274: Could you re-phrase to fix the grammar and clarity on: “whereas 29 (37%) used fixed intervals of 15- to 30-s (or 2 × 15-s), both averaged and fixed times (1%) [61]… etc.”. I guess the 29 (37%) relates specifically to the 15/30-s fixed interval data, so the sentence needs to be re-structured and improved to clarify this in relation to the other methods listed.

L279, 281: I think you need to include the “min” unit after 5, 6, 6, 9 and 15 – or not if you were to remove all the references in brackets (another example of how difficult the interesting numbers and results are to decipher from the long [and unnecessary?] lists of references).

L285-286: Why are two different %ages presented (19 and 19.7)?

L291: Suggest removing “i.e.”? Not included elsewhere.

L295: Should this be “the” maximal-intensity work rate, rather than “a” (presumably it was specific to that study and the preceding VO2max test).

L297: Could you briefly describe in this sentence what the formula was based on?

L300: “Forty-two studies (54%)” – consistent reporting.

L300: obtained “during” (rather than “at”)?

L366-onwards: Is there a reason for changing the presentation (order of using) round and square brackets again? And see my point above (in the Abstract) about the number of decimal places on the P values < 0.001. Is there any statistical reason/need for this?

L383: (performed on the same day as vs. a different day from the CPET) – suggestion.

L387-388: Could you include the P value here for this no sig diff, as it is a key result.

Discussion

At times I struggle to follow the logic of the arguments in this (very long) discussion, so I think the interpretations can be written more clearly and concisely in places. In addition, there is a lot of discussion of previously published studies and concepts, without reference to the findings from the current results. This seems inappropriate for a systematic review/meta-analysis, so I would encourage the authors to focus more on their own findings in light of previous work, rather than merely presenting a review of the existing literature.

L422: Reconsider “over” in this sentence. Maybe “rather than”?

L433-435: This study did not analyse children or clinical groups, so where is the “current evidence suggest[ing] that the verification phase is a safe and well-tolerated procedure to confirm attainment of true VO2max” in these groups? This particular study can surely only make this claim about the apparently healthy adults who were analysed, or am I missing something?

L445: “of a ramp-incremented…” (missing article).

L455-456: Are 17% and 33% comparable in this sentence? If so, please use the same unit (either CPETs, or participants) in order to compare like with like (e.g., “17% of participants (2 of the 12) during a cycling ramp-incremented CPET, while 33%…”.

L477-478: Is this statement true? If so then I’m missing something. Re-reading ref #100 (McGawley 2017) it is stated that: “There was a signiﬁcant eﬀect of test type on VO2max, with higher values recorded during STEP compared with VER (P = 0.013)”. Can you clarify how you’ve come to this conclusion (three studies to-date) and how you conducted this analysis/check?

L494: Is there a typo here: “4 of the 7 participants (9%)”?

L496: “11 participants (9 men; age…)” – it appears as if you are only reporting the descriptives for the men, is this the case? Please clarify.

L499-503: From your results and Figure 2 this looks like an outlier. Has/should there been any accounting for outliers in your analyses?

L477-516: This is a very long paragraph (> 1 page). Please consider shortening. I don’t think all the detail of the three specified studies is required (L478-503) – this could be condensed and written more concisely.

L508-516: This seems to explain this result as an outlier. What happens to your findings (CPET vs verification phase VO2max) if this study is removed from your overall analyses?

L517-519: I’m not sure I agree with this statement (or maybe I misunderstand what you mean). If CPET = VP or if CPET > VP then is that confirmation of a “true” VO2max during the CPET? Can both tests not elicit a VO2max that is lower than an individual’s “true” VO2max in this scenario?

L520-521: I don’t quite follow the logic of this follow-up sentence. Are you saying that there would need to be a difference in order for the statement in the first sentence to be true? Why? Please clarify.

L526: Why “only” 25 (i.e. 27%)? To me this 27% of the studies is important in demonstrating that the CPET doesn’t always do its job properly (i.e., in eliciting a “true” VO2max). This is where the analysis of agreement is important too – what is the similarity (or dissimilarity) in VO2max values derived from a CPET vs VP “within” individuals? Please comment.

L517-528: I actual don’t follow the logic or point being made in this section. Could you please try to clarify?

L542: “who” underwent?

L544: was similar “to the”?

L547: At this point I’m really struggling to follow the logic and arguments presented over the last few pages. Are you saying that CPET should be higher than VP in order to accept that a true VO2max has been attained in the CPET? Why? What is the problem with CPET = VP? A more fundamental question, in my opinion: Why is it not acknowledged/discussed that individuals can very easily underperform on BOTH tests, and that we really don’t have any idea as to whether we have attained a “true” VO2max at all. Please comment.

L552-557: The “different” methods described previously for study #94 are also relevant here, as is my comment above (i.e., that it is always possible that neither test was truly maximal and elicited a “true” VO2max).

L557: What is meant by “this put into question” in this context?

L561-562: This is the first time this endeavor has been mentioned (except in the title). Please re-consider the phrasing here (and in the title!) – especially given the conclusion of this sentence (L567-569), i.e., that no best practice can actually be recommended.

L570-571: This list of 6 references does not seem complete, or to reflect “most studies”. Please clarify.

L570-578: What new insight does this paragraph add, from the current results, which was not already known? Please embellish with additional information, or remove.

L584-585: I don’t think decimals are needed on these %ages.

L579-596: Again, how do the current results relate to the previous literature? This is not a review article, so as I see it the discussion section should be used to present the results of the current study in the context of previous results. The information presented here (that 105% was different from 115% according to Nolan et al.) is not supported by your results, as I understand, since you saw no significant effect of VP intensity. This is what ought to be discussed, in my opinion.

L597-610: Again, this is a review of the existing literature. Please discuss the results of the present study.

L613: Remove the extra space(s) between Small and sampling.

L614: rapid changes

L611-621: Same issue again - this is a review of the existing literature without reference to the current study results. Please reconsider.

L624: should not exceed

L636: on the duration of

L642: when “what” are short? The VPs? Please clarify.

L649-653: I don’t think the second sentence is a good enough “get-out” given the significance of the criticism stated by Noakes. This underpins the entire concept of “validity”. Do not overlook or underplay the fact that your final sentence, which you say you are not doing (“rather than the question of whether an individual has elicited a ‘true’ VO2max.”), is exactly what you say you are doing in the title (as I interpret things)!

L655: Have effect sizes been presented anywhere?

L656-657: “in cycle ergometry and treadmill running”?

L660: “compromising their ability” (plural)

L670: I don’t understand this, in the context of the previous sentences: “The mandatory application of the verification phase in all situations may be therefore questioned”. Why questioned?

L671: settings?

Tables & Figures

The studies appear to have been ordered chronologically and then alphabetically in Table 1, with this ordering system then continued throughout the later tables & figures. This seems arbitrary (chronologically then alphabetically) and makes it difficult to locate any specific study in the later tables/figures. Could you order the studies entirely alphabetically or according to the reference numbering from the outset?

Is there any reason for presenting the subgroup analyses according to the characteristics of the verification phase protocol in a “figure”, while the subgroup analyses regarding sex, cardiorespiratory fitness level, exercise modality, and CPET protocol are presented in a “table”? Could this method of presentation be standardized? Also, I’m not familiar with the presentation used in Figure 4. Can you provide more information about how to read it (top line with green box, middle line with green box and black diamond), as it won’t be clear to all readers.

Table 1: The heading “mean values” should probably be aligned over the final three columns to the right, as sex and N are not means. Also, ranges should be differentiated in this heading, if that’s what those are and if they can’t be expressed as means (e.g., 25-35 and 19-61). And can/should the number of decimal places be standardized in the data? Any reason that some terms (e.g., Sedentary, Cyclists, Runners, Athletes) are capitalized, but others aren’t?

Tables 3/4: Can Total be clarified (presumably it’s the number of participants, but this is not stated anywhere). The %Weight is hard to comprehend – I have no experience of this measure or its calculation, but the statistical power seems to bear no relation to N, which seems odd to me. Can you explain?

Table 5: Can horizontal lines be used to clarify where each category (TTE, VO2max, HR) starts and ends (i.e., to the right of each N)?

Figure 1: Can you clarify (even if just to me) why the 1 full article excluded in the Eligibility stage due to “Non-maximal exercise test protocols…” had not already been excluded for the same reason in the Screening stage?

Figure 2: The data suggests to me a tendency for CPET to be higher than VP. Is there any accounting for potential outliers (e.g., Colakoglu et al.)? What happens if this study is removed from the analyses (if there is good reason to do so, which reading the discussion there might be)?

Figures 2-4: The quality of these figures is poor (due to the high level of detail). Can they be presented at a higher resolution?

Reference list

L684: (1985) should presumably be removed from the JAP title?

6. PLOS authors have the option to publish the peer review history of their article (what does this mean?). If published, this will include your full peer review and any attached files.

Reviewer #1: No

Reviewer #2: **Yes: **Kerry McGawley, Ph.D.

---

## [Author Response · Author response to Decision Letter 0]

7 Nov 2020

- Title: Verification phase for confirming maximal oxygen uptake in apparently healthy adults: A systematic review and meta-analysis

- Corresponding Author: Felipe A. Cunha 

- E-mail: felipeac@globo.com

- Manuscript ID: PONE-D-20-25408

Dear Editor,

Please find below our responses to the reviewers’ comments concerning the article PONE-D-20-25408, titled “Verification phase for confirming maximal oxygen uptake in apparently healthy adults: A systematic review and meta-analysis”. The manuscript has been revised according to the reviewers’ suggestions and an itemized, point-by-point response to each of the reviewers’ comments has been provided. 

Yours Sincerely,

Felipe A. Cunha

Review comments:

Reviewer: 1

This meta -analysis evaluated the validity of a verification phase to confirm the achievement of VO2max. Although the authors will notice that I am very critical of the model for reasons that I hope are clear and evident in my comments, I have to recognize that this is a very detailed analysis and that the data have the potential to make a meaningful contribution to the literature. However, I believe that the authors need to make some major adjustments in the structure of this manuscript so that the correct message is delivered. I believe that this could be solved relatively easily, but this would imply a shift in the interpretation of physiological bases for supporting the idea of the verification phase as a valid approach to satisfy the plateau criterion for achievement of VO2max. In fact, towards the end of the manuscript, I realized that the authors fully understand the limitations of the model. However, for reasons that are difficult for me to understand at this point, they still present the model as valid to test something that it cannot test. I have provided extensive (and admittedly repetitive) feedback on this aspect throughout my review. I apologize for this, but I was trying to be as clear as possible with my words so that the chances of misunderstanding my position are minimized. I truly hope that the authors are willing to make some important changes, as I consider the data very strong and I fully respect and value the amount of work put into this analysis.

Introduction

Lines 55-58: This sentence is too long and difficult to read as presented. For example, I do not think that the word “neuromuscular” is needed here. It is obvious that neuromuscular function is important for performance and it will impact vascular function to a given extent. However, VO2max is not defined by neuromuscular performance in my view. Regardless, the authors should shorten this sentence.

ANSWER: Thank you for the observation. We have reviewed the paragraph according to the reviewer’s suggestion and it now reads as follows: “Maximal oxygen uptake (VO2max) represents the upper physiological limit of the utilization of oxygen for producing energy during strenuous exercise performed until volitional exhaustion”.

Line 59: It is interesting that the definition of VO2max included cardiovascular, pulmonary, neuromuscular, and metabolic responses, but the authors refer to the test as cardiopulmonary (CPET). I never understood this term, as it excludes the vascular component, which is a key component of O2 distribution. I do not think that I am asking for this to be changed, but I wanted to highlight that I do not like it.

ANSWER: Since the definition of VO2max has been revised to address your previous comment, we believe the revision also addresses this comment.

Lines 70-72: It is surprising that the work of Iannetta et al. (Am J Physiol Regul Integr Comp Physiol. 2020 Jul 22. doi: 10.1152/ajpregu.00126.2020; J Appl Physiol. 2019 Dec 1;127(6):1519-1527) and Keir et al. (Appl Physiol Nutr Metab. 2018 Sep;43(9):882-892) is not mentioned here, as it highlights important aspects of ramp incremental tests that are often ignored and that have resulted in misinterpretations of: 1) the link between ramp incremental and constant-load VO2 and work rates; 2) the lack of validity in the idea of using a supra peak work rate intensity for the verification of the VO2max response (more on this will certainly arrive in later comments).

ANSWER: Apologies for missing this. We have read all the articles to address the reviewer’s suggestions. The aforementioned studies are now included in the revised manuscript and also meta-analyzed (e.g. Iannetta et al. Am J Physiol Regul Integr Comp Physiol. 2020 Jul 22. doi: 10.1152/ajpregu.00126.2020).

Lines 74: It remains surprising that the authors discuss that more needs to be learned about using information from ramp incremental tests and have not even mentioned the manuscripts indicated above (I understand that one of them might be too recent, but I find the other omissions surprising).

ANSWER: Please see our response to your last comment.

Lines 75- 76: This is true. For example, the supra peak work rate intensities that are often used represent a good example of what could be a wrong protocol. Additionally, if motivation is an issue, then the verification ride would not fix this problem!

ANSWER: We are in agreement with the reviewer’s opinion. Thank you for your comment.

Lines 77-104: I think the authors do a good job in this paragraph identifying important issues. However, this paragraph is a bit redundant and convoluted. This section could be shortened by ~30% to streamline the message. 

ANSWER: This paragraph (now starting on line 72) has been shortened according to the reviewer’s suggestion.

Line 94: Second VO2max is missing the letter “x”.

ANSWER: The correction has been made.

Lines 105-129: Well, there are several conceptual limitations in this section. First, I will start by saying that a wrong concept repeated by many people, remains a wrong concept (even when some of the people supporting the concept are dominant figures in the field). Let’s start by saying that, by definition, there is no such thing as supramaximal. In the end, the work rate obtained at the end of a ramp incremental test is far from maximal. It is simply a peak value, that will be greater or smaller depending on the characteristics of the ramp. In fact, it has been shown that whereas VO2max (or peak if you prefer) remains constant across a wide range of slope during a ramp incremental test, the power output (PO) varies widely depending on the slope of the ramp. For example, Iannetta et al. (J Appl Physiol. 2019 Dec 1;127(6):1519-1527) showed that whereas steeper ramps elicit the greatest peakPOs, less steep ramps result in progressively lower peak POs. This demonstrates that the so-called “supramaximal” work rates are just an illusion. “Supramaximal” in relation to what? This has been further indicated in a study that showed that performing sub peak work rate exercise to exhaustion is more likely to result in the achievement of higher VO2 values than performing supra peak work rate exercise (Am J Physiol Regul Integr Comp Physiol. 2020 Jul 22. doi: 10.1152/ajpregu.00126.2020). Although this might sound counter intuitive to the naïve reader, it should be physiologically expected by anyone who understands how ramp incremental testing affects the development of the slow component of VO2. See Appl Physiol Nutr Metab. 2018 Sep;43(9):882-892 and Med Sci Sports Exerc. 2020 Mar 20. doi: 10.1249/MSS.0000000000002343 for more details.

ANSWER: We have added these references to the paper and the so-called “supramaximal” term has been changed to “supra peak” work rate throughout the manuscript in accordance with the reviewer’s comments (the revised paragraph starts on line 98).

Second, and based on this comment, I will add that: 1) the statement that “a continuous CPET followed by an appended supramaximal verification phase is conceptually similar to the discontinuous tests most commonly used from the 1920s to the 1970s, but with the notable advantage of requiring only a single visit to the laboratory” is simply wrong. I would highly recommend that the authors read the papers indicated above to fully appreciate the idea that there is no such thing as a “supramaximal” work rate; 2) it is surprising that experts in VO2 kinetics such as Poole and Jones have made the mistake or recommending that “supramaximal” intensities should be used for the verification rides. As indicated in the papers highlighted above, 110% of a typical ramp test (or even 1-min step) might not allow for enough time for the VO2 response to be fully expressed, thus resulting in VO2 values that are lower than those obtained during the ramp incremental test, and creating the false idea that VO2max has been achieved when this cannot be demonstrated.

ANSWER: The statement cited above by the reviewer in double inverted commas has been removed from the revised manuscript. We have addressed the issue of using the term ‘supramaximal’ in a previous comment.

Lines 134-137: I would argue that a recent paper (Am J Physiol Regul Integr Comp Physiol. 2020 Jul 22. doi: 10.1152/ajpregu.00126.2020) has demonstrated that the verification phase is a flawed approach to confirm that VO2max has been achieved. In fact, this paper has indicated what type of verification ride would be most adequate in order to achieve the highest possible VO2 (which most likely represents VO2max, as derived from the ramp incremental test!)

ANSWER: This part of the “Introduction” section has been rewritten to address the reviewer’s suggestions (please see text in red).

Lines 137-139: Unfortunately, this meta-analysis cannot achieve this goal. All the authors can do is to provide information on what others have obtained from a ramp or step test and compare those values with the verification rides. Given some of the comments that I made before, the authors should understand that the idea of a “supramaximal” work rate to determine the plateau criterion is flawed. This is not something that is up to debate. It is just a fact based on published data. Then, given the limitation of the model and the inability to establish a plateau response as originally proposed, the verification ride is simply an extra ride that might make people feel more confident with the idea that the highest possible VO2 has been achieved. From a personal perspective, I have no issues with people doing this. However, the vast majority of the data indicate no significant differences between the highest VO2 from the ramp incremental test and the verification ride. When differences exist, those typically fall within the measurement error and on both sides of the probability (i.e., VO2 could be higher in either the ramp or the verification).

ANSWER: Thank you for the observation and we agree with the reviewer. The aim of the study has been rewritten to address the reviewer’s comment (please see text in red starting from line 126).

As a summary of this section, I think that this meta-analysis, as proposed, is set to fail as it cannot answer the question that it is proposed to answer. My recommendation would be that the authors simply try to answer whether there are differences in the highest VO2 values obtained from ramp or step tests compared to a verification ride. Even if differences existed, there is no way that the authors could claim that VO2max has been verified as the “supramaximal” intensities are rather arbitrary and often ignore the most basic physiological responses for VO2 adjustments. In my view, the introduction should describe the limitations with the idea that VO2max can be verified (i.e., the “supramaximal” concept is flawed and “submaximal” verification phases has been shown to work as well), while highlighting that there might still be value in trying to confirm that the highest possible VO2 was achieved, which can be done in a variety of ways (as demonstrated later in this study). However, I find it critical that the erroneous idea that VO2max is verified because a “supramaximal” intensity was added to the evaluation is not perpetuated.

ANSWER: The research questions and study aim at the end of the introduction section have been revised to address the reviewer’s concerns. Thank you for your comment.

Methods

General comment: Although I have participated in meta-analysis studies and reviewed some of them, I have to admit that statistical analysis is not my area of expertise. Thus, even though the information seems correct, I will rely on others for proper evaluation of this section.

ANSWER: Comment acknowledged.

Results

Line 291: Here and throughout the text, I think it would be more appropriate to refer to the intensity as “supra peak work rate” or something like that, so that it is clear that maximal work rate has not been determined in any of those studies. The same idea would apply to the term “submaximal”.

ANSWER: The manuscript has been revised throughout in accordance with the reviewer’s recommendation.

Line 298: I am happy to see that the authors used the right term here and referred to “peak work rate”.

ANSWER: Comment acknowledged.

Line 301: By design, this so-called verification phase cannot confirm the attainment of VO2max. As discussed earlier, the VO2max response remains a mystery. From my perspective, I would argue that VO2max was most likely achieved. However, given that the model used in many studies does not even allow for the VO2 response to be fully expressed, then this method risks falsely accepting the attainment of VO2max simply because the “supramaximal” intensity was too high to allow for the duration of exercise to be long enough for the VO2 response to be fully expressed (sorry for the extra-long sentence!). I would strongly recommend that the authors stay away from the idea of confirming that VO2max was attained as this is an untestable hypothesis. However, the authors can try to verify whether the VO2 associated with the “verification” ride was lower/higher than, or similar to, the VO2 observed during the incremental test.

ANSWER: We agree with the reviewer’s opinion and any statements about the verification phase “confirming that VO2max was attained” have been removed from the revised manuscript.

Line 326: I like the wording here, as the author refer to the highest VO2. There might be value in knowing whether the verification phase results in a greater VO2 compared to the incremental test. However, as I indicated earlier, it is important not to confuse that with determination of VO2max.

ANSWER: Thank you for your comment.

General comment: I think that a further analysis that could be added would include the duration of the verification phases. I guess it will not change much as the mean data will hide the potential effect of duration. However, it is likely worth exploring whether shorter verification phases resulted in comparatively lower values.

ANSWER: We have included the further analysis according to the reviewer’s suggestion. 

Discussion

Line 417: As I indicated already, the verification phase cannot confirm the achievement of VO2max as the use of this approach is physiologically flawed. Thus, this meta-analysis should re-focus its aim and indicate a testable objective (i.e., identifying whether the VO2 during the verification phase is different from that obtained during the incremental test).

ANSWER: The aim stated in the introduction section and the discussion section have been revision to address the reviewer’s suggestions (please see text in red).

Lines 428-432: This section seems almost out of place or, at least, unnecessarily long. I understand that the authors might want to highlight the safety of the procedure in other populations. However, I would argue that the point of safety can be made in relation to the current data, and then the authors could add that several studies in clinical populations also confirm the safety of the procedure.

ANSWER: The reviewer is correct, thank you for the comment. The relevant sentence has been moved to the last paragraph of the revised discussion section.

Line 434- 435: Well, I guess my position on this wording is already clear to the authors. However, I will repeat it. This statement is unwarranted as no confirmation of VO2max has been obtained. Such confirmation requires a different experimental model as the intensity used in most verification phases are: 1) often too high for allowing a full development of the VO2 response; 2) below or at the peak PO, which invalidates the premise of “supramaximal” intensity to achieve the plateau criterion. In the end, the model helps you feeling more comfortable with the idea that the performance was maximal. However, the same limitations that applied to not feeling comfortable with a maximal effort during the incremental test also apply during the verification phase. Considering this and the idea that a plateau response is just an illusion (i.e., constant work rate loads become “supramaximal” much earlier than the peak PO observed during incremental testing), then the wording needs to be changed. Thus, the authors could say that “the verification phase is a safe and well-tolerated procedure to evaluate if a higher VO2 compared to that observed during the incremental test was obtained”, or something in line with that.

ANSWER: We have revised the text in accordance with the reviewer’s suggestion. The revised text is located at the end of the second paragraph of the discussion section.

Lines 441- 443: These lines include some information that is correct, and some that sound wrong (at least as presented). It is true that incremental tests do not allow for the VO2 slow component to be fully expressed (with this being more noticeable with steeper ramps), which dissociates the relationship between VO2 and PO when comparing incremental and constant work rate exercise. However, the authors mentioned that the VO2 response is accelerated towards the end of the incremental test. This is incorrect. In fact, during the most used ramps (i.e., 20-30 W/min), the increase in VO2 for a given increase in PO is reduced towards the end of the ramp. This is reflective of the inability of the VO2 response to adjust fast enough to the rapidly increasing ramp. In fact, what the authors mentioned is observed during slow ramps (i.e., 5-10 W/min). This can be seen in Iannetta el al. (J Appl Physiol. 2019 Dec 1;127(6):1519-1527). Interestingly, that paper showed consistent VO2max responses with a wide range of peak POs, and a follow up paper showed that POs do not need to be “supramaximal” to confirm that the highest VO2 has been obtained (Am J Physiol Regul Integr Comp Physiol. 2020 Jul 22. doi: 10.1152/ajpregu.00126.2020). In fact, “supramaximal” intensities are discouraged. Importantly, the current data confirm that the incremental test is sufficient to obtain the highest possible VO2, which demonstrates that a verification phase does not offer much additional value.

ANSWER: We agree with the reviewer’s comment and the sentence has been rewritten to address this concern. 

Lines 444-449: This is perfectly in line with what I had just indicated. Given the wide range of ages and fitness levels, and the variety of ramps (i.e., 15, 20, and 25 W/min), the phenomenon that I just described is to be expected. In other words, slow ramps are far more likely to result in a plateau in the VO2 response as the slow component can be expressed to a larger extent, even within a narrower range of POs, as compared to fast ramps.

ANSWER: Thank you for your comment.

Lines 449-450: Exactly! If you think about it, the same ramp would be a relatively faster ramp for a less fit person compared to a fitter one (i.e., less time for the VO2 response to fully adjust).

ANSWER: Thank you for your comment.

Lines 465-476: This section could be just deleted as, aside from the use of HR responses, no other secondary criteria were considered in this analysis. Given that the discussion is excessively long, I would remove this section.

ANSWER: The section has been deleted in accordance with the reviewer’s suggestion.

Lines 491-497: In fact, this information is in line with a recent paper from DiMenna’s group (Arad et al., PLoS One. 2020 Jul 6;15(7):e0235567.doi: 10.1371/journal.pone.0235567). In this study (which should also be added to the analysis in my view), they showed that the verification phase resulted in a significantly (albeit minimally) greater VO2 compared to the incremental response. Interestingly, the authors commented that they selected 100% of peak PO to make sure that the duration of the verification phase was not too short. Based on the definition by Poole and Jones, the authors say that they cannot claim that VO2max was achieved because the intensity was not “supramaximal”. However, they know that this idea of “supramaximal” is flawed anyway. Regardless, it is likely that a verification phase can help achieving a higher VO2 compared to the ramp test in some very specific circumstances. However, there is still no proof that VO2max has been achieved!

ANSWER: We have added this reference to the manuscript and also meta-analyzed their data to in response to the reviewer’s comments.

Lines 499-516: Well, this section is difficult to interpret as the results from Colakoglu’s work are suspicious. How can you reconcile the results presented in Figure 2 in that study? I would be the first to admit (and I have made this point already) that 110% of the peak PO might result in a short duration for the verification ride, which might not allow for the VO2 response to be fully expressed. However, a near 20% drop in the VO2 response during the 110% verification ride is unthinkable! In fact, I went to the original study as the difference between the incremental test VO2 and the 100% verification ride VO2 was extremely high. After seeing the even greater difference at 110% of peak PO, I simply cannot trust those data (which is likely the reason why the paper is published in a very low impact journal). Regardless, the point is that the data from this study should probably not be considered. However, we are not the judges of science and I think it is fair to include the published results. That being said, a discussion on the surprising ~30% difference between the highest and lowest VO2 should be presented.

ANSWER: Although we agree with the reviewer’s concerns, we understand that that it was not appropriate to omit these data from our analyses and should allow the reader to draw their own conclusions about the cited study. Notably, after omitting the data from Colakoglu’s study, there was no change in the main results of the present meta-analysis.

Lines 517-519: This should be reworded. What effectiveness has been demonstrated? In the end, the verification phase did not produce any significant differences in the results. Thus, rather than effective, I would consider this a waste of time. Based on your data, the recommendation should be that a verification phase is not included as it does not add any value to what was already informed by the incremental test. At best, you can say that “the verification phase procedure did not result in a higher VO2 response than that observed in continuous ramp or pseudo-ramp CPET protocols”. Perhaps, you might want to feel that this adds validity to the procedure, and I can accept that as a personal preference. However, by no means can the authors say that VO2max was confirmed.

ANSWER: The sentence has been changed and now reads as follows: “To-date, this systematic review has demonstrated that the majority of studies have shown that the highest mean VO2 values elicited by verification phase bouts were similar (or not statistically or meaningfully different) to the attained VO2 values in continuous ramp or pseudo-ramp CPET protocols [104, 114, 36, 125, 97, 37, 77, 81, 38, 60, 96, 39, 57, 102, 101, 63, 86, 88, 119, 61, 69, 73, 79, 99, 64, 65, 94, 113, 124, 126, 103, 107, 115, 53, 66, 75, 76, 112, 67, 80, 85, 117, 123, 56, 72, 84, 87, 91, 116, 118, 120, 59, 71, 74, 78, 83, 89, 90, 105, 108, 109, 15, 121, 70, 106].” (Page 35, lines 358-363).

Lines 520-521: Why do the authors in theory agree with this premise? The premise of a “supramaximal” verification phase as a means of establishing the plateau criterion is physiological flawed. I hope that the authors can appreciate that now. In fact, this pre-conception sounds as a bias that pushes the authors to try to accept a perceived reality, even though the data that they presented showed the opposite. I honestly think that the authors are to be commended for the detailed analysis that they did, and I truly think that the data are useful. I just cannot understand the origin of the authors’ convictions on this topic.

ANSWER: The ‘Discussion’ section has been rewritten in order to avoid the premise of a “supramaximal” verification phase as a means of establishing the plateau criterion.

Line 529: Here, the authors refer to “the utility of the verification phase”. I believe this wording, although vague, is the closest you can get to reality. Anything that links the verification phase to a confirmation of VO2max is not scientifically verifiable using this model. Thus, this goes in line with what I have been saying from the beginning of my comments.

ANSWER: We have changed this sentence to keep coherence with the revised manuscript and now it reads as follows: “In addition, the present findings also provide consistent and unbiased confirmatory evidence that the reproducibility of the highest VO2 value during CPET and verification phase does not appear to be affected by sex, cardiorespiratory fitness, exercise modality, CPET protocol design, or even how the verification phase was performed (see Table 4 and Figure 4).”

Lines 532-534: This is simply incorrect. I mean, that your results coincide with those of Poole and Jones is not. What is incorrect is that the verification phase provides evidence that VO2max was achieved.

ANSWER: This sentence has been deleted to address the reviewer’s concerns.

Lines 535-540: As the authors might suspect, I do not agree with this overall statement. Any reference to this approach as a confirmation of VO2max should be avoided. It can be said that this was the goal of the approach, but that this goal cannot be achieved with this model. Referring to it as the “gold-standard” approach is simply ridiculous.

ANSWER: This sentence has been deleted to address the reviewer’s concern.

Lines 545- 547: I would argue that the authors in that study should not limit their interpretation to older, less experienced or unfit participants. I mean, if the need for a verification phase is often purported to be connected to the idea that some participants might not be willing to complete a maximal effort during the incremental test, the same argument could be used as a reason to suspect that a maximal effort would not be performed during the verification phase. In other words, why would someone who is unwilling to push hard during the incremental test would suddenly become willing to push hard during the verification phase. In my view, this is another reason why thinking that the verification phase confirms VO2max is erroneous. First, the idea of a “supramaximal” intensity is flawed. Second, a verification phase cannot ensure that a supposedly absent maximal effort during the incremental test is now performed during the verification phase. In fact, if a suboptimal performance occurs during the verification phase, many would erroneously interpret that as proof that VO2max was achieved!

ANSWER: This sentence has been deleted to address both of the reviewer’s suggestions, focusing more on our own findings.

Line 551: This is correct and, as indicated above, the results from that study are highly irregular. I would attribute the differences to poor measurement rather than to physiological variability. A ~30% difference between the two extreme conditions is simply unacceptable.

ANSWER: Thank you for your comment.

Lines 551-557: Perhaps, neither “a)” nor “b)” are the correct interpretation. What your data clearly show (and I commend the author for doing an impressive work with this), is that the highest VO2 values during the incremental test is not different from the highest VO2 value during the verification phase. In neither case achievement of VO2max can be confirmed. If you ask for my personal opinion, I would argue that VO2max was achieved, and that the verification phase helped demonstrating that a supra critical intensity bout to exhaustion during the verification phase resulted in the same highest VO2 value as seen during the incremental test. In that sense, I am confident that the verification phase, although unnecessary in this population, adds confidence to the measure. However, given the reasons explained before, the plateau criterion based on a “supramaximal” intensity is flawed, especially when the intensity for the verification phase is too high for the VO2 response to be fully expressed (which would lead to the wrong interpretation that VO2max has been confirmed).

ANSWER: This sentence has been deleted to address the reviewer’s concern.

Lines 558-560: I would argue that this research has been done (Am J Physiol Regul Integr Comp Physiol. 2020 Jul 22. doi: 10.1152/ajpregu.00126.2020) and that it should be incorporated into this manuscript.

ANSWER: The reviewer is right, thank you for the observation. The suggested reference has been incorporated into the revised manuscript. 

Lines 563-564: As I mentioned earlier, the terms “supramaximal” and “submaximal” should be changed. In fact, defining the terminology would be important in relation to some of the concepts that I discussed before.

ANSWER: The terms “supramaximal” and “submaximal” has been changed to “supra peak WR” and “sub peak WR”, respectively, throughout the manuscript.

Lines 561-569: I like this paragraph because it highlights that any effort to exhaustion that is high enough above the critical intensity of exercise will result in the achievement of VO2max. In fact, if someone’s goal was just to measure VO2max, any “aggressive” constant work rate to exhaustion should be good enough (however, the goal of the test is often more ambitious than just evaluating VO2max). What I do not like though is the final sentence. I do not think that the authors can recommend that “procedures that are within the scope of the reviewed studies” should be used. As I mentioned, it sounds to me as if anything above the critical intensity would do it. The problem is that people do not normally do a verification phase at 75% of peak PO from a 30 W/min ramp! Thus, the recommendation is not easy to justify.

ANSWER: The sentence has been revised and now reads as follows: “Considering that differences in the verification procedure itself do not appear to influence the utility of the procedure, a specific verification procedure cannot be currently recommended. However, some caution must be exercised in the application of WR for the verification phase, to avoid an insufficient protocol duration that does not allow the highest possible VO2 achievement as compared to CPET.” (Page 39, Lines 491-494)

Lines 570-576: I guess I do not need to say again why this is not correct.

ANSWER: This sentence has been changed and now reads as follows: “In this sense, most studies reviewed used verification phase protocols incorporating WRs above 100% of the peak WR achieved in the CPET [114, 97, 77, 81, 98, 38, 96, 39, 57, 54, 63, 111, 119, 61, 69, 99, 94, 124, 126, 92, 107, 115, 53, 75, 76, 82, 93, 112, 80, 117, 122, 123, 95, 110, 56, 58, 62, 87, 120, 59, 71, 74, 78, 83, 89, 90, 15, 121, 70, 25]. However, peak WR utilised in verification phases has varied between 85% [100] and 130% peak WR [81]. According to Poole and Jones [2], researchers must select a WR that is sufficiently higher than that attained on the CPET to give the VO2 signal for the higher WR the opportunity to emerge from the extant noise. In the event that the subsequent verification phase produces a VO2 plateau signifying VO2max, this signal would be lower than expected for the WR based on the previous VO2-WR slope. Hence, Poole and Jones [2] recommended that the verification phase should apply ~110% of the WRpeak attained in the CPET. The authors recognized that this WR may not be ideal for testing all participants, groups or circumstances. On the other hand, Iannetta et al. [25] advocated the adoption of sub peak WR verification bouts in order to allow the VO2max attainment, since WR above the critical power should result in VO2max, as long as the time to exhaustion is sufficiently prolonged.”

Lines 576-578: This is great. Then, almost everything written in this manuscript and most of my justifications were unnecessary! If the authors accept this fact, and thus accept that “supramaximal” efforts are not necessary, then they should have presented this idea from the beginning (even in the introduction), as it simply demonstrates that the concept of the “supramaximal” effort satisfying the plateau criterion is nonsense!

ANSWER: Thank you for your comment.

Lines 585- 589: Brilliant! I agree with this 100%. It is disappointing though that the authors are saving the valuable physiological information for last! Seriously, I think that, in the last two paragraphs, the authors have debunked the “supramaximal” work rate theory. I think that this meta-analysis should not only present data, but also discuss the topic in a more physiologically relevant manner. This important section comes just towards the end of an excessively long discussion.

ANSWER: Thank you for your comment.

Lines 622-632: I think this is all good for discussion. However, in line with previous comments, these different criteria would be useful, in this model, to determine whether the highest VO2 from the incremental test is different from the verification phase beyond normal variability and/or measurement error. Once again, the verification phase approach cannot determine that VO2max has been achieved as its premise is flawed.

ANSWER: Thank you for your comment. The correction has been made and now reads as follows: “A final issue to be addressed refers to appropriate criteria to accept that the highest possible VO2 has been achieved. The most commonly used criterion in the presently reviewed studies stated that the highest VO2 observed in the verification phase should not exceed 3% of the highest VO2 obtained in the CPET. This threshold can be justified by the technical error of measurement and intra-individual biological variation observed in the VO2max attainment [57, 63, 86, 69, 113, 107, 82, 122, 95, 56, 62, 91, 116, 120, 71, 78, 89, 90, 108, 15, 121]. The more restrictive value of ≤ 2% [97, 110] and the less restrictive values of ≤ 5-5.5% [104, 111, 105, 106] may also be appropriate for single or different day variability. In this context, for example, some studies investigated the test-retest reliability of VO2max attainment applying two [97, 87, 120] or even five [95] trials with the same CPET and verification protocols, reporting a coefficient of variation of less than 5% between the highest VO2 values observed in the CPET and verification phase. However, further research is required before recommendations can be made to determine whether the highest VO2 from the ramp or continuous-incremented CPET is different from the verification phase beyond the technical error of measurement and intra-individual biological variation.”

Line 633: Please change wording. The verification phase does not verify VO2max.

ANSWER: The sentence has been deleted.

Lines 633-642: In general, this paragraph is weak. The authors presented the HR responses, but the discussion is very limited. For example, the idea that the kinetics of HR is slower than the kinetics of VO2 is not as evident as the authors make it sound (there is more information on this than the single reference that the authors presented). This could explain the differences in some studies, but likely not in the majority of them. I would argue that a priming effect with already improved blood flow availability before the onset of the verification phase might also play a role (although this is just speculation). Additionally, given that having a lower HR during the verification phase did not affect the VO2 responses, then the sex, intensity, etc. effects become irrelevant in my view. To be honest, I feel that given that this study is about the verification phase for determining whether or not a higher VO2 value can be achieved during this process, I would simply delete the HR analysis as it does not add anything to the story. In fact, it makes it longer and less focused.

ANSWER: The heart rate data analysis has been omitted to focus solely on VO2 data and therefore the mentioned paragraph has been deleted.

Lines 655-672: The conclusion in general is too long as reads more as a summary than a conclusion.

ANSWER: The conclusion section has been revised to address both reviewers’ concerns.

Line 660: This is incorrect. Your analysis did show that different procedures can be applied to establish the same VO2 response during the verification phase as compared to the incremental test. However, by no means you can say that this procedure contributed to establish a true VO2max response (not even if using quotation marks for the word “true”).

ANSWER: We have revised the text and it now reads as follows: “From a practical perspective, our findings indicate that different procedures may be applied to establish similar highest mean VO2 responses during the verification phase as compared to the ramp or continuous step-incremented CPETs.” 

Lines 660-662: As I indicated before, what this meta-analysis highlighted is that any hard-enough intensity of exercise (i.e., “respectably” above the critical intensity) performed for long-enough will result in the highest possible VO2 response. Then, the recommendation that the verification phase should be constrained to the ones seen in the current analysis is unfair to me (or at least unnecessary). By doing this, the authors provide some level of validity to the proposed procedures and, indirectly, take validity away from other options that might be equally effective.

ANSWER: The reviewer is correct, thank you for the comment. The following information has been included: “Even then, it is worth mentioning that some caution must be exercised concerning the selection of sub or supra peak WRs since any exercise above the critical power must also be sustainable for sufficient duration to allow the achievement of the highest possible VO2 response in the verification phase.” 

Lines 662-666: This makes no sense to me as reaching the same HR response is not a prerequisite to reach the highest possible VO2.

ANSWER: This sentence has been deleted.

Lines 668: No, this verification phase has not validity as proof that VO2max has been achieved. It simply demonstrates that a higher VO2 value cannot be achieved during the verification phase. I insist in the idea that I would argue that this was, in fact, VO2max. However, that was already known from the incremental test (which is clearly shown in your data). What I oppose is the wrong idea that VO2max can be confirmed because the plateau criterion has been met. This is clearly not the case as the model is inappropriate to show that. Some people in the scientific community seems to lack the understanding of the differences between constant work rate and incremental exercise responses, which has caused this misinterpretation of the verification phase as a tool to satisfy the plateau criterion.

ANSWER: The correction has been made.

Lines 668-670: This is correct. The verification phase did not add anything but some level of confidence that people pushed hard enough. I am not against that thought (as long as people do not confuse this with the idea that confirmation of VO2max has been achieved).

ANSWER: Thank you for your comment.

Lines 670-672: This is not a conclusion from the present study, but rather an opinion. I would delete this.

ANSWER: This sentence has been deleted.

Reviewer #2: 

General Comments

This study aimed to provide a systematic review and meta-analysis of the validity of deriving VO2max and HRmax from a verification phase, with a cardiopulmonary exercise test (CPET) considered the gold standard. It is a thorough investigation of the studies published within this field to date, and the data appears to have been presented objectively and in a clear manner. The research question is relevant and the analysis necessary.

ANSWER: Thank you for your comment.

One fundamental problem I have with studies discussing the attainment of a “true” VO2max (or maximal anything) via CPET and VP (or equivalent) is that we actually don’t know in any scenario whether a “true” maximal value really has been attained. This almost philosophical point (although I find it to be quite obvious and, in this case, physiological) is almost always overlooked and it has been once again in this paper. I think it would be judicious of the authors to acknowledge – and discuss in some detail – this fact (rather than just skim over it at the end of the discussion, in relation to Noakes’ [150] critique).

ANSWER: The reviewer is correct, thank you for the comment. The manuscript has been revised in order to focus on the comparison between the highest VO2 values during the CPET vs. verification phase. We agree with both reviewers that it remains unclear whether a ‘true’ maximal value really has been attained. Therefore, terms and sentences related to the confirmation of a ‘true’ VO2max have been discarded in the revised manuscript.

Another fairly major concern I have is that the analyses are limited to comparing maximal VO2 measures derived from a CPET versus a verification phase at a group level (also highlighted as an issue by Noakes [150]), which is not the same as measuring validity (i.e., through the agreement of two measures). Two measures can be similar (or not significantly different) at a group level, but the agreement can still be very poor. Without an analysis of agreement, how can validity be inferred?

ANSWER: We agree with the reviewer that this is an important issue of the present study. Unfortunately, after countless failed attempts, we concluded that would only be possible to compare VO2 data derived from a CPET vs. verification phase at a group level. Even so, 26 studies were not meta-analyzed because the authors did not answer our emails. Upon reflection, we have now clearly stated that the aim of the study was “to systematically review and provide a meta-analysis on the application of the verification phase for confirming whether the highest possible VO2 has been attained during ramp or step-incremented CPETs in apparently healthy adults” and the term “validity” has been excluded throughout the manuscript. 

Specific Comments

Title

I’m not sure why single quotation marks are needed around the selected terms. Also, “recommendations for best practice” were not scientifically investigated, but are (a small) part of the discussion/conclusion (as is the case in many papers). So I suggest removing these components of the title. This would give something more succinct and specific, like: “The verification phase for confirming maximal oxygen uptake in apparently healthy adults: A systematic review and meta-analysis”.

ANSWER: The title has been revised according to the reviewer’s suggestion.

Abstract

On analyzing the title and aim of the study, I don’t feel this abstract is necessarily a clear summary of the most relevant findings. The results are rather heavily focused on the HR data, which is not a main focus of the study. Please reflect and re-consider.

ANSWER: The results have been rewritten in the ‘Abstract’. To address both reviewers’ concerns, HR data have been excluded from the revised manuscript to focus solely on VO2 data.

L37: CPET on “a” cycle… (the article seems to be missing).

ANSWER: The correction has been made.

L41: The punctuation suggests that this is the age range (and even VO2max) of the women only. Please clarify.

ANSWER: Thank you for the observation. The descriptive data refer to all individuals. The comma has been changed to a semicolon. 

L42, 44: n = 52/36… presumably these are the number of studies? Please clarify.

ANSWER: The reviewer is correct. The first version included 78 studies in the systematic review, but only 52 (VO2max) and 36 (HRmax) studies were meta-analyzed. The current version added 2 studies and excluded HR data and now reads as follows: “The highest VO2 in the CPET and verification phase was similar [n = 54, mean difference = 0.03 (95% CI = -0.01 to 0.06) L/min, P = 0.15] (…)”. 

L44: Can (should) bpm be expressed to decimal places? Also, 5 d.p. on the P value seems quite excessive (as is the case throughout the manuscript for very small P values – is there any reason for this?).

ANSWER: The HR data have been excluded from the revised manuscript so this has resolved the issue of decimal places for HR data. P values in the revised manuscript have been given to a maximum of three decimal places.

L44-47: It is unclear what the comparisons (3 bpm greater HR) relate to for the three P values. What are they being compared to? Please clarify. That said, I’m not sure why so much focus is given to HR in the abstract, when the study is really about verifying VO2max.

ANSWER: The HR data have been omitted from the revised manuscript, so this issue has been resolved.

L50. Why would concordance (agreement) “put [this] into question”? This seems contradictory. Please clarify.

ANSWER: The conclusion has been revised to address the concerns of both reviewers, and now reads as follows: “The verification phase seems a robust procedure to establish consistent values for highest mean VO2 responses following a ramp or continuous step-incremented CPETs (…)”.

Introduction

The Introduction is long and there is rather a lot of discussion around the general topic of VO2max testing, but sparse detail relating to the actual topic of the study (i.e., the use of verification phases in all their various make-ups). The sentence at L126-129 to me is the crux of the problem and the study, and this is what the introduction should focus on more exclusively. The study referenced in L130-131 ([53]) requires more discussion/explanation, so that there is clearer context for the current study and justification for the sentence at L131-134. I have no issue with the importance of this work, but a bit more clarity around the actual problem (and existing literature) is required.

ANSWER: Thank you for the insightful comments. The introduction has been revised to address the reviewer’s suggestions (please see text in red).

L55-58: This is a very long opening sentence. I suggest breaking it in two (if all content is to remain).

ANSWER: This comment has been addressed in our response to your previous comment.

L61: Should the refs be listed in numerical order (e.g. [1, 3-6])? (There are other examples throughout the manuscript as well.)

ANSWER: The corrections have been made.

L69-70: A transition from that stated, but to what? Continuous fast ramp tests? It feels like the end of the sentence is missing – maybe combine (and condense?) it with some of the text from the following sentence (L70-72).

ANSWER: The sentence has been revised and now reads as follows “These technological advances have contributed to a transition from the original time-consuming discontinuous step-incremented protocols to more time-efficient continuous ramp or pseudo-ramp protocols for determining VO2max [20-25].”. 

L74: Should this be “limitations of VO2max” (i.e., the method of measurement)? The limitations “to” VO2max is a different topic, as I see it, more related to training, genetics, etc.

ANSWER: We have now corrected this - thank you.

L83-86: There seem to be two contrasting definitions of the Taylor et al. VO2 plateau criterion here. Please clarify.

ANSWER: The reviewer is correct, thank you for the comment. In the 'Conclusion' section, Taylor et al. stated that the increase in VO2 was associated with an increase of 2.5 % grade (below the VO2max), which was ~ 300 mL/min. If the VO2 at two different grades differs by less than 150 mL/min (i.e. 50% of the expected increase), it can be assumed that a VO2max was attained. To avoid misunderstanding and make the text simpler and clearer, the text now reads as follows: “The landmark study of Taylor et al. [34] was the first to use a formal VO2 plateau criterion, which was defined as an increase in VO2 of less than 150 L/min (or ≤ 2.1 mL·kg-1·min-1, considering an average body mass of 72 kg) in response to a specific discontinuous step-incremented protocol performed over 3-5 laboratory visits.”

L87: Should the ref ([32]) not be included with Taylor et al. even here?

ANSWER: The correction has been made.

L94: Typo on VO2max (the second time).

ANSWER: The correction has been made.

L97-99: Check and change the grammar/punctuation. Something is going on around “investigators; however, due to”, which makes the sentence incoherent (to me).

ANSWER: The revised manuscript text reads as follows: “However, this approach has been widely criticized by numerous investigators due to the individual variability in maximal physiological responses for these variables and lack of specificity in identifying individuals who did not continue the CPET to their limit of exercise tolerance [40, 36, 37, 29, 38, 2].”

L77-104: This is a very long paragraph. I suggest splitting it in two – the first relating to VO2max criteria, the second to the secondary criteria. Or write the content more concisely to produce one shorter paragraph.

ANSWER: The reviewer is right, thank you for the observation. The mentioned paragraph has been split into two paragraphs: the first one introducing the VO2 plateau and the other one addressing the limitations of secondary criteria.

L105-139: These two paragraphs are most important in justifying the current study, so I think a more in-depth discussion of this literature is required (instead of the level of detail presented in the three preceding paragraphs, which could be significantly condensed).

ANSWER: The introduction has been revised to address the reviewer’s suggestions (please see text in red).

Methods

L162: “ergometer or treadmill” – was this limited to bi-pedal running on a treadmill? Please specify, as many other modes of exercise are possible on a treadmill (e.g., cycling, hand-cycling, wheelchair running, inline skating, roller skiing, etc.). It would be useful to make this clarification through a clear definition somewhere in the paper, that by “treadmill” (see for example Figure 1, “Only treadmill”) you actually mean “treadmill running” (if that’s the case).

ANSWER: The sentence has been revised as follows: “the CPET was carried out on a cycle ergometer (i.e. bipedal cycling) or treadmill (bipedal running or walking)”. 

L171-172: “In the final review, we provided…” – is this referring to what is presented in the current manuscript? Please clarify, as “final review” and “we provided” is a bit unclear to me.

ANSWER: Thank you for the observation. The text has been revised and now reads as follows: “We provided a flowchart of included and excluded studies, with reasons for their exclusion.”

L196: You have previously written abbreviations within round brackets in square brackets: (95% confidence interval [CI]).

ANSWER: The correction has been made. 

L197: Out of interest, what did you do in cases where VO2max was reported in mL/kg/min?

ANSWER: In the absence of absolute VO2 data, we performed the metabolic conversion. However, most of the studies presented the data in L/min or mL/min units and the few studies that only presented the data related to body mass we got to 'rescue' ~90% of the data from the authors via e-mail. 

L210: P-values “were” obtained…?

ANSWER: The correction has been made – thank you.

L215: Is this less than 50%, or less than or equal to? The symbol looks unclear to me.

ANSWER: Less than or equal to (≤).

L223: The studies were also…

ANSWER: The correction has been made.

L225: Stratified analyses were also…

ANSWER: The correction has been made.

Results

The main issue for me throughout this section is the long lists of references accompanying each result. This is not common in other studies of this type that I’m familiar with, and to me it makes deciphering the interesting information nigh on impossible. I would recommend removing these long number strings.

ANSWER: The correction has been made.

L241: (interquartile range [IQR]) – or consistent with previous presentation.

ANSWER: The correction has been made.

L240-245: You write that “the sex of 130 participants was not specified”, and then that “one study did not specify the sex of the participants (see Table 1)”. In that study (Scheadler and Devor [92]) n = 13, so I don’t understand the mismatch between 130 and 13. Please clarify.

ANSWER: Some studies presented data from males and females, but without reporting how many were males and how many were females. Scheadler and Devor recruited 13 individuals; however, the sex of them was not stated. 

L246: BMI should presumably be defined after the words (body mass index) and doesn’t then need to be included in the brackets.

ANSWER: BMI has been previously defined in ‘Data Extraction and Management’ section (Page 7, Line 164). The parentheses and square brackets were used here to clarify what measures were adopted [i.e. mean ± standard deviation (range)]. 

L246-247: The square and round brackets seem to have switched places in this sentence, any reason?

ANSWER: As answered in your previous comment, there are other results within square brackets, such as mean, SD, range. This is consistent with values from those reported in global analyses (i.e. VO2max).

L247: Writing “(VO2max normalized to body mass)” seems superfluous when you have the unit as mL/kg/min. Consider removing.

ANSWER: The correction has been made. 

L251: “Characteristics of studies using CPET…”?

ANSWER: The correction has been made.

L253, 255: “on a cycle/treadmill” (again, the article is missing).

ANSWER: The correction has been made.

L253-307: These long strings of references make the results very unreadable. I suggest removing them all, as finding the actual interesting numbers (i.e., the results) through the long lists is so difficult.

ANSWER: The correction has been made.

L272-274: Could you re-phrase to fix the grammar and clarity on: “whereas 29 (37%) used fixed intervals of 15- to 30-s (or 2 × 15-s), both averaged and fixed times (1%) [61]… etc.”. I guess the 29 (37%) relates specifically to the 15/30-s fixed interval data, so the sentence needs to be re-structured and improved to clarify this in relation to the other methods listed.

ANSWER: The text has been revised for grammar and clarity according to the reviewer’s recommendation.

L279, 281: I think you need to include the “min” unit after 5, 6, 6, 9 and 15 – or not if you were to remove all the references in brackets (another example of how difficult the interesting numbers and results are to decipher from the long [and unnecessary?] lists of references).

ANSWER: Thank you for the observation. In accordance with the Reviewer’s suggestion, we have added the “min” unit.

L285-286: Why are two different %ages presented (19 and 19.7)?

ANSWER: We apologize for this error. The correction has been made and now the text reads as follows: “Fifteen studies (19%) carried out the verification phase on a different day to the CPET”. 

L291: Suggest removing “i.e.”? Not included elsewhere.

ANSWER: The correction has been made.

L295: Should this be “the” maximal-intensity work rate, rather than “a” (presumably it was specific to that study and the preceding VO2max test).

ANSWER: The correction has been made.

L297: Could you briefly describe in this sentence what the formula was based on?

ANSWER: The following sentence has been included in the revised manuscript to address this issue: “predicted WR based on a formula (1%) to elicit the subject’s limit of tolerance within 180 s as follows: power output = (finite work capacity ÷ 180 s) + critical power”.

L300: “Forty-two studies (54%)” – consistent reporting.

ANSWER: Thank you for your comment.

L300: obtained “during” (rather than “at”)?

ANSWER: The correction has been made.

L366-onwards: Is there a reason for changing the presentation (order of using) round and square brackets again? And see my point above (in the Abstract) about the number of decimal places on the P values < 0.001. Is there any statistical reason/need for this?

ANSWER: Usually, the main findings of the meta-analysis are described from the mean difference, 95% CI, and P-value. We have revised the number of decimal places on the P-values throughout the manuscript.

L383: (performed on the same day as vs. a different day from the CPET) – suggestion.

ANSWER: The correction has been made.

L387-388: Could you include the P value here for this no sig diff, as it is a key result.

ANSWER: The required information has been included in the revised manuscript. 

Discussion

At times I struggle to follow the logic of the arguments in this (very long) discussion, so I think the interpretations can be written more clearly and concisely in places. In addition, there is a lot of discussion of previously published studies and concepts, without reference to the findings from the current results. This seems inappropriate for a systematic review/meta-analysis, so I would encourage the authors to focus more on their own findings in light of previous work, rather than merely presenting a review of the existing literature.

ANSWER: The Discussion has been revised to address both reviewer’s suggestions (please see text in red).

L422: Reconsider “over” in this sentence. Maybe “rather than”?

ANSWER: The HR data have been removed from the revised manuscript, which has resolved this issue.

L433-435: This study did not analyse children or clinical groups, so where is the “current evidence suggesting that the verification phase is a safe and well-tolerated procedure to confirm attainment of true VO2max” in these groups? This particular study can surely only make this claim about the apparently healthy adults who were analysed, or am I missing something?

ANSWER: The reviewer is correct, thank you for the comment. The relevant sentence has been moved to the last paragraph of the revised ‘Discussion’ to emphasize the need for future meta-analysis studies focusing on special populations.

L445: “of a ramp-incremented…” (missing article).

ANSWER: The correction has been made.

L455-456: Are 17% and 33% comparable in this sentence? If so, please use the same unit (either CPETs, or participants) in order to compare like with like (e.g., “17% of participants (2 of the 12) during a cycling ramp-incremented CPET, while 33%…”.

ANSWER: This information has been revised as follows: “Similarly, Rossiter et al. [37] reported the occurrence of a deceleration in the VO2 response at the limit of exercise tolerance in only 17% (2 of the 12) of cycling ramp-incremented CPETs (WR increment of 20 W/min), while 33% of the incremental tests elicited an accelerated VO2 response, and 50% demonstrated a linear VO2 response.”

L477-478: Is this statement true? If so then I’m missing something. Re-reading ref #100 (McGawley 2017) it is stated that: “There was a signiﬁcant eﬀect of test type on VO2max, with higher values recorded during STEP compared with VER (P = 0.013)”. Can you clarify how you’ve come to this conclusion (three studies to-date) and how you conducted this analysis/check?

ANSWER: Apologies for missing this. We have double-checked the results from all meta-analyzed studies and included the mentioned study as one of those who reported significant differences between the VO2 data from CPET vs. verification phase.

L494: Is there a typo here: “4 of the 7 participants (9%)”?

ANSWER: The correction has been made.

L496: “11 participants (9 men; age…)” – it appears as if you are only reporting the descriptives for the men, is this the case? Please clarify.

ANSWER: To address this comment the revised text now reads as follows: “In another study with 9 men and 2 women (age: 22.4 ± 3.21yr.; VO2max: 51.6 ± 4.47 mL·kg-1·min-1)”.

L499-503: From your results and Figure 2 this looks like an outlier. Has/should there been any accounting for outliers in your analyses?

ANSWER: We are in agreement with the reviewer’s comments about Colakoglu’s data. Notably, after omitting their data, there was no change in the main results of the present meta-analysis. It was therefore not appropriate to omit these data from our analysis and we prefer that readers draw their own conclusions about the quoted study.

L477-516: This is a very long paragraph (> 1 page). Please consider shortening. I don’t think all the detail of the three specified studies is required (L478-503) – this could be condensed and written more concisely.

ANSWER: The mentioned paragraph has been reviewed in order to focus on the main findings of the present meta-analysis.

L508-516: This seems to explain this result as an outlier. What happens to your findings (CPET vs verification phase VO2max) if this study is removed from your overall analyses?

ANSWER: As commented previously, after removing the VO2 data from the Colakoglu’s study there was no change on the present meta-analysis findings regarding the highest VO2 values from the CPET vs. the verification phase (e.g. P-value changed from 0.15 to 0.17 after removal of this study).

L517-519: I’m not sure I agree with this statement (or maybe I misunderstand what you mean). If CPET = VP or if CPET > VP then is that confirmation of a “true” VO2max during the CPET? Can both tests not elicit a VO2max that is lower than an individual’s “true” VO2max in this scenario?

ANSWER: Within this scenario (i.e. highest VO2 from the CPET ≥ to highest VO2 from the verification), we can only confirm that VO2max was likely attained. According to Midgley et al. [97], if the mean highest VO2 attained in the verification phase is significantly higher than in the CPET, the investigator should consider that the CPET protocol was inadequate in eliciting a highest possible VO2 response in all, or at least some of the participants.

L520-521: I don’t quite follow the logic of this follow-up sentence. Are you saying that there would need to be a difference in order for the statement in the first sentence to be true? Why? Please clarify.

ANSWER: We are saying our data concurs with the notion that there was no significant difference between the highest VO2 values attained during the CPET and in the verification phase. In other words, this means that a verification phase applied after ramp or continuous step-incremented CPETs may offer robust evidence that the highest possible VO2 has been achieved, and it is not affected by sample’s characteristics, exercise modality, or CPET and verification protocol designs. The mentioned sentence has been revises and now reads as follows: “The present meta-analysis findings in 54 CPET/verification phase studies are in good agreement with this premise. Our data did not confirm the existence of significant differences between the effect sizes of the highest mean VO2 values attained in CPET and subsequent verification phase in primary studies with relevant data available [n = 54; mean difference = 0.03 (95% CI = -0.01 to 0.06) L/min, P = 0.15] (see Figure 2).” 

L526: Why “only” 25 (i.e. 27%)? To me this 27% of the studies is important in demonstrating that the CPET doesn’t always do its job properly (i.e., in eliciting a “true” VO2max). This is where the analysis of agreement is important too – what is the similarity (or dissimilarity) in VO2max values derived from a CPET vs VP “within” individuals? Please comment.

ANSWER: Considering the 103 experimental conditions that were meta-analyzed, 25 displayed average VO2 values in the verification that exceeded the average value of the VO2 attained during the CPET, however, without statistical significance. Only one study (i.e. Colakoglu et al. 2016) showed a significant difference in favor of the verification phase (i.e. higher ‘peak’ of VO2 in the verification phase vs. the CPET). 

L517-528: I actual don’t follow the logic or point being made in this section. Could you please try to clarify?

ANSWER: This paragraph has been revised as follows: “To-date, this systematic review has demonstrated that the majority of studies have shown that the highest mean VO2 values elicited by verification phase bouts were similar (or not statistically or meaningfully different) to the attained VO2 values in continuous ramp or pseudo-ramp CPET protocols [104, 114, 36, 125, 97, 37, 77, 81, 38, 60, 96, 39, 57, 102, 101, 63, 86, 88, 119, 61, 69, 73, 79, 99, 64, 65, 94, 113, 124, 126, 103, 107, 115, 53, 66, 75, 76, 112, 67, 80, 85, 117, 123, 56, 72, 84, 87, 91, 116, 118, 120, 59, 71, 74, 78, 83, 89, 90, 105, 108, 109, 15, 121, 70, 106]. The present meta-analysis findings in 54 CPET/verification phase studies are in good agreement with this premise. Our data did not confirm the existence of significant differences between the effect sizes of the highest mean VO2 values attained in CPET and subsequent verification phase in primary studies with relevant data available [n = 54; mean difference = 0.03 (95% CI = -0.01 to 0.06) L/min, P = 0.15] (see Figure 2). In fact, the mean absolute difference of 0.03 L/min represents a relative error of only 0.85% between the highest VO2 values attained in the CPET and verification phase (this is within the most commonly adopted measures of test variability at 2-3%). Comparing only the mean values from the highest VO2 responses in either CPETs and verification phase bouts (i.e. regardless statistical significance reported in the primary investigations), 27 (i.e. 26%) out of 103 specific experimental conditions included for overall comparisons presented average VO2 values during the CPETs that were below those attained in the verification phase [mean diff = -0.06 (-1.6%) L/min]. Whereas, 76 conditions the highest VO2 responses during the CPETs were similar or above to those attained in the verification phase [mean diff. 0.06 (1.7%) L/min] (see Table 3 and Figure 2). In addition, the present findings also provide consistent and unbiased confirmatory evidence that the reproducibility of the highest VO2 value during CPET and verification phase does not appear to be affected by sex, cardiorespiratory fitness, exercise modality, CPET protocol design, or even how the verification phase was performed (see Table 4 and Figure 4). Collectively, these findings indicate that the verification phase procedure provides some level of confidence that the highest possible VO2 has been likely elicited during a single session CPET.”

L542: “who” underwent?

ANSWER: This paragraph has been deleted. 

L544: was similar “to the”?

ANSWER: The correction has been made.

L547: At this point I’m really struggling to follow the logic and arguments presented over the last few pages. Are you saying that CPET should be higher than VP in order to accept that a true VO2max has been attained in the CPET? Why? What is the problem with CPET = VP? A more fundamental question, in my opinion: Why is it not acknowledged/discussed that individuals can very easily underperform on BOTH tests, and that we really don’t have any idea as to whether we have attained a “true” VO2max at all. Please comment.

ANSWER: This sentence has been deleted to address the review’s concerns.

L552-557: The “different” methods described previously for study #94 are also relevant here, as is my comment above (i.e., that it is always possible that neither test was truly maximal and elicited a “true” VO2max).

ANSWER: The relevant paragraph has been omitted from the revised manuscript to address the reviewer’s comments. 

L557: What is meant by “this put into question” in this context?

ANSWER: The mentioned expression has been deleted.

L561-562: This is the first time this endeavor has been mentioned (except in the title). Please re-consider the phrasing here (and in the title!) – especially given the conclusion of this sentence (L567-569), i.e., that no best practice can actually be recommended.

ANSWER: The correction has been made.

L570-571: This list of 6 references does not seem complete, or to reflect “most studies”. Please clarify.

ANSWER: The reviewer is right, thank you for the observation. The mentioned list of references has been updated.

L570-578: What new insight does this paragraph add, from the current results, which was not already known? Please embellish with additional information, or remove.

ANSWER: This sentence has been changed to address the first reviewer’s comments and to keep coherence with the main findings of the present meta-analysis. 

L584-585: I don’t think decimals are needed on these %ages.

ANSWER: The correction has been made.

L579-596: Again, how do the current results relate to the previous literature? This is not a review article, so as I see it the discussion section should be used to present the results of the current study in the context of previous results. The information presented here (that 105% was different from 115% according to Nolan et al.) is not supported by your results, as I understand, since you saw no significant effect of VP intensity. This is what ought to be discussed, in my opinion.

ANSWER: This is a good point. Although the present meta-analysis had not detected any potential moderator between the highest VO2 values during the CPET and verification phase, it is worth mentioning that an inappropriately high-intensity in the verification phase protocol (as expressed by the WR selected from that achieved in the CPET) would result in a short test duration that results in insufficient time to reach VO2max. In a recent study of Iannetta et al. [25], the question about “how much lower/higher than WRpeak should the WR of the verification phase be set at?” have been widely discussed and included in the revised discussion section. Thus, we considered it important to highlight some aspects of the verification phase design even in the lack of statistical significance in our data.

L597-610: Again, this is a review of the existing literature. Please discuss the results of the present study.

ANSWER: The ‘Discussion’ section has been rewritten in order to address both reviewer’s demands.

L613: Remove the extra space(s) between Small and sampling.

ANSWER: The correction has been made.

L614: rapid changes

ANSWER: The correction has been made.

L611-621: Same issue again - this is a review of the existing literature without reference to the current study results. Please reconsider.

ANSWER: The ‘Discussion’ section has been rewritten in order to address both reviewer’s demands.

L624: should not exceed

ANSWER: The correction has been made.

L636: on the duration of

ANSWER: The mentioned paragraph has been deleted. 

L642: when “what” are short? The VPs? Please clarify.

ANSWER: The mentioned paragraph has been deleted. 

L649-653: I don’t think the second sentence is a good enough “get-out” given the significance of the criticism stated by Noakes. This underpins the entire concept of “validity”. Do not overlook or underplay the fact that your final sentence, which you say you are not doing (“rather than the question of whether an individual has elicited a ‘true’ VO2max.”), is exactly what you say you are doing in the title (as I interpret things)!

ANSWER: This paragraph has been revised and the ‘validity’ term has been excluded throughout the manuscript.

L655: Have effect sizes been presented anywhere?

ANSWER: These data have been presented in Table 3. 

L656-657: “in cycle ergometry and treadmill running”?

ANSWER: The correction has been made.

L660: “compromising their ability” (plural)

ANSWER: The correction has been made.

L670: I don’t understand this, in the context of the previous sentences: “The mandatory application of the verification phase in all situations may be therefore questioned”. Why questioned?

ANSWER: This sentence has been deleted.

L671: settings?

ANSWER: This word has been deleted.

Tables & Figures

The studies appear to have been ordered chronologically and then alphabetically in Table 1, with this ordering system then continued throughout the later tables & figures. This seems arbitrary (chronologically then alphabetically) and makes it difficult to locate any specific study in the later tables/figures. Could you order the studies entirely alphabetically or according to the reference numbering from the outset?

ANSWER: The correction has been made and now the studies are ordered alphabetically. 

Is there any reason for presenting the subgroup analyses according to the characteristics of the verification phase protocol in a “figure”, while the subgroup analyses regarding sex, cardiorespiratory fitness level, exercise modality, and CPET protocol are presented in a “table”? Could this method of presentation be standardized? Also, I’m not familiar with the presentation used in Figure 4. Can you provide more information about how to read it (top line with green box, middle line with green box and black diamond), as it won’t be clear to all readers.

ANSWER: This design is similar to that adopted in a previous meta-analysis (Cornelissen and Smart. Journal of the American Heart Association, v. 2, n. 1, p. e004473, 2013). The Table 5 shows 4 sub-analyses, as follows: according to sex, cardiorespiratory fitness level, exercise modality and CPET protocol. Each of them has 2 or 3 groups. In addition, this table presents the duration from CPET and verification. Therefore, we decided to use a table instead of a figure to report relevant outcomes, but in a smaller illustration. A similar plot was presented in that aforementioned meta-analysis. The green boxes represent the VO2max obtained in the CPET vs. verification for each subgroup. For example, there is a green box for VO2max between CPET and verification for sub peak WR and another for supra peak WR analysis. There is a green box for active and another one for passive recovery, and so on. The black diamond represents a combined effect between groups (i.e. sub peak WR and supra peak WR for intensity, active and passive for recovery, etc.). In other words, the black diamond takes into consideration the effect size of all subgroups within each analysis to provide a final result. 

Table 1: The heading “mean values” should probably be aligned over the final three columns to the right, as sex and N are not means. Also, ranges should be differentiated in this heading, if that’s what those are and if they can’t be expressed as means (e.g., 25-35 and 19-61). And can/should the number of decimal places be standardized in the data? Any reason that some terms (e.g., Sedentary, Cyclists, Runners, Athletes) are capitalized, but others aren’t?

ANSWER: Table 1 has been revised accordingly.

Tables 3/4: Can Total be clarified (presumably it’s the number of participants, but this is not stated anywhere). The %Weight is hard to comprehend – I have no experience of this measure or its calculation, but the statistical power seems to bear no relation to N, which seems odd to me. Can you explain?

ANSWER: Total is the number of participants who performed CPET and verification phase. According to the Review Manager (RevMan), we can observe the following order of the data: Test A: mean, standard deviation and N (participants) vs. Test B: mean, standard deviation and N (participants). The %Weight is attributed to each study due to its statistical power. The %Weight is affected by N and mainly by the standard deviation. Please, look at two studies: Weatherwax et al. [122] (second experimental condition) and Kramer et al. [91]. Although the sample size (n = 6 and 15, respectively) were not huge, the standard deviation was low. The %Weight from the studies of Astorino and DeRevere [56] (second experimental condition) and Niemeyer et al. [126] overcame 3% due to the large sample size (n = 79 and 46, respectively). 

Table 5: Can horizontal lines be used to clarify where each category (TTE, VO2max, HR) starts and ends (i.e., to the right of each N)?

ANSWER: The correction has been made.

Figure 1: Can you clarify (even if just to me) why the 1 full article excluded in the Eligibility stage due to “Non-maximal exercise test protocols…” had not already been excluded for the same reason in the Screening stage?

ANSWER: In the records screened for inclusion after removing duplicates, we have looked at the title and abstracts and during this stage it was possible to identify 26 articles to be excluded. However, only after the following stage (i.e. full articles assessed for eligibility), it was possible to detect another study to be excluded. These numbers match with our flowchart. We have also contacted the authors to check whether or not the verification phase was performed until volitional exhaustion.

Figure 2: The data suggests to me a tendency for CPET to be higher than VP. Is there any accounting for potential outliers (e.g., Colakoglu et al.)? What happens if this study is removed from the analyses (if there is good reason to do so, which reading the discussion there might be)?

ANSWER: As previously mentioned, there was no impact on our results after removing the VO2 data from the aforementioned study.

Figures 2-4: The quality of these figures is poor (due to the high level of detail). Can they be presented at a higher resolution?

ANSWER: The figures have been remade with 300 DPI to improve resolution.

Reference list

ANSWER: The correction has been made.

L684: (1985) should presumably be removed from the JAP title?

ANSWER: The correction has been made.

---

## [Decision Letter · Decision Letter 1]

7 Dec 2020

PONE-D-20-25408R1

Verification phase for confirming maximal oxygen uptake in apparently healthy adults: A systematic review and meta-analysis

PLOS ONE

Dear Dr. Cunha,

Thank you for submitting your manuscript to PLOS ONE. After careful consideration, we feel that it has merit but does not fully meet PLOS ONE’s publication criteria as it currently stands. Therefore, we invite you to submit a revised version of the manuscript that addresses the points raised during the review process.

Specifically, a thorough revision of the English is required, as well as a revision of the Discussion section to be clearer.

We look forward to receiving your revised manuscript.

Kind regards,

Laurent Mourot

Academic Editor

PLOS ONE

Reviewers' comments:

Reviewer's Responses to Questions

**Comments to the Author**

1. If the authors have adequately addressed your comments raised in a previous round of review and you feel that this manuscript is now acceptable for publication, you may indicate that here to bypass the “Comments to the Author” section, enter your conflict of interest statement in the “Confidential to Editor” section, and submit your "Accept" recommendation.

Reviewer #1: All comments have been addressed

Reviewer #2: (No Response)

2. Is the manuscript technically sound, and do the data support the conclusions?

Reviewer #1: Yes

Reviewer #2: Yes

3. Has the statistical analysis been performed appropriately and rigorously? 

Reviewer #1: Yes

Reviewer #2: Yes

4. Have the authors made all data underlying the findings in their manuscript fully available?

Reviewer #1: Yes

Reviewer #2: Yes

5. Is the manuscript presented in an intelligible fashion and written in standard English?

Reviewer #1: Yes

Reviewer #2: No

6. Review Comments to the Author

Reviewer #1: I would like to thank the authors for their attention to my comments. I think this revised version is much better and I believe that this manuscript would be an excellent reference on the topic. From my perspective, I only have on minor suggestion that I would leave up to the authors to accept or reject. This is: As presented, the title seems to imply that a verification phase is something necessary. Would it be better crafting the title as a question? Something like “Is a verification phase for confirming maximal oxygen uptake useful in apparently healthy adults: A systematic review and meta-analysis”.

Regardless, I would like to congratulate the authors for the high quality of this meta-analysis. I often feel that this type of scientific contributions are not as meritorious as original research, but I think that with the amount of information that is currently available on this topic, this particular meta-analysis is fully warranted. Thanks!

Reviewer #2: General comments

Preparing this manuscript has clearly been a huge job, so I commend the authors on this significant undertaking. I have again reviewed this paper in thorough detail and have a number of observations and feedback points, specified below.

In general, I would firstly urge the native English speakers on the author list – or a professional proof reader – to take responsibility for thoroughly checking the language (especially the grammar) before resubmitting, as the current level of writing makes the text difficult to comprehend. There are particular issues in the new sections of the Discussion that need re-writing/correcting to improve the clarity and flow.

Secondly, and perhaps most importantly, the Discussion to me is too long. It makes it impossible for the reader to grasp the key findings and messages from this paper, as there is just so much detail of previous studies, their protocols and specific findings (data, P values, etc.). I personally feel that the paper would be far more comprehensible and impactful if the Discussion was more concise.

Abstract:

L33-35: The use of semicolons in this list seems odd, and makes it difficult to understand how the search was conducted. Should they be commas? Please re-consider.

L43: Why the comma after VO2max but not after age? Also, the -1 on the VO2max unit looks too high.

L44: Can you clarify that the VO2 values were similar in 54 of the 80 studies, because at the moment it seems like this is a result for a total of 54 studies analyzed, which I don’t think is the case. Suggestion: “The highest mean VO2 in the CPET and verification phase was similar in 54 of the 80 studies (mean difference…)”. Also, I still don’t understand the inconsistencies in the use of square/round brackets. Can this be standardized throughout the manuscript? Also, why change to L/min in this sentence, after presenting mL/kg/min in the previous sentence? Can this be standardized?

L51: “following a… CPETs” = incorrect grammar. Maybe write CPET.

L51-53: I like this idea, but to me it needs to be related specifically to “your” findings, not attributed to “some [other?] researchers”. Maybe something like: “However, given the high concordance between the highest mean VO2 achieved in the CPET and verification phase, findings from the current study would question its necessity in all testing circumstances.”

Introduction:

L70: Maybe mention the Douglas bag method here, since that is the main reason for the improved time-efficiency (i.e., changing from DBs to B-b-B systems). The type of protocol is merely a by-product.

L73-74: This sentence is not quite right. The “confirmation of VO2max attainment” is not “due to” the listed factors; the “lack of attaining” VO2max might be due to those factors. Please re-phrase.

L82: Presumably this is average “male” body mass. Please specify (there needs to be transparency when sex biases exist in data sets).

L85-89: Is this a new paragraph? It seems very short (2 sentences). Maybe it should be a continuation of the previous paragraph, since it’s the same topic (VO2 plateau)?

L93: “is attained”?

L95-96: I’m not sure what this means: “lack of specificity in identifying individuals who did not continue the CPET to their limit of exercise tolerance”. What is meant by “specificity” in this context?

L103: CP is specific to power (usually analogous to cycling) and not practically applicable to other exercise modes (such as running). Also, “i.e.” is specific (“that is”). So maybe change i.e. to e.g. (“for example”). Or add speed/velocity to power when referring to CP (so CS/CV).

L106: Suggestion (to avoid the use of “each other”): “the highest VO2 values in the CPET are consistent with the verification phase”

L108-111: Please present the “evidence” for Poole & Jones’ statement (i.e., “that a verification phase must be performed at a higher WR than attained in the ramp-based CPET protocols in all future studies”). Where/what is the evidence for this suggestion?

L111: The terminology “On the other hand” seems inappropriate, as this is a different topic compared with the previous sentence (VP WR vs CPET increments). The correct counter-argument here would presumably be the data showing that VP WR can actually be “lower” than WRpeak to still elicit a VO2peak/max.

L113-116: This is now confusing, with “In contrast” following “On the other hand”. Please clarify the arguments that you are making, and the key information you’re trying to get across. I also think WRpeak needs defining. Also, is it necessary to list average W values?

L108-120: In general, this section needs to be more clear and concise.

L120-132: I think all of these ideas can be incorporated into one far more clear and concise paragraph explicitly stating the aims of the study. I suggest moving the questions (at L120-125) down to L138-141 (Methods) and just focusing on the aims here, as required.

Methods:

L136: “is” shown in S1 Text.

L138-141: These are not completely consistent with the text in the Introduction (L120-132). Also, you are still using the terminology “valid alternative to confirm”, which I understood had been changed throughout (since this is not a study of validity). And I’m not sure the study does actually identify “the most appropriate protocol for applying the verification phase”. So I would use questions similar to those currently posed in the Introduction (L120-125) here instead.

L155: participants “who” were

L157-158: Suggestion: “…carried out using bipedal cycle ergometry or bipedal treadmill running or walking.”

L160: Remove “included” (as you already have “involved”, above). Also, importantly, “and” should presumably be changed to “or”, since any one of those three situations would surely lead to exclusion.

L166-167: Writing in the first person seems odd here (we). Are you saying that a flowchart has been included in the paper? If so, where is it? If not, why mention it?

L170-174: There is still an inconsistent use of round and square brackets. Please re-consider (round brackets are typically used in the first instance, and [square brackets] within round brackets.)

L175: Can you specify that you mean “other” authors than yourselves, so “authors of the original articles were contacted…”.

L185: Does “selective reporting” need to be written twice here?

L187: You are inconsistent with your capitalization (or not) of the sub-title words

L206: “the primary study groups results” requires an apostrophe somewhere on groups, depending on the specific meaning (one group or multiple groups) – please correct.

L207: Same here – “groups” needs an apostrophe.

L208: The less than or equal to symbol is still odd here. The lower “equal to” line should be horizontal.

L209: using “a” funnel plot, or using funnel “plots”?

L216: This new red text is unnecessarily complicated. Firstly, you have previously used the term WRpeak (although this doesn’t seem to have been defined anywhere in the manuscript). Secondly, sub and supra peak does not need to be reinforced with < or > 100%, that’s obvious by definition. Suggestion: (i.e. < 100% WRpeak vs. > 100% WRpeak)

L218-220: This sentence could be clearer, suggestion: “as the CPET or on a different day, and the duration of the verification phase (i.e. ≤ 80 s, 81–120 s, > 120 s).”

L222: Should this be “cut-off points”?

Results:

L229-230: This (“Figure 1 summarizes the screening and selection process”) seems like repetition from the Methods section (L166-167). Should the flowchart and reference to it be moved up to the Methods (L167)?

L234: Presumably your sub-section headings should use a different font/presentation from your main section headings.

L235-239: In L235 you use the term “eligible studies”, in L236 you write “included studies” and in L239 you write “primary studies”. This is confusing, because I think you are talking about the same 80 studies in all cases. Can you use consistent terminology for clarity?

L240-241: BMI has already been defined in the paper, so I suggest: “participants had a BMI within the normal range (mean ± SD [range]: 24.4…”. And once again, you need to be consistent with the use of round/square brackets.

L242: VO2mx does not need to be written twice; I suggest removing the first reference to it: “cardiorespiratory fitness (VO2max mean ± SD [range]: 46.9…”.

L243: Delete “according to… [53]” as these have already been defined in the Methods (L222-223).

L252-256: I think this could be clearer, e.g.: “Thirty-three (41%) of the 80 studies included in the review adopted one or more of the traditionally reported plateau or secondary criteria to confirm the attainment of a VO2max, with 30 using a VO2 plateau criteria, 21 using the heart rate plateau or age-predicted maximal heart rate, 18 using RERmax, and 8 using post-CPET blood lactate concentration.”. Line 256 is missing a full stop in any case.

L258: I don’t think the second “time averages” should be hyphenated.

L263: Should this read something like “Regarding the period between CPET and VP…”? Because you seem to be referring to the rest period after the CPET here, right?

L266: What do you mean by “self-paced approach”? A self-paced recovery period? Of any duration?

L270-271: You need to be clearer that you are now referring to the VP exercise intensity. I also think you should decide upon the terminology earlier in the paper, define it and stick to it (e.g. peak WR or WRpeak, supra or > 100%, etc.). Because at the moment there is a mix.

L272: Maybe write 105-130%, consistent with the presentation of ranges in the previous paragraph.

L273: What do you mean by “or both peak or supra peak WR (1%)”? Is this one study, a different study from the “Eight studies” previously stated? If so, please specify. Also, do you mean both peak “and” supra peak? This is unclear to me.

L275: Again, does 1% relate to one study? If so, you could maybe write (one study, 1%) for clarity/consistency. Also, you have previously used the term “participant” rather than “subject”.

L277: “85-95% of…”

L281-282: “1.5-2.2”; ”50-150” (consistent with previously stated ranges).

L294: Figure 2 (not Figures).

L322: I suggest removing the two commas, as this “middle” information is imperative to understanding the analyses.

L338-339: Please add units to the IQR, presumably s.

L339-340: This very last line of the Results section: “There were no significant differences between the CPET and verification phase for VO2max (P = 0.18 to P = 0.71)”… Can you be more specific about what the sentence, and the range of P values, relate to. For example, how does the information differ from that stated in L295-297 (i.e., “Notably, the mean highest VO2 values were similar between the CPET and verification phase [mean difference = 0.03 (95% CI = -0.01 to 0.06) L/min, P = 0.15].”)? Presumably you are referring to the sub-groups, but it’s currently unclear from what you have written.

Discussion:

In general the writing in this section, particularly the new parts in red, need reviewing and revising to improve the clarity of the messages being communicated. The grammar is poor in places, which makes the important information difficult to understand. Also, the Discussion to me is far too long. There are unnecessarily lengthy descriptions of studies and protocols throughout, so I suggest writing more concisely. Also, there are lengthy descriptions relating to points not supported by the current data analysis. In my mind, the authors need to make a significant overhaul of the discussion and cut it down in length, such that the important points and messages are communicated far more clearly.

L357-359: The word “repeated” appears out of place here, since the VP is not repeated (it is only carried out once). Also “compared to that observed” is unclear – what are you comparing to? The maximal effort and VO2max (in which case, plural = those)? Please review and revise this sentence for clarity.

L360: “To date” should not be hyphenated (the term also seems odd/unnecessary). Also, consider re-phrasing: “has demonstrated that… have shown that…”.

L364: “findings from 54 CPET…”. Also, I don’t really understand what this sentence adds to the previous sentence. They seem to say the same thing. Can they be combined?

L366: Please revisit the grammar and revise this sentence.

L369-370: I suggest removing this important information from the brackets and supporting the statement with a reference (i.e., that this is within the most commonly adopted measures of test variability at 2-3%).

L371: Revise the grammar.

L374-375: Revise the grammar.

L383: “for example” appears out of place here, in a new paragraph.

L383-393: This is a long account of a previous study. What point is being made?

L383-403: This is a long paragraph describing a few previous studies and their results, but there is no reference to the overall findings of current study. Please revise.

L404: A new paragraph should not begin with “On the other hand”. Also, is this point (“54 studies meta-analyzed”) now in relation to your meta-analysis? Please clarify. In general, I suggest you revise your paragraph structures and arguments to arrive more quickly at the points you are trying to make, in light of previous work, and link the story to the findings of the current analysis. At the moment I am wondering where all this is going (particularly the paragraph above), and what the relevance is to the findings of your systematic review/meta-analysis.

L406-409: The use of brackets seems inconsistent here too.

L425-426: Please revise the writing here: “since low cardiorespiratory fitness are more susceptible to stopping early during the CPET”

L404-432: This is another very long paragraph describing previous studies in length. I think these points can be made far more concisely and interpreted more clearly in light of the general findings from your study.

L427-432: You seem to have spent a long time describing a possibility that is not actually supported by your own data analysis. I suggest turning your thinking around, and discussing what you did find and what that actually means, in relation to the specific studies analysed. I get your point, but it is not actually supported by your stats, which is the issue. Alternatively, make this point more concisely so that it is not over-inflated.

L433: Due to the shear length of this discussion I have read the remainder with less of a focus on details. As stated above, my advice is that the writing is more clear and focused, with less extensive description of previous studies.

L499-501: Again, is there any data to support Poole and Jones’ suggestion here (i.e., that researchers “must”…), or is it a subjective view (because other research would suggest that this is not necessary)?

Table 1:

Title: (N = 80) – include space; CPET has not been defined here, but it has in the Fig 2 title.

Table 2:

Abbreviations for TR and CYC seem to be missing (I haven’t read every detail of Table 2, so there might be other bits missing).

Table 3:

Can you clarify in the table heading row what is meant by “Total”

For Arad et al., % Weight should probably be written to 2 d.p. (1.40%).

Figure 1:

This is a nice figure. Is it what you refer to in L167?

“Hand searches (reference lists from the previously identified studies)”

Should it be “Records excluded”? (Box to the right in “Screening”?)

Eligibility: can the horizontal arrow stem from the left box?

Figure 2:

Legend on L319: Suggest “reported as mean differences (MD) adjusted for” (and remove the final sentence on this line).

7. PLOS authors have the option to publish the peer review history of their article (what does this mean?). If published, this will include your full peer review and any attached files.

Reviewer #1: No

Reviewer #2: **Yes: **Kerry McGawley

---

## [Author Response · Author response to Decision Letter 1]

24 Dec 2020

Review comments:

Reviewer: 1

I would like to thank the authors for their attention to my comments. I think this revised version is much better and I believe that this manuscript would be an excellent reference on the topic. From my perspective, I only have on minor suggestion that I would leave up to the authors to accept or reject. This is: As presented, the title seems to imply that a verification phase is something necessary. Would it be better crafting the title as a question? Something like “Is a verification phase for confirming maximal oxygen uptake useful in apparently healthy adults: A systematic review and meta-analysis”.

Regardless, I would like to congratulate the authors for the high quality of this meta-analysis. I often feel that this type of scientific contributions are not as meritorious as original research, but I think that with the amount of information that is currently available on this topic, this particular meta-analysis is fully warranted. Thanks!

ANSWER: Thank you for your comments. The title has been revised according to the reviewer’s suggestion.

Reviewer #2: 

General Comments

Preparing this manuscript has clearly been a huge job, so I commend the authors on this significant undertaking. I have again reviewed this paper in thorough detail and have a number of observations and feedback points, specified below.

In general, I would firstly urge the native English speakers on the author list – or a professional proof reader – to take responsibility for thoroughly checking the language (especially the grammar) before resubmitting, as the current level of writing makes the text difficult to comprehend. There are particular issues in the new sections of the Discussion that need re-writing/correcting to improve the clarity and flow.

Secondly, and perhaps most importantly, the Discussion to me is too long. It makes it impossible for the reader to grasp the key findings and messages from this paper, as there is just so much detail of previous studies, their protocols and specific findings (data, P values, etc.). I personally feel that the paper would be far more comprehensible and impactful if the Discussion was more concise.

ANSWER: The discussion section has undergone a comprehensive revision to improve the grammar and make it more concise by removing specific details that provide limited information for the interpretation of the results.

Abstract:

L33-35: The use of semicolons in this list seems odd, and makes it difficult to understand how the search was conducted. Should they be commas? Please re-consider.

ANSWER: The correction has been made. 

L43: Why the comma after VO2max but not after age? Also, the -1 on the VO2max unit looks too high.

ANSWER: The correction has been made.

L44: Can you clarify that the VO2 values were similar in 54 of the 80 studies, because at the moment it seems like this is a result for a total of 54 studies analyzed, which I don’t think is the case. Suggestion: “The highest mean VO2 in the CPET and verification phase was similar in 54 of the 80 studies (mean difference…)”. Also, I still don’t understand the inconsistencies in the use of square/round brackets. Can this be standardized throughout the manuscript? Also, why change to L/min in this sentence, after presenting mL/kg/min in the previous sentence? Can this be standardized?

ANSWER: Thank you for your comment. We have revised the paragraph according to the reviewer’s suggestion and it now reads as follows: “The highest mean VO2 values attained in the CPET and verification phase were similar in the 54 studies that were meta-analyzed (mean difference = 0.03 [95% CI = -0.01 to 0.06] L/min, P = 0.15).” In addition, we have standardized the use of square/round brackets throughout the manuscript. On the other hand, VO2max relative to body weight has been used as descriptive data for defining levels of cardiorespiratory fitness for sub-analysis, whereas absolute values of VO2 have been used for the meta-analysis since most studies assumed L/min. We have now also included mean ± SD VO2 values in L/min in the abstract as follows: “Eighty studies were included in the systematic review (total sample of 1,680 participants; 473 women; age 19-68 yr.; VO2max 3.3 ± 1.4 L/min or 46.9±12.1 mL·kg-1min-1).” 

L51: “following a… CPETs” = incorrect grammar. Maybe write CPET

ANSWER: The correction has been made.

L51-53: I like this idea, but to me it needs to be related specifically to “your” findings, not attributed to “some [other?] researchers”. Maybe something like: “However, given the high concordance between the highest mean VO2 achieved in the CPET and verification phase, findings from the current study would question its necessity in all testing circumstances.”

ANSWER: This sentence has been rewritten according to the reviewer’s suggestion.

Introduction:

L70: Maybe mention the Douglas bag method here, since that is the main reason for the improved time-efficiency (i.e., changing from DBs to B-b-B systems). The type of protocol is merely a by-product.

ANSWER: The correction has been made.

L73-74: This sentence is not quite right. The “confirmation of VO2max attainment” is not “due to” the listed factors; the “lack of attaining” VO2max might be due to those factors. Please re-phrase.

ANSWER: Thank you for the observation. The correction has been made. The sentence now reads as follows: “One particularly problematic aspect has been the challenge in identifying a lack of VO2max attainment due to inappropriate test protocols, premature fatigue, or poor participant motivation and lack of effort.”

L82: Presumably this is average “male” body mass. Please specify (there needs to be transparency when sex biases exist in data sets).

ANSWER: The reviewer is correct, thank you for the comment. The following information has been included: “The landmark study of Taylor et al. [34] was the first to use a formal VO2 plateau criterion, which was defined as an increase in VO2 of less than 150 L/min (or ≤ 2.1 mL·kg-1·min-1, considering an average body mass of 72 kg from 115 male subjects) (…)”

L85-89: Is this a new paragraph? It seems very short (2 sentences). Maybe it should be a continuation of the previous paragraph, since it’s the same topic (VO2 plateau)?

ANSWER: It is the same paragraph that was summarized to attend both reviewers. Lines 78 to 89 describe the same topic – VO2 plateau. This is more clearly shown in the revised manuscript.

L93: “is attained”?

ANSWER: We have revised the text to “has been attained” since VO2max criteria are applied after the VO2max test has been performed.

L95-96: I’m not sure what this means: “lack of specificity in identifying individuals who did not continue the CPET to their limit of exercise tolerance”. What is meant by “specificity” in this context?

ANSWER: The following sentence has been included to address this issue: “Research has shown that some individuals can satisfy some of the secondary criteria thresholds long before the highest VO2 value observed in the CPET has been attained [2, 29, 37, 39]. The maximal RER criterion, for example, can be satisfied at VO2 values 27%-39% lower than the highest VO2 value achieved in the CPET [37, 39].”

L103: CP is specific to power (usually analogous to cycling) and not practically applicable to other exercise modes (such as running). Also, “i.e.” is specific (“that is”). So maybe change i.e. to e.g. (“for example”). Or add speed/velocity to power when referring to CP (so CS/CV).

ANSWER: The correction has been made.

L106: Suggestion (to avoid the use of “each other”): “the highest VO2 values in the CPET are consistent with the verification phase”

ANSWER: The correction has been made and now the sentence reads as follows: “The verification phase is based on the premise that when the highest VO2 values in the CPET are consistent with the verification phase (typically within 2-3% in accordance with the test-retest reliability of VO2max), this provides substantial empirical support that the highest possible VO2 has been elicited. 

L108-111: Please present the “evidence” for Poole & Jones’ statement (i.e., “that a verification phase must be performed at a higher WR than attained in the ramp-based CPET protocols in all future studies”). Where/what is the evidence for this suggestion?

ANSWER: We described the exact term employed by Poole and Jones (2017). However, to avoid further controversy on this issue, we decided to replace “must be” to “should be”.

L111: The terminology “On the other hand” seems inappropriate, as this is a different topic compared with the previous sentence (VP WR vs CPET increments). The correct counter-argument here would presumably be the data showing that VP WR can actually be “lower” than WRpeak to still elicit a VO2peak/max.

ANSWER: This section of the introduction has been considerably revised and includes addressing this issue. 

L113-116: This is now confusing, with “In contrast” following “On the other hand”. Please clarify the arguments that you are making, and the key information you’re trying to get across. I also think WRpeak needs defining. Also, is it necessary to list average W values?

ANSWER: The text has been revised as follows: “Poole and Jones [2] recently stated that to confirm the attainment of VO2max a verification phase should be performed at a higher WR than attained in the CPET in all future studies. Conversely, Iannetta et al. [25] recommended WRs within the upper limit of the severe exercise intensity domain to allow the verification phase to be maintained long enough for VO2max attainment.”

L108-120: In general, this section needs to be more clear and concise.

ANSWER: This section has been revised for conciseness, so it is now only 8 lines.

L120-132: I think all of these ideas can be incorporated into one far more clear and concise paragraph explicitly stating the aims of the study. I suggest moving the questions (at L120-125) down to L138-141 (Methods) and just focusing on the aims here, as required.

ANSWER: This text has been revised according to the reviewer’s suggestion.

Methods:

L136: “is” shown in S1 Text.

ANSWER: The correction has been made.

L138-141: These are not completely consistent with the text in the Introduction (L120-132). Also, you are still using the terminology “valid alternative to confirm”, which I understood had been changed throughout (since this is not a study of validity). And I’m not sure the study does actually identify “the most appropriate protocol for applying the verification phase”. So I would use questions similar to those currently posed in the Introduction (L120-125) here instead.

ANSWER: The correction has been made according to the reviewer’s suggestion as follows: “The main questions addressed by the present study were: To what extent does the highest VO2 attained in the CPET differ from that attained in the verification phase? Secondly, are the highest VO2 values in the CPET and verification phase affected by the verification phase characteristics (e.g. intensity, adoption of a criterion threshold, and aspects of the recovery period between the CPET and the verification phase), or even with respect to particular subgroups (e.g. sex, cardiorespiratory fitness levels, exercise test modality, and CPET protocol design) in apparently healthy adults?”

L155: participants “who” were

ANSWER: The correction has been made.

L157-158: Suggestion: “…carried out using bipedal cycle ergometry or bipedal treadmill running or walking.”

ANSWER: We agree with the reviewer’s comment and the sentence has been rewritten accordingly.

L160: Remove “included” (as you already have “involved”, above). Also, importantly, “and” should presumably be changed to “or”, since any one of those three situations would surely lead to exclusion.

ANSWER: The correction has been made.

L166-167: Writing in the first person seems odd here (we). Are you saying that a flowchart has been included in the paper? If so, where is it? If not, why mention it?

ANSWER: The correction has been made.

L170-174: There is still an inconsistent use of round and square brackets. Please re-consider (round brackets are typically used in the first instance, and [square brackets] within round brackets.)

ANSWER: We have standardized this throughout the manuscript according to your suggestion. 

L175: Can you specify that you mean “other” authors than yourselves, so “authors of the original articles were contacted…”.

ANSWER: Thank you for the observation. We have revised the paragraph according to the reviewer’s suggestion and now it reads as follows: “When the relevant quantitative data were not reported, authors of the original studies were contacted to request the data”.

L185: Does “selective reporting” need to be written twice here?

ANSWER: We have shortened it to “selective reporting of outcomes”.

L187: You are inconsistent with your capitalization (or not) of the sub-title words

ANSWER: The capitalization of the sub-title words is now standardized.

L206: “the primary study groups results” requires an apostrophe somewhere on groups, depending on the specific meaning (one group or multiple groups) – please correct.

ANSWER: The text has been revised to address this issue and now reads as follows: “(mean ± standard deviation [SD] values for group VO2max and protocol duration during the CPET and verification phase from primary study groups)”.

L207: Same here – “groups” needs an apostrophe.

ANSWER: The text has been revised to address this issue and now reads as follows: “The I2 statistic measures the extent of inconsistency among the results of the primary study groups, interpreted approximately as the proportion of total variation in point estimates that is due to heterogeneity rather than sampling error.”

L208: The less than or equal to symbol is still odd here. The lower “equal to” line should be horizontal.

ANSWER: The correction has been made.

L209: using “a” funnel plot, or using funnel “plots”?

ANSWER: We have added the word “a” funnel plot. Thank you for the observation.

L216: This new red text is unnecessarily complicated. Firstly, you have previously used the term WRpeak (although this doesn’t seem to have been defined anywhere in the manuscript). Secondly, sub and supra peak does not need to be reinforced with < or > 100%, that’s obvious by definition. Suggestion: (i.e. < 100% WRpeak vs. > 100% WRpeak)

ANSWER: The correction has been made.

L218-220: This sentence could be clearer, suggestion: “as the CPET or on a different day, and the duration of the verification phase (i.e. ≤ 80 s, 81–120 s, > 120 s).”

ANSWER: The correction has been made.

L222: Should this be “cut-off points”?

ANSWER: The correction has been made.

Results:

L229-230: This (“Figure 1 summarizes the screening and selection process”) seems like repetition from the Methods section (L166-167). Should the flowchart and reference to it be moved up to the Methods (L167)?

ANSWER: The flowchart has been moved up to the Methods section.

L234: Presumably your sub-section headings should use a different font/presentation from your main section headings.

ANSWER: The sub-titles have been standardized throughout the manuscript.

L235-239: In L235 you use the term “eligible studies”, in L236 you write “included studies” and in L239 you write “primary studies”. This is confusing, because I think you are talking about the same 80 studies in all cases. Can you use consistent terminology for clarity?

ANSWER: “Primary” has been removed and “eligible” has been replaced by “included” to keep the text standardized. You are correct, this means the same 80 studies.

L240-241: BMI has already been defined in the paper, so I suggest: “participants had a BMI within the normal range (mean ± SD [range]: 24.4…”. And once again, you need to be consistent with the use of round/square brackets.

ANSWER: We have incorporated your suggestion and square and round brackets are now standardized throughout the manuscript.

L242: VO2max does not need to be written twice; I suggest removing the first reference to it: “cardiorespiratory fitness (VO2max mean ± SD [range]: 46.9…”.

ANSWER: We have incorporated your suggestion.

L243: Delete “according to… [53]” as these have already been defined in the Methods (L222-223).

ANSWER: This sentence has been removed.

L252-256: I think this could be clearer, e.g.: “Thirty-three (41%) of the 80 studies included in the review adopted one or more of the traditionally reported plateau or secondary criteria to confirm the attainment of a VO2max, with 30 using a VO2 plateau criteria, 21 using the heart rate plateau or age-predicted maximal heart rate, 18 using RERmax, and 8 using post-CPET blood lactate concentration.”. Line 256 is missing a full stop in any case.

ANSWER: We have attempted to make this text clearer. The text now reads as follows: “Thirty-three (41%) of the 80 studies included in the review used one or more VO2 plateau or secondary VO2max criteria to confirm the attainment of VO2max. Thirty studies used the VO2 plateau, 21 used the heart rate plateau or a criterion based on age-predicted maximal heart rate, 18 used the RERmax, and 8 used the post-CPET blood lactate concentration.”

L258: I don’t think the second “time averages” should be hyphenated.

ANSWER: The hyphen has been removed.

L263: Should this read something like “Regarding the period between CPET and VP…”? Because you seem to be referring to the rest period after the CPET here, right?

ANSWER: We have incorporated your suggestion.

L266: What do you mean by “self-paced approach”? A self-paced recovery period? Of any duration?

ANSWER: This text has been revised and now reads as follows: “Two studies (3%) employed a combination of passive and active recovery and another (1%) used a self-paced approach where participants were permitted to choose their own WR.”

L270-271: You need to be clearer that you are now referring to the VP exercise intensity. I also think you should decide upon the terminology earlier in the paper, define it and stick to it (e.g. peak WR or WRpeak, supra or > 100%, etc.). Because at the moment there is a mix.

ANSWER: The revised text now specifically refers to the verification phase. Consistent terminology also has now been used for supra in instead of > 100% and for WRpeak instead of peak WR.

L272: Maybe write 105-130%, consistent with the presentation of ranges in the previous paragraph.

ANSWER: The correction has been made.

L273: What do you mean by “or both peak or supra peak WR (1%)”? Is this one study, a different study from the “Eight studies” previously stated? If so, please specify. Also, do you mean both peak “and” supra peak? This is unclear to me.

ANSWER: The study of Wingo et al. (Medicine & Science in Sports & Exercise, v. 37, n. 2, p. 248-255, 2005) stated the following sentence “To ensure that a plateau in VO2 was attained, subjects completed an additional bout of cycling following 20 min of rest. Subjects cycled to exhaustion at a power output equivalent to the last workload performed during the graded test (if ≤ 1 min was completed during the last stage of the graded test) or at a power output 25 W higher than the last workload performed during the graded test (if ≥ 1 min was completed during the last stage of the graded test)”. This means that the usage of either peak (100% WRpeak) or supra peak WR (>100% WRpeak) depended on the duration of the last CPET stage. Another study (COLAKOGLU, Muzaffer et al. Stroke volume responses may be related to the gap between peak and maximal O2 consumption. Isokinetics and Exercise Science, v. 24, n. 2, p. 133-139, 2016.) also applied both peak and supra peak WRs. Considering this, the sentence has been rewritten and we think it is now clearer. 

L275: Again, does 1% relate to one study? If so, you could maybe write (one study, 1%) for clarity/consistency. Also, you have previously used the term “participant” rather than “subject”.

ANSWER: “One study” has been added. “Participants” is now used consistently throughout the manuscript,

L277: “85-95% of…”

ANSWER: The correction has been made.

L281-282: “1.5-2.2”; ”50-150” (consistent with previously stated ranges).

ANSWER: The correction has been made.

L294: Figure 2 (not Figures).

ANSWER: The correction has been made.

L322: I suggest removing the two commas, as this “middle” information is imperative to understanding the analyses.

ANSWER: The correction has been made.

L338-339: Please add units to the IQR, presumably s.

ANSWER: The units have been added.

L339-340: This very last line of the Results section: “There were no significant differences between the CPET and verification phase for VO2max (P = 0.18 to P = 0.71)”… Can you be more specific about what the sentence, and the range of P values, relate to. For example, how does the information differ from that stated in L295-297 (i.e., “Notably, the mean highest VO2 values were similar between the CPET and verification phase [mean difference = 0.03 (95% CI = -0.01 to 0.06) L/min, P = 0.15].”)? Presumably you are referring to the sub-groups, but it’s currently unclear from what you have written.

ANSWER: We have added the following sentence “Considering all sub-analyses presented in the Table 4…”. By looking at the Table 4, we can see the minimal P value (0.18 for the high cardiorespiratory fitness) and the maximal P value (0.71 for female subjects). 

Discussion:

In general the writing in this section, particularly the new parts in red, need reviewing and revising to improve the clarity of the messages being communicated. The grammar is poor in places, which makes the important information difficult to understand. Also, the Discussion to me is far too long. There are unnecessarily lengthy descriptions of studies and protocols throughout, so I suggest writing more concisely. Also, there are lengthy descriptions relating to points not supported by the current data analysis. In my mind, the authors need to make a significant overhaul of the discussion and cut it down in length, such that the important points and messages are communicated far more clearly.

ANSWER: The discussion section has been comprehensively revised to improve the grammar and make the it more concise, including removal of the unnecessary descriptions of studies.

L357-359: The word “repeated” appears out of place here, since the VP is not repeated (it is only carried out once). Also “compared to that observed” is unclear – what are you comparing to? The maximal effort and VO2max (in which case, plural = those)? Please review and revise this sentence for clarity.

ANSWER: The correction has been made. We have replaced “repeated” by “carried out” and “that” by “highest VO2” observed in the preceding CPET, which is the dependent variable measured in the two protocols.

L360: “To date” should not be hyphenated (the term also seems odd/unnecessary). Also, consider re-phrasing: “has demonstrated that… have shown that…”.

ANSWER: Thank you for the observation. We have removed “to-date” from the sentence.

L364: “findings from 54 CPET…”. Also, I don’t really understand what this sentence adds to the previous sentence. They seem to say the same thing. Can they be combined?

ANSWER: The first sentence refers to results from the systematic review that includes all 80 studies, while the second sentence refers to the meta-analyzed data that includes only 54 of the 80 studies). It means that the qualitative results (systematic review) support the quantitative ones (meta-analysis). A hypothetical situation could happen whereby data from the systematic review could indicate VO2max attainment in both protocols that are not be confirmed through the statistical analysis. Although the two sentences appear similar, they are reinforcing the consistency of results from a different amount of studies in each case (80 vs. 54).

L366: Please revisit the grammar and revise this sentence.

ANSWER: This sentence has been removed from the revised discussion section.

L369-370: I suggest removing this important information from the brackets and supporting the statement with a reference (i.e., that this is within the most commonly adopted measures of test variability at 2-3%).

ANSWER: The correction has been made. 

L371: Revise the grammar.

ANSWER: This text has been removed from the revised discussion section.

L374-375: Revise the grammar.

ANSWER: This text has been removed from the revised discussion section.

L383: “for example” appears out of place here, in a new paragraph.

ANSWER: This text has been removed from the revised discussion section.

L383-393: This is a long account of a previous study. What point is being made?

ANSWER: This relevant study by Day et al. reported that traditional VO2max criteria are protocol dependent (the plateau occurrence is not a commonly observed phenomenon in continuous step-incremented tests and even less so in individuals with low cardiorespiratory fitness). In contrast, a verification bout induced no difference in VO2 compared to that observed in the CPET. In summary, the verification phase does not appear to be affected by protocol design and is not population dependent. This is an important point and the findings by Day et al. are consistent with the results from the current meta-analysis, because all sub-analyses indicated no differences according to protocol and individual characteristics. 

L383-403: This is a long paragraph describing a few previous studies and their results, but there is no reference to the overall findings of current study. Please revise.

ANSWER: Rossiter et al. investigated similar questions as those by Day et al., which already have been commented on in our previous answer.

L404: A new paragraph should not begin with “On the other hand”. Also, is this point (“54 studies meta-analyzed”) now in relation to your meta-analysis? Please clarify. In general, I suggest you revise your paragraph structures and arguments to arrive more quickly at the points you are trying to make, in light of previous work, and link the story to the findings of the current analysis. At the moment I am wondering where all this is going (particularly the paragraph above), and what the relevance is to the findings of your systematic review/meta-analysis.

ANSWER: “On the other hand” has been removed from the sentence. The 54 studies are from our meta-analysis. The next paragraphs discuss a small quantity of the included studies (~11%) that found significant mean differences between the CPET and verification phase. Moreover, we tried to explain the possible reasons for these differences. Forty-eight out of 54 studies showed, by analyzing individually, no difference in the highest VO2 between the CPET and verification phase, being consistent with the present meta-analysis. We have comprehensively revised the discussion section to more quickly arrive at the points being made to address the reviewer’s comment.

L406-409: The use of brackets seems inconsistent here too.

ANSWER: The round and square brackets are now standardized throughout the manuscript.

L425-426: Please revise the writing here: “since low cardiorespiratory fitness are more susceptible to stopping early during the CPET”

ANSWER: We have added “individuals with” low...

L404-432: This is another very long paragraph describing previous studies in length. I think these points can be made far more concisely and interpreted more clearly in light of the general findings from your study.

ANSWER: We have removed the unnecessary detail as part of the comprehensive revision of the discussion section.

L427-432: You seem to have spent a long time describing a possibility that is not actually supported by your own data analysis. I suggest turning your thinking around, and discussing what you did find and what that actually means, in relation to the specific studies analysed. I get your point, but it is not actually supported by your stats, which is the issue. Alternatively, make this point more concisely so that it is not over-inflated.

ANSWER: The percentages are from our data analysis. The difference across cardiorespiratory fitness levels was not significant, supporting the use of the verification phase regardless of individual characteristics, which contrasts with the low occurrence of the VO2 plateau during a CPET in subjects with low cardiorespiratory fitness. 

L433: Due to the shear length of this discussion I have read the remainder with less of a focus on details. As stated above, my advice is that the writing is more clear and focused, with less extensive description of previous studies.

ANSWER: The discussion section has undergone a comprehensive revision to improve the grammar and make it more concise by removing specific details that provide limited information for the interpretation of the results.

L499-501: Again, is there any data to support Poole and Jones’ suggestion here (i.e., that researchers “must”…), or is it a subjective view (because other research would suggest that this is not necessary)?

ANSWER: The correction has been made.

Table 1:

Title: (N = 80) – include space; CPET has not been defined here, but it has in the Fig 2 title.

ANSWER: A space has been included and CPET has now been defined. 

Table 2:

Abbreviations for TR and CYC seem to be missing (I haven’t read every detail of Table 2, so there might be other bits missing).

ANSWER: We have double-checked the abbreviations to ensure there are none missing. 

Table 3:

Can you clarify in the table heading row what is meant by “Total”

ANSWER: “Total” means the number of participants that have performed both CPET and the verification phase. As we used Review Manager to calculate the statistics, the software layout requires input data in this order: mean, standard deviation and number of participants for each intervention.

For Arad et al., % Weight should probably be written to 2 d.p. (1.40%).

ANSWER: Thank you, the correction has been made.

Figure 1:

This is a nice figure. Is it what you refer to in L167?

ANSWER: Yes. This figure is a flowchart with the process of inclusion and exclusion of the studies.

“Hand searches (reference lists from the previously identified studies)”

ANSWER: Thank you. The correction has been made.

Should it be “Records excluded”? (Box to the right in “Screening”?)

ANSWER: Yes. We have changed to “Records excluded”. 

Eligibility: can the horizontal arrow stem from the left box?

ANSWER: Figure 1 has been formatted according to PRISMA guidelines and we would prefer to keep it in its current format.

Figure 2:

Legend on L319: Suggest “reported as mean differences (MD) adjusted for” (and remove the final sentence on this line).

ANSWER: We have incorporated your suggestion.

---

## [Decision Letter · Decision Letter 2]

14 Jan 2021

PONE-D-20-25408R2

Is a verification phase useful for confirming maximal oxygen uptake in apparently healthy adults? A systematic review and meta-analysis

PLOS ONE

Dear Dr. Cunha,

Thank you for submitting your manuscript to PLOS ONE. After careful consideration, we feel that it has merit but does not fully meet PLOS ONE’s publication criteria as it currently stands. Therefore, we invite you to submit a revised version of the manuscript that addresses the  minor points raised during the review process.

We look forward to receiving your revised manuscript.

Kind regards,

Laurent Mourot

Academic Editor

PLOS ONE

Reviewers' comments:

Reviewer's Responses to Questions

**Comments to the Author**

1. If the authors have adequately addressed your comments raised in a previous round of review and you feel that this manuscript is now acceptable for publication, you may indicate that here to bypass the “Comments to the Author” section, enter your conflict of interest statement in the “Confidential to Editor” section, and submit your "Accept" recommendation.

Reviewer #2: (No Response)

2. Is the manuscript technically sound, and do the data support the conclusions?

Reviewer #2: Yes

3. Has the statistical analysis been performed appropriately and rigorously? 

Reviewer #2: Yes

4. Have the authors made all data underlying the findings in their manuscript fully available?

Reviewer #2: Yes

5. Is the manuscript presented in an intelligible fashion and written in standard English?

Reviewer #2: Yes

6. Review Comments to the Author

Reviewer #2: Congratulations on producing a great paper! I have once again read the manuscript in detail and have a series of small suggestions. Most reflections relate to the Discussion, but I am happy to accept the manuscript following a final revision by the authors.

Abstract:

Really clear.

Introduction:

Really informative. Just a few small suggestions…

L68: “fast-responding” – NOTE that this rule should be applied throughout the manuscript in my opinion… two words forming an adjective to describe a subsequent noun should be hyphenated (there are lots of examples, but e.g. verification-phase characteristics, verification-phase duration, highly-trained endurance athletes, test-protocol dependent, 10-min recovery, different-day variability etc.).

L100: 27-39%

L117: Could you just double check whether WRpeak has already been defined. If not then I suggest you explicitly define the abbreviation here.

Methods:

Really clear. Just a few small suggestions…

L135: Suggest writing the PROSPERO weblink and reg number in one single sentence.

L161: This final criteria seems redundant when you have specified (at L158) “bipedal treadmill running or walking”. I would remove it (otherwise the reader could wonder why you haven’t included outdoor/track cycling as an exclusion criteria).

L204: You are missing a full stop.

Results:

Really thorough. Some simple suggestions to consider…

L231-233: “All potential… for eligibility”. These two sentences seem like methods. I suggest removing them.

L263-266: This makes 82 studies in total (103%). I suggest that the two studies using both are not included in the other lists, so that they are not duplicated (if that’s what’s happened) and so that the total = 80. If not then at least re-phrase “and another two”, because that’s not strictly true. It’s two of those already mentioned (so something like “two of which”). I would actually re-order the list to have continuous, then discontinuous, then both cont+discont, then self-paced, just to improve the logical flow for the reader. But it’s up to you!

L269: Could you double check that the RERmax abbreviation has been defined already. If not then I suggest you explicitly define it here.

L271-275: It’s a bit unclear why you’ve included %ages for the first two examples, but not throughout the rest of the sentence (especially when %ages are included in the previous and following sentences). Maybe it’s all the small numbers, but something to re-consider.

L293: Forty-two studies (53%) employed

L296: researchers’ laboratories

L313: were judged to have a low risk of bias…

Discussion:

I have some minor suggestions and reflections…

L362-363: of these studies, 90% of which have been published since 2009.

L363: I would probably remove “of the review”, because it’s both a review and a MA.

L367-369: I suggest removing these two sentences as both seem out of place, with neither relating directly to the aims of the study. Firstly, you didn’t explicitly investigate safety across fitness groups. And the second sentence implies that you could run a VP as a stand-alone test to measure VO2max, which you didn’t investigate (and shouldn’t be implied, as the effect of the previous CPET is a confounder).

L373: Is “error” the correct term here? Isn’t it just a “difference”?

L375: You didn’t measure agreement, specifically. Is a better term “similarity”.

L379-382: Have you got this the wrong way round? Are low fitness groups not “less” likely to exhibit a VO2 plateau as a result of VO2 not decelerating?

L385: When you write “mean VO2 values”, do you actually mean “mean VO2max values”? There is of course an important difference between mean VO2 and mean VO2max. Also in this sentence, you should probably write “by” 0.03 and 0.04 L/min (not “of”).

L386-389: This sentence is confusing. Suggestion: “However, sub-group analyses revealed that while maximal VO2 in the CPET was higher than that attained in the verification phase for participants with moderate and high cardiorespiratory fitness, the opposite was true for those with lower cardiorespiratory fitness.”

L394: Same here as in L385, do you mean “mean VO2max”, rather than just “mean VO2”?

L399: Regarding verification phase… (or “In regards to…”)

L402: who performed five repeated treadmill CPET trials appended by a verification phase…

L406: I would remove the term “inappropriately”, unless you can support the inappropriateness clearly with evidence/a reference.

L408-409: which allowed sufficient time (i.e., ~ XXX s) for VO2max attainment [can you include the VP duration here, to clarify how much greater it was than the previously highlighted 80 s].

L412-413: found no difference for verification phase durations of ≤ 80 s, 81-120 s, and > 120 s

L414: Again, where is the justification for deeming this duration “inappropriate”? Please provide evidence/a reference.

L422: 1-hr should not be hyphenated.

L423-424: What about fatigue?

L426: The opposite is also plausible, though, as prior exercise (warm-up/priming) is known to have favorable effects on VO2. So I think you need to justify the greater advantage of no prior “fatiguing” exercise compared to no warm-up.

L436: Would “effectiveness” be better than “utility”?

L440: Suggest inserting a comma after [128]

L441: Suggest removing the comma after “phase”

L442: suggested that researchers…

L448: is sufficiently long.

L456-458: To me the problem seems to be more an issue of confounding results rather than limited data. There are loads of studies, as you’ve highlighted, but they all show different results!

L458: evidence-based

L459: 10-20 min

L463: Where is the evidence that this “might be better tolerated”? Is there evidence that performing the VP on the same day is not well tolerated? I would remove this idea and just write “An alternative method is to perform the VP on a separate day…”

L498: Suggest changing “patient” to “participant”, as this is not just relevant to clinical populations, but also healthy/athletes, etc.

L500: I don’t understand the term “would be ideally indicated in”, in this context. Something like “is applicable to” might be better.

L502: Does “those” refer to the wheelchair athletes? Presumably not, so maybe write more clearly: “individuals with spinal-cord injuries”.

L504-505: This seems repetitive from L499-500, so I suggest removing this sentence.

Tables:

Table 3 heading: Would it be better as “Overall comparisons in the meta-analyzed studies…”

Table 4 (see also L242-243): I might have missed it, but have you referenced/justified your CR fitness level classifications (Low, Moderate, High) in the Methods? Please check and add if necessary.

7. PLOS authors have the option to publish the peer review history of their article (what does this mean?). If published, this will include your full peer review and any attached files.

Reviewer #2: **Yes: **Kerry McGawley

---

## [Author Response · Author response to Decision Letter 2]

16 Jan 2021

Reviewer #2: 

General Comments

Reviewer #2: Congratulations on producing a great paper! I have once again read the manuscript in detail and have a series of small suggestions. Most reflections relate to the Discussion, but I am happy to accept the manuscript following a final revision by the authors.

ANSWER: Thank you for your comments and for the obvious substantial time and effort you have given to help improve our paper.

Abstract: 

Really clear.

ANSWER: Thank you for your comment.

Introduction:

Really informative. Just a few small suggestions…

L68: “fast-responding” – NOTE that this rule should be applied throughout the manuscript in my opinion… two words forming an adjective to describe a subsequent noun should be hyphenated (there are lots of examples, but e.g. verification-phase characteristics, verification-phase duration, highly-trained endurance athletes, test-protocol dependent, 10-min recovery, different-day variability etc.).

ANSWER: You are correct. The corrections have been made.

L100: 27-39%

ANSWER: The alteration has been made.

L117: Could you just double check whether WRpeak has already been defined. If not then I suggest you explicitly define the abbreviation here.

ANSWER: Thank you. We have included its definition: […] a verification phase should be performed at a higher WR than the last load attained in the CPET (i.e. > WRpeak) […].

Methods: 

Really clear. Just a few small suggestions…

L135: Suggest writing the PROSPERO weblink and reg number in one single sentence.

ANSWER: We have incorporated your suggestion.

L161: This final criteria seems redundant when you have specified (at L158) “bipedal treadmill running or walking”. I would remove it (otherwise the reader could wonder why you haven’t included outdoor/track cycling as an exclusion criteria).

ANSWER: Thank you. We have removed the final criterion.

L204: You are missing a full stop.

ANSWER: A full stop has been added.

Results:

Really thorough. Some simple suggestions to consider…

L231-233: “All potential… for eligibility”. These two sentences seem like methods. I suggest removing them.

ANSWER: The two sentences have been removed.

L263-266: This makes 82 studies in total (103%). I suggest that the two studies using both are not included in the other lists, so that they are not duplicated (if that’s what’s happened) and so that the total = 80. If not then at least re-phrase “and another two”, because that’s not strictly true. It’s two of those already mentioned (so something like “two of which”). I would actually re-order the list to have continuous, then discontinuous, then both cont+discont, then self-paced, just to improve the logical flow for the reader. But it’s up to you!

ANSWER: We have revised the sentence according to your suggestion and it now reads as follows: “Seventy-three studies (91%) used continuous step-incremented or ramp/pseudo-ramp CPET protocols. Three (4%) used only discontinuous step-incremented protocols. Two studies (3%) used both discontinuous and continuous step-incremented protocols and another two studies (3%) applied self-paced protocols.” 

L269: Could you double check that the RERmax abbreviation has been defined already. If not then I suggest you explicitly define it here.

ANSWER: We have included its definition: […] 18 used the maximal RER attained in the CPET (RERmax) […].

L271-275: It’s a bit unclear why you’ve included %ages for the first two examples, but not throughout the rest of the sentence (especially when %ages are included in the previous and following sentences). Maybe it’s all the small numbers, but something to re-consider.

ANSWER: We have added the percentages so every count in the paragraph has a corresponding percentage.

L293: Forty-two studies (53%) employed

ANSWER: The alteration has been made.

L296: researchers’ laboratories

ANSWER: The alteration has been made.

L313: were judged to have a low risk of bias…

ANSWER: The alteration has been made.

Discussion: 

I have some minor suggestions and reflections…

ANSWER: OK.

L362-363: of these studies, 90% of which have been published since 2009.

ANSWER: The alteration has been made.

L363: I would probably remove “of the review”, because it’s both a review and a MA.

ANSWER: “Review” has been removed.

L367-369: I suggest removing these two sentences as both seem out of place, with neither relating directly to the aims of the study. Firstly, you didn’t explicitly investigate safety across fitness groups. And the second sentence implies that you could run a VP as a stand-alone test to measure VO2max, which you didn’t investigate (and shouldn’t be implied, as the effect of the previous CPET is a confounder).

ANSWER: These sentences have been removed.

L373: Is “error” the correct term here? Isn’t it just a “difference”?

ANSWER: We have changed “error” to “difference”. 

L375: You didn’t measure agreement, specifically. Is a better term “similarity”.

ANSWER: We have changed “agreement” to “similarity”. 

L379-382: Have you got this the wrong way round? Are low fitness groups not “less” likely to exhibit a VO2 plateau as a result of VO2 not decelerating?

ANSWER: You are correct. Thank you for spotting this error. The correction has been made.

L385: When you write “mean VO2 values”, do you actually mean “mean VO2max values”? There is of course an important difference between mean VO2 and mean VO2max. Also in this sentence, you should probably write “by” 0.03 and 0.04 L/min (not “of”).

ANSWER: We have written “mean VO2max values”. The word “mean” means the highest VO2 the individuals could achieve. The correction has been made.

L386-389: This sentence is confusing. Suggestion: “However, sub-group analyses revealed that while maximal VO2 in the CPET was higher than that attained in the verification phase for participants with moderate and high cardiorespiratory fitness, the opposite was true for those with lower cardiorespiratory fitness.”

ANSWER: Thank you. We have incorporated your suggestion.

L394: Same here as in L385, do you mean “mean VO2max”, rather than just “mean VO2”?

ANSWER: We have revised the text to include “mean VO2max”.

L399: Regarding verification phase… (or “In regards to…”)

ANSWER: The text has been changed to “Regarding verification intensity…”

L402: who performed five repeated treadmill CPET trials appended by a verification phase…

ANSWER: The alteration has been made.

L406: I would remove the term “inappropriately”, unless you can support the inappropriateness clearly with evidence/a reference.

ANSWER: We have removed “inappropriately” as suggested.

L408-409: which allowed sufficient time (i.e., ~ XXX s) for VO2max attainment [can you include the VP duration here, to clarify how much greater it was than the previously highlighted 80 s].

ANSWER: The suggested information has been added.

L412-413: found no difference for verification phase durations of ≤ 80 s, 81-120 s, and > 120 s

ANSWER: The correction has been made.

L414: Again, where is the justification for deeming this duration “inappropriate”? Please provide evidence/a reference.

ANSWER: We have removed the term “inappropriate”, consistent with our response to your previous comment on this issue.

L422: 1-hr should not be hyphenated.

ANSWER: The correction has been made.

L423-424: What about fatigue?

ANSWER: We have added information on fatigue and the sentence now reads as follows: “It is feasible that the procedures performed before the maximal CPET may have led to poor participant motivation, lack of effort and premature fatigue in the following test”.

L426: The opposite is also plausible, though, as prior exercise (warm-up/priming) is known to have favorable effects on VO2. So I think you need to justify the greater advantage of no prior “fatiguing” exercise compared to no warm-up.

ANSWER: The aim of this sentence is to highlight the potential effect of residual fatigue from the maximal CPET on subsequent verification phase performance. We have added the word “maximal” to the sentence to help clarify this.

L436: Would “effectiveness” be better than “utility”?

ANSWER: We have incorporated your suggestion.

L440: Suggest inserting a comma after [128]

ANSWER: The comma has been inserted.

L441: Suggest removing the comma after “phase”

ANSWER: The comma has been removed.

L442: suggested that researchers…

ANSWER: The correction has been made.

L448: is sufficiently long.

ANSWER: The correction has been made.

L456-458: To me the problem seems to be more an issue of confounding results rather than limited data. There are loads of studies, as you’ve highlighted, but they all show different results!

ANSWER: We have changed “limited data” to “confounding results”.

L458: evidence-based

ANSWER: The alteration has been made.

L459: 10-20 min

ANSWER: The correction has been made.

L463: Where is the evidence that this “might be better tolerated”? Is there evidence that performing the VP on the same day is not well tolerated? I would remove this idea and just write “An alternative method is to perform the VP on a separate day…”

ANSWER: We have incorporated your suggestion.

L498: Suggest changing “patient” to “participant”, as this is not just relevant to clinical populations, but also healthy/athletes, etc.

ANSWER: We have incorporated your suggestion.

L500: I don’t understand the term “would be ideally indicated in”, in this context. Something like “is applicable to” might be better.

ANSWER: We have incorporated your suggestion.

L502: Does “those” refer to the wheelchair athletes? Presumably not, so maybe write more clearly: “individuals with spinal-cord injuries”.

ANSWER: We have incorporated your suggestion.

L504-505: This seems repetitive from L499-500, so I suggest removing this sentence.

ANSWER: We have removed this sentence.

Tables: 

Table 3 heading: Would it be better as “Overall comparisons in the meta-analyzed studies…”

ANSWER: We have incorporated your suggestion.

Table 4 (see also L242-243): I might have missed it, but have you referenced/justified your CR fitness level classifications (Low, Moderate, High) in the Methods? Please check and add if necessary.

ANSWER: We provided the Astorino et al. 2015 reference for the CR classifications in line 226 (line 224 in the revised manuscript).

---

## [Decision Letter · Decision Letter 3]

1 Feb 2021

Is a verification phase useful for confirming maximal oxygen uptake in apparently healthy adults? A systematic review and meta-analysis

PONE-D-20-25408R3

Dear Dr. Cunha,

We’re pleased to inform you that your manuscript has been judged scientifically suitable for publication and will be formally accepted for publication once it meets all outstanding technical requirements.

Kind regards,

Laurent Mourot

Academic Editor

PLOS ONE

Additional Editor Comments (optional):

Reviewers' comments:

Reviewer's Responses to Questions

**Comments to the Author**

1. If the authors have adequately addressed your comments raised in a previous round of review and you feel that this manuscript is now acceptable for publication, you may indicate that here to bypass the “Comments to the Author” section, enter your conflict of interest statement in the “Confidential to Editor” section, and submit your "Accept" recommendation.

Reviewer #2: All comments have been addressed

2. Is the manuscript technically sound, and do the data support the conclusions?

Reviewer #2: Yes

3. Has the statistical analysis been performed appropriately and rigorously? 

Reviewer #2: Yes

4. Have the authors made all data underlying the findings in their manuscript fully available?

Reviewer #2: Yes

5. Is the manuscript presented in an intelligible fashion and written in standard English?

Reviewer #2: Yes

6. Review Comments to the Author

Reviewer #2: Well done, I look forward to seeing this paper in print.

7. PLOS authors have the option to publish the peer review history of their article (what does this mean?). If published, this will include your full peer review and any attached files.

Reviewer #2: **Yes: **Kerry McGawley

---

## [Editor Report · Acceptance letter]

4 Feb 2021

PONE-D-20-25408R3 

Is a verification phase useful for confirming maximal oxygen uptake in apparently healthy adults? A systematic review and meta-analysis

Dear Dr. Cunha:

I'm pleased to inform you that your manuscript has been deemed suitable for publication in PLOS ONE. Congratulations! Your manuscript is now with our production department. 

Kind regards, 

on behalf of

Dr Laurent Mourot 

Academic Editor

PLOS ONE